# Conflict-Aware Additive Guidance for Flow Models under Compositional Rewards

Xuehui Yu [1]   Fucheng Cai [2]   Meiyi Wang [3]   Xiaopeng Fan [2]   Harold Soh [1 3]

## Abstract

Inference-time guided sampling steers state-of-the-art diffusion and flow models without fine-tuning by interpreting the generation process as a controllable trajectory. This provides a simple and flexible way to inject external constraints (e.g., cost functions or pre-trained verifiers) for controlled generation. However, existing methods often fail when composing multiple constraints simultaneously, which leads to deviations from the true data manifold. In this work, we identify root causes of this off-manifold drift and find that the approximation error scales severely with gradient misalignment. Building on these findings, we propose Conflict-Aware Additive Guidance ($g^{car}$), a lightweight and learnable method, which actively rectifies off-manifold drift by dynamically detecting and resolving gradient conflicts. We validate $g^{car}$ across diverse domains, ranging from synthetic datasets and image editing to generative decision-making for planning and control. Our results demonstrate that $g^{car}$ effectively rectifies off-manifold drift, surpassing baselines in generation fidelity while using light compute. Code is available at `github.com/yuxuehui/CAR-guidance`.

## 1. Introduction

Continuous-time flow models, such as Rectified Flow (Liu et al., 2023a), Flow Matching (Lipman et al., 2023; Tong et al., 2024a), and Stochastic Interpolants (Albergo et al., 2023), have emerged as a simple yet highly effective generative modeling paradigm. Through data-driven training at

[1]Smart Systems Institute, National University of Singapore, Singapore [2]Faculty of Computing, Harbin Institute of Technology, Harbin, China [3]School of Computing, National University of Singapore, Singapore. Correspondence to: Xuehui Yu <yuxuehui@nus.edu.sg>.

*Proceedings of the 43rd International Conference on Machine Learning*, Seoul, South Korea. PMLR 306, 2026. Copyright 2026 by the author(s).

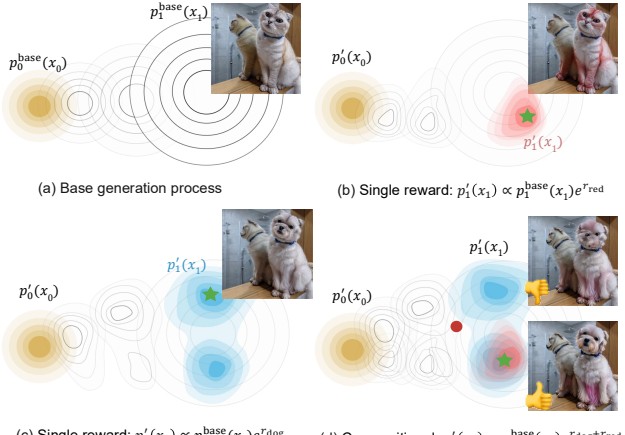

*Figure 1.* At inference time, our goal is to sample from the tilted target $p'_1(x_1) \propto p_1^{base}(x_1)e^{r(x_1)}$. (b-c) A single reward reweights the distribution towards specific attributes ("red" or "dog"). (d) Under compositional rewards ("red" and "dog"), the ideal samples lie at the intersection of high-reward regions (⋆, 👍); however, existing methods often suffer from off-manifold drift (i.e., the distorted image, ●, 👎).

scale, these models acquire a robust generative prior capable of generalizing across a wide spectrum of applications.

In this work, we study the problem of steering powerful generative priors to satisfy multiple, potentially competing objectives simultaneously, a setting commonly referred to as the **compositional reward problem** (Du & Kaelbling, 2024). This challenge is central to the real-world deployment of large-scale generative flow models, where inference-time samples are required to satisfy a diverse and often heterogeneous set of runtime constraints. For example, in generative decision-making, generated trajectories must respect heterogeneous requirements, including safety constraints (Eiras et al., 2022), trajectory smoothness (Urain et al., 2023), and dynamic consistency with learned world models (Du & Song, 2025). Similarly, in text-guided image manipulation, one often seeks to exploit large vision–language models such as CLIP (Radford et al., 2021) to modify images according to natural language prompts (Yu et al., 2023).

Inference-time alignment methods (Lipman et al., 2023)

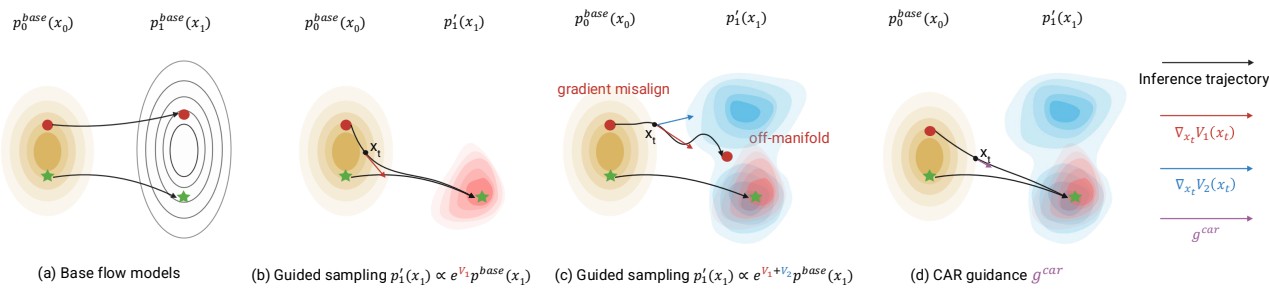

*Figure 2.* (a) The base sampling trajectories. (b) Guided sampling adds guidance to inference trajectories and with a fixed source $p^{\text{base}}(x_0)$; this forces the trajectories to curve significantly to satisfy the constraint, resulting in unnecessarily long and high-curvature paths. (c) Gradient misalignment (between → and →) aggravates local curvature, yielding incorrect and unstable guidance that pushes the trajectory off-manifold (•) into low-density regions (i.e., the "vanishing energy" traps visualised in Figure 4). (d) Under gradient conflict, $g^{\text{car}}$ rectifies the misaligned gradient via learned guidance (→) that points directly to the target $x_1$, recovering straight and on-manifold trajectories.

address these challenges by encoding task requirements as reward functions and steering the sampling process directly at inference time, without retraining or fine-tuning the underlying generative prior. Among these, inference-time guided sampling (or guidance) offers a lightweight and flexible interface for adapting pretrained generative models to complex and heterogeneous constraints by directly injecting reward signals during generation, which has been successfully applied to enforce a wide range of runtime constraints (Lipman et al., 2023; Pokle et al., 2024). In particular, their compute-efficient approximate forms even enable control over objectives unseen during training through heuristic guidance terms (Feng et al., 2025).

However, these approximate guidance methods often push samples into low-density regions where the pretrained vector field is poorly calibrated, causing systematic deviations from the true data distribution. This failure mode, commonly referred to as **off-manifold drift**, becomes particularly severe in compositional reward settings (Figure 1), where competing objectives induce conflicting gradients that collectively drive samples away from the data manifold. While exact guidance methods (Feng et al., 2025) can mitigate this issue and achieve high-fidelity alignment, they are typically computationally expensive and lack the flexibility required to efficiently adapt across diverse reward functions.

In this paper, we develop a theoretical analysis that establishes an upper bound on the approximation error of guided sampling, decomposing it into three terms: *coupling shift*, *gradient misalignment*, and *localized approximation* error. Our analysis reveals that, in compositional reward settings, **approximation error grows sharply with gradient misalignment $(1 - \cos\phi)$ and the number of reward functions $G$**, where $\phi$ denotes the average angular divergence between guidance channels (Figure 2(c); see Section 4).

Motivated by this insight, we propose **C**onflict-**A**wa**R**e

Additive Guidance (**CAR guidance**, denoted $g^{\text{car}}$), a lightweight guided sampling strategy that actively mitigates off-manifold drift by detecting and correcting gradient misalignment. CAR guidance introduces a conflict-aware gating mechanism that selectively activates learnable corrections in regions of significant gradient conflict.

We empirically evaluate $g^{\text{car}}$ across a broad range of domains, including 2D synthetic benchmarks, pixel-space image editing, and generative decision-making tasks spanning state-based planning and 3D point-cloud manipulation. Across all settings, $g^{\text{car}}$ consistently outperforms state-of-the-art baselines in challenging compositional reward settings, achieving stronger alignment with on-the-fly constraints while preserving high sample fidelity. $g^{\text{car}}$ improves identity preservation by $25.4\%$ in image editing, and increases planning success rates by $38.75\%$. On robot manipulation tasks, where naïve inference-time guidance is particularly prone to drifting off-manifold and producing out-of-distribution trajectories, $g^{\text{car}}$ cuts violation rates by $78\%$ and lifts success from $9\%$ to $61\%$, enabling reactive obstacle avoidance. Code, data, and pretrained checkpoints are available at `github.com/yuxuehui/CAR-guidance`.

**Conflict of Interest Disclosure.** The authors declare no financial conflicts of interest related to this work.

## 2. Preliminaries

### 2.1. Flow matching and conditional flow matching

Flow Matching (FM) (Lipman et al., 2023) is a simulation-free method for training continuous normalizing flows, which learns a time-dependent vector field $v_t(x_t, t)$ to transport a source distribution $p_0$ to a target distribution $p_1$. A vector field $v_t$ is said to generate a probability density path $p_t$ if its flow $\phi_t : [0, 1] : \mathbb{R}^d \to \mathbb{R}^d$ satisfies the continuity equation $\partial_t p_t + \nabla \cdot (p_t v_t) = 0$ with boundary conditions

matching the source and target distributions at $t = 0$ and $t = 1$, respectively. Given a target probability density path $p_t$ and a corresponding target vector field $v_t$, which generates $p_t$, the FM objective is:

$$\mathbb{E}_{t \sim \mathcal{U}(0,1), \, x_t \sim p_t} \left[ \| v_\theta(x_t, t) - v_t(x_t, t) \|^2 \right], \quad (1)$$

where $\theta$ denotes learnable parameters of the vector field $v_\theta$.

To make training tractable, Conditional Flow Matching (CFM) (Lipman et al., 2023) introduces a latent variable $z$ distributed according to a coupling measure $\pi(z)$, and defines conditional probability paths $p_t(x_t|z)$ together with corresponding conditional vector fields $u_t(x_t, t|z)$. The CFM objective is given by:

$$\mathbb{E}_{\substack{t \sim \mathcal{U}(0,1) \\ z \sim \pi(z)}} \mathbb{E}_{x_t \sim p_t(\cdot|z)} \left[ \| v_\theta(x_t, t) - v_t(x_t, t|z) \|^2 \right]. \quad (2)$$

For arbitrary choices of the latent variable $z$ and coupling measure $\pi(z)$, minimizing the conditional flow matching loss in (2) is equivalent to minimizing the marginal flow matching objective in (1), and yields the optimal marginal vector field $v_t(x_t, t) = \mathbb{E}_{z \sim \pi(z|x_t)}[v_t(x_t, t|z)]$ (Tong et al., 2024b). In the general formulation, the latent variable is defined as the coupling pair $z = (x_0, x_1)$, and the coupling measure $\pi(z) = \pi(x_0, x_1)$ represents the joint coupling between the source and target distributions.

### 2.2. Guided sampling

At inference time, our goal is to alter the base vector field $v_t(x_t, t)$, which generates the base target distribution $p_1(x_1)$, into a new vector field $v'_t(x_t, t)$ that generates samples from a reweighted target distribution $p'_1(x_1) \propto p_1(x_1)e^{r(x_1)}$. Here, $r(x_1) = \sum_{j=1}^{G} r_j(x_1)$ is a composition of measurable reward functions $r_j : \mathbb{R}^d \to \mathbb{R}$ to be maximized.

Inference-time guided sampling steers the generation process by injecting an additive term $g_t(x_t, t)$ into the base vector field (Pokle et al., 2024; Feng et al., 2025):

$$v'_t(x_t, t) = v_t(x_t, t) + g_t(x_t, t), \quad (3)$$

while still initializing trajectories from the fixed source distribution $p_0$, as illustrated in Figure 2 (b,c). Recall that the modified vector field $v'_t(x_t, t)$ preserves the pretrained prior; that is, the conditional vector field $v'_t(x_t, t|z) = v_t(x_t, t|z)$ and the conditional probability path $p'_t(x_t|z) = p_t(x_t|z)$ remain invariant. Consequently, guided sampling is formally equivalent to reweighting the coupling measure $\pi(z)$ over the latent coupling paths. The modified marginal probability path and vector field are given by:

$$p'_t(x_t) = \int p_t(x_t|z) \, \pi'(z) \, dz, \quad (4)$$

$$v'_t(x_t, t) = \int v_t(x_t, t|z) \, \pi'(z|x_t) \, dz, \quad (5)$$

with $\pi'(z) = \frac{1}{\mathcal{Z}} \mathcal{P}(z)\pi(z)e^{r(x_1)}$. Here, $\mathcal{P}(z) \triangleq \frac{\pi'(x_0|x_1)}{\pi(x_0|x_1)}$ is the coupling ratio, which captures the shift in the coupling measure: from a coupling perspective, constraining the target to the reweighted distribution $p'_1(x_1)$ inherently changes the optimal coupling between source and target. And the coupling shift term $\mathcal{P}(z)$ ensures that the source $p_0(x_0)$ remains unchanged under the guidance operation. $\mathcal{Z} = \iint \pi(x_0, x_1)e^{r(x_1)} \, dx_0 \, dx_1$ is the normalizing constant, and $r(x_1)$ is the reward function evaluated at the terminal state of the path $z$.

By Equations (3) and (5), Feng et al. (2025) gave a closed-form guidance term:

$$g_t(x_t, t) = \int \left( \frac{\mathcal{P}(z)e^{r(x_1)}}{\mathcal{Z}_t(x_t)} - 1 \right) v_t(x_t, t|z) \, \pi(z|x_t) \, dz. \quad (6)$$

where $\mathcal{Z}_t(x_t) = \int \mathcal{P}(z)e^{r(x_1)}\pi(z \mid x_t)dz$ is the normalizer.

Existing guided sampling methods (Pokle et al., 2024; Yu et al., 2023; Patel et al., 2025) typically assume $\mathcal{P}(z) \approx 1$ and approximate the reward function via a first-order Taylor expansion $\hat{x}_1(x_t) \triangleq \mathbb{E}_{z \sim \pi(z|x_t)}[x_1]$. Then, the approximate guidance term:

$$g_t^{\text{approx}}(x_t, t) \approx \text{Cov}_{\pi(z|x_t)} \left( v_t(\cdot \mid z), x_1 \right) \nabla_{\hat{x}_1} r(\hat{x}_1). \quad (7)$$

In practice, this covariance matrix is often further simplified to a hyperparameter or a time-dependent scalar value.

## 3. Related works

Inference-time guided sampling methods aim to estimate the additive guidance $g_t(x_t, t)$, which can be broadly categorized into approximate guidance and exact guidance.

**Approximate Guidance.** Lots of works use gradient $\nabla_{x_t} r(\hat{x}_1)$ as defined in Equation (7) as guidance, and implicitly assume the coupling ratio $\mathcal{P}(z) \approx 1$. For example, in diffusion models, it is widely adopted by DPS (Chung et al., 2023) and LGD (Song et al., 2023b). In flow models, FlowDPS (Pokle et al., 2024) applies this approximation to the OT-ODE (guiding trajectories via $\nabla_{x_t} \log p(y|\hat{x}_1)$), and FlowChef (Patel et al., 2025) derives similar guidance under assumptions of a locally linear vector field and a constant Jacobian. Both approaches converge to the form $\nabla_{x_t} r(\hat{x}_1)$, which we refer to as $g^{\text{cov-G}}$ following the notation of Feng et al. (2025). These approximate methods are simple, compute-light, and flexible. Particularly for rectified flows (Liu et al., 2023a), straight paths enable efficient terminal prediction via a single Euler step. However, these methods often result in off-manifold drift.

**Exact Guidance.** Exact guidance methods explicitly learn the ground-truth guidance, falling into two sub-categories. *Training-based* methods regress to the ground-truth guidance (Equation (6)). For example, Guidance Matching

(GM) (Feng et al., 2025) directly learns the guidance by regressing to $v_t(x_t|x_1) = (x_1 - x_t)/(1-t)$, where $x_1$ is sampled from the unnormalized target distribution $p'_1(x_1)$. Although GM achieves high fidelity, it is often impractical: ground-truth samples $x_1$ are inaccessible in tasks such as image editing, and training a guidance network per reward function is computationally expensive and introduces additional confounding errors from a learned network. *Sample-based, training-free* methods instead estimate $g_t(x_t)$ via SDE or ODE sampling (Feng et al., 2025; Holderrieth et al., 2026). The recent GLASS-FKS (Holderrieth et al., 2026), which steers GLASS flows via Feynman-Kac sampling, improves sampling efficiency over prior particle methods, but still still inherits the high variance and heavy per-sample compute cost characteristic of this family.

Positioned between these two categories, we aim to improve compute-light approximate guidance by adding a fraction of extra compute to correct the approximation error (see Appendix A for further discussion).

## 4. Approximation errors of guided sampling

In this section, we analyze approximation errors within the framework of measure transport.

Guided sampling targets a tilted terminal distribution $p_1^\star(x_1) \propto p_1(x_1)e^{r(x_1)}$. Consistent with this objective, the guided marginal density at any intermediate time $t$ can be expressed as $p_t^\star(x_t) \propto p_t(x_t)e^{V(x_t,t)}$, where the value function $V(x_t, t)$ is defined as the log-expected exponentiated future reward:

$$V(x_t, t) \triangleq \log \mathbb{E}_{x_1 \sim p(\cdot|x_t)}\left[e^{r(x_1)}\right] \qquad (8)$$

$$= \log \mathbb{E}_{z \sim \pi(\cdot|x_t)}\left[e^{R(z)}\right], \qquad (9)$$

where the second equality holds due to the deterministic mapping from latent space to data space, i.e., $x_1 = \Psi_1(z)$ (where $\Psi_t$ is the flow map at time $t$), inherent in flow models. This enables evaluating the reward directly on the latent variable $z$ via $R(z) \triangleq r(\Psi_1(z))$.

The optimal coupling measure is defined as:

$$\pi^\star(z) = \frac{1}{\mathcal{Z}}\mathcal{P}(z)e^{R(z)}\pi(z), \qquad (10)$$

where $\pi(z)$ is the base prior, $e^{R(z)}$ is the trajectory reward, and $\mathcal{P}(z)$ is the coupling shift ratio. From exact to the approximate guidance, we follow the approximation process

$$\pi^\star(z) \xrightarrow[\mathcal{P}(z)\approx 1]{\text{Coupling-Invariant}} \pi^{\text{CI}}(z) \xrightarrow[\hat{V}(x_t)]{\text{Localize}} \pi^{\text{approx}}(z), \quad (11)$$

with a fixed source $p_0$. We can then attribute the approximation error to two approximation steps: First, by defining

the shift ratio as $\mathcal{P}(z) \triangleq \frac{d\pi^\star}{d\pi^{\text{CI}}}(z)$ and assuming $\mathcal{P}(z) \equiv 1$, we have $\pi^{\text{CI}}(z) \propto e^{R(z)}\pi(z)$ which ignores the coupling shift in the optimal transport, i.e., optimal coupling is invariant. Second, the intractable value gradient $\nabla_{x_t} V(x_t)$ is locally approximated via a two-step proxy: (i) In diffusion models, a common simplification is to move the expectation inside the log-exponential via Jensen's inequality, yielding $\nabla \log \mathbb{E}_{z|x_t}[e^{R(z)}] \approx \nabla \mathbb{E}_{z|x_t}[R(z)]$. (ii) We approximate the expected reward using a first-order Taylor expansion around the mean, $\mathbb{E}[R(z)] \approx r(\hat{x}_1)$, with $\hat{x}_1 = \mathbb{E}[x_1|x_t]$. Then, we have a surrogate $\hat{V}(x_t) \triangleq \sum_{j=1}^{G} r_j(\hat{x}_1)$, which drives the guidance through the aggregated gradients $\sum_{j=1}^{G} \nabla r_j(\hat{x}_1)$.

To set the stage for our error analysis, we first introduce the gradient misalignment between rewards.

**Definition 4.1** (Gradient Misalignment). Let $g_k(x_t) \triangleq \nabla_{x_t} r_k(\hat{x}_1)$ denote the guidance contributed by the $k$-th reward, evaluated at the predicted terminal state $\hat{x}_1 = \mathbb{E}[x_1|x_t]$. Let $\phi_{jk}$ be the angle between $g_j$ and $g_k$. The gradients are defined as *misaligned* when they are not perfectly collinear, i.e., $1 - \cos\phi_{jk} > 0$.

For simplicity, we define the average pairwise cosine similarity as $\cos\phi := \frac{2}{G(G-1)}\sum_{j<k}\cos\phi_{jk}$. Broadly, gradient misalignment is quantified by $1 - \cos\phi > 0$.

**Theorem 4.2** (Upper Bound of Approximation Error). *The approximation error $\mathcal{E} \triangleq W_2^2(p_1^\star, \hat{p}_1)$ between the exact and realized target distributions admits the three-term decomposition:*

$$\mathcal{E} \lesssim \underbrace{C_{\text{CI}}\int_0^1 \mathbb{E}\left[\mathbb{E}_{\pi^{\text{CI}}}\left[(\mathcal{P}(z) - 1)^2\right]\right]dt}_{\textit{(A) coupling shift error}}$$

$$+ \underbrace{G(G-1)\mu^2\int_0^1 \mathbb{E}[1 - \cos\phi_t(x_t)]dt}_{\textit{(B) gradient misalignment error}}$$

$$+ \underbrace{G\int_0^1 \mathbb{E}\left[\left(\frac{\lambda_h \sigma_1 d}{e^{r(\hat{x}_1)}}\right)^2(C_1 + C_2)\right]dt}_{\textit{(C) localized approximation error (Feng et al., 2025)}} \quad (12)$$

*where $\mu := \|g_j^{\text{CI}}\|$ is the per-reward gradient norm, $C_{\text{CI}}$ only depends on base velocity field, $\lambda_h$ is the spectral norm of the Hessian of $e^r$, $G$ is the number of rewards, $(1-\cos\phi_t)$ is the gradient misalignment, $\sigma_1(t)$ measures posterior uncertainty, and $C_1, C_2$ are constants depending on the base velocity field. We provide an intuitive interpretation in Appendix B and a more detailed proof in Appendix C.*

Theorem 4.2 provides an error bound and offers a theoretical roadmap, identifying the key design space for optimizing guided sampling:

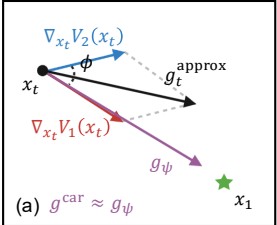
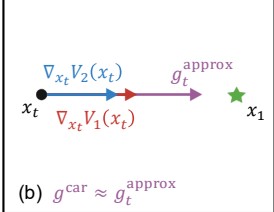

*Figure 3.* (a) When gradients are misaligned (large $\phi$), the approximate guidance $g^{\text{approx}}$ ($\rightarrow$), the vector sum of $\rightarrow$ and $\rightarrow$, points off-manifold; the conflict-aware weight $w_t \approx 1$, so $g^{\text{car}} \approx g_\psi(x_t, t)$ ($\rightarrow$) corrects the trajectory toward the true target $x_1$ ($\star$). (b) When gradients align ($\phi \approx 0$), $g^{\text{approx}}$ is already accurate; $w_t \approx 0$, so $g^{\text{car}} \approx g^{\text{approx}}$ ($\rightarrow$).

1. the error scales with the number of reward functions $G$ and gradient misalignment $(1 - \cos\phi)$;
2. the error is small when the coupling shift is negligible ($\mathcal{P}(z) \approx 1$), which is reasonable in many practical flow matching methods with dependent couplings, such as mini-batch OT-FM (Tong et al., 2024a), but can be problematic in OT-FM (Onken et al., 2021);
3. the error is small when the reward landscape is smooth, i.e., small $\lambda_h = \|\nabla^2 e^r\|_2$;
4. the error is small when the predicted endpoint $\hat{x}_1 = \mathbb{E}[x_1|x_t]$ lies in a high-reward region (large $r(\hat{x}_1)$);
5. the error decreases as $\sigma_1$ shrinks (e.g., $t \to 1$).

# 5. Conflict-aware additive guidance

We improve the compute-light approximate guidance by adding extra compute to correct the approximation error:

**Definition 5.1** (Conflict-Aware Additive Guidance). Let $\mathcal{M} = (v_t, p_0, p_1)$ be a base flow model. Given a target distribution $p'_1(x_1) \propto p_1(x_1)e^{r(x_1)}$ with rewards $r(x_1) = \sum_{j=1}^{G} r_j(x_1)$, the conflict-aware additive guidance ($g^{\text{car}}$) transforms the base velocity $v_t$ into a guided velocity $v'_t$ via:

$$v'_t(x_t, t) = v_t^{\text{base}}(x_t, t) + g^{\text{car}}(x_t, t), \quad (13)$$

$$g^{\text{car}}(x_t, t) = (1 - w_t)\, g^{\text{approx}} + w_t\, g_\psi(x_t, t), \quad (14)$$

where the guidance is a composition of a fast approximation $g^{\text{approx}}$ and a learnable guidance $g_\psi$, which is controlled by a conflict-aware weight

$$w_{\text{raw}}(x_t) = 1 - \frac{2}{G(G-1)} \sum_{j<k} \frac{\langle g_j, g_k \rangle}{\|g_j\|\|g_k\| + \varepsilon}, \quad (15)$$

The raw conflict score $w_{\text{raw}}(x_t) \in [0, 2]$ is then mapped to $w(x_t) \in (0, 1)$. The $\varepsilon$ ensures numerical stability. Finally, $g_\psi$ is trained to approximate the exact guidance by data (see Section 5.1 for details).

As illustrated in Figure 3, in regions of gradient misalignment, our $g^{\text{car}}$ leverages a learned lightweight guidance $g_\psi$ to

correct the systematic errors inherent in approximate methods, thereby effectively rectifying off-manifold drift. This design positions $g^{\text{car}}$ effectively between approximate and exact approaches, delivering near-exact performance with significantly lighter compute compared to exact guidance.

## 5.1. Value gradient

To bypass the intractability of the closed-form target in Eq. (6), we seek to learn the exact guidance $g_\psi$ directly from data. By viewing the flow generation process as deterministic ODE dynamics, we evaluate $V(x_t, t)$ as the cumulative return of a trajectory induced by the current policy $v'_t$ defined in Eq. (13). Mathematically, $V(x_t, t)$ serves as the fixed point of the Bellman backup operator $\mathcal{T}^{v'}$. Details can be found in Appendix D.

**Proposition 5.2** (Fitted Value Evaluation). *Let $\mathcal{F}$ denote the function class (e.g., neural networks) used to approximate the value function. We collect a dataset of transitions $\mathcal{D} = \{(x, t, r, x', t')\}$ generated under the current guided dynamics, where $t' = t + \Delta t$ and $r$ is the reward. The value function can be estimated empirically by minimizing a least-squares Bellman residual:*

$$\hat{V}_{k+1} = \arg\min_{V \in \mathcal{F}} \mathbb{E}_{\mathcal{D}}\left[\left(r + \gamma \hat{V}_k(x', t') - V(x, t)\right)^2\right]. \quad (16)$$

*subject to the boundary condition $\hat{V}_k(x, 1) \equiv r(x)$ for terminal states. Here, $\gamma \in (0, 1]$ is the discount factor, and $\hat{V}_k$ is the target from the previous iteration [1]. Equation (16) empirically approximates the Bellman backup operator $\mathcal{T}^{v'}$ using finite data and a function class $\mathcal{F}$. Upon convergence, the learnable guidance is the gradient of the estimated value:*

$$g(x_t, t) \triangleq \nabla_x \hat{V}(x_t, t), \quad (17)$$

*which is a Markovian surrogate for the exact guidance.*

Fitted Value Evaluation can diverge due to the deadly triad, the instability arising from the interplay of function approximation, off-policy data, and bootstrapping (Proposition D.2, Appendix D). Leveraging the deterministic dynamics and terminal-only rewards of flow matching, we propose Terminal Value Regression (TVR). By regressing directly against the terminal reward $r(x_1)$, TVR eliminates the need for bootstrapping, effectively breaking the deadly triad and ensuring stable convergence.

**Proposition 5.3** (Terminal Value Regression). *Let $\mathcal{F}$ denote a function class used to approximate the value function. We collect a dataset of terminal rollouts $\mathcal{D} = \{(x_t, t, x_1)\}$, where $x_1$ is the terminal state reached from $x_t$ by integrating the current guided dynamics. The value function is estimated by minimizing the following regression objective:*

$$\hat{V} = \arg\min_{V \in \mathcal{F}} \mathbb{E}_{(x_t, t, x_1) \sim \mathcal{D}}\left[\left(r(x_1) - V(x_t, t)\right)^2\right]. \quad (18)$$

---

[1]For pure terminal optimization, $r = 0$ and $\gamma = 1$.

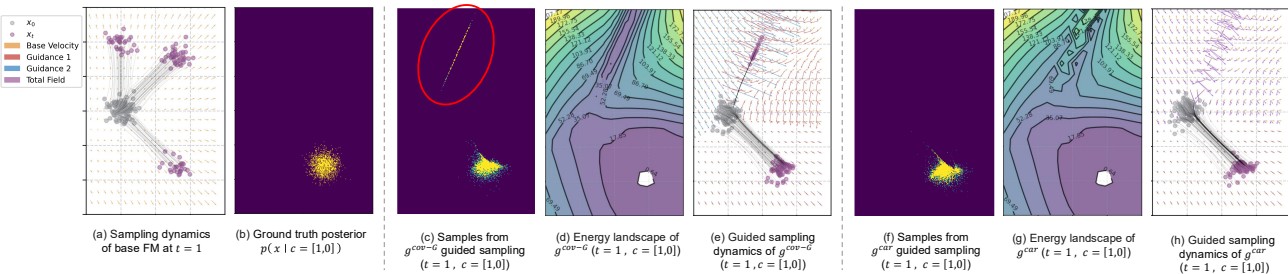

*Figure 4.* Visualization results on the synthetic dataset under $[1, 0]$ constraints. (c–e) $g^{\text{cov-G}}$ shows significant off-manifold drift due to an "energy trap" (highlighted by the red circle), where conflicting gradients lead to erratic sampling trajectories. (f–h) Our $g^{\text{car}}$ restores the accurate reward landscape, thereby rectifying the off-manifold drift. An intuitive interpretation of the "energy trap" caused by gradient misalignment can be found in Appendix B.

*Unlike the bootstrapped target in Eq.* (16)*, the terminal reward* $r(x_1)$ *serves as a stable, unbiased regression target, which is enabled by the deterministic nature of the flow.*

We parameterize the scalar value function $V(x_t, t)$, then define the guidance as $g_\psi(x_t, t) \triangleq \nabla_{x_t} V_\psi(x_t, t)$. Building on Proposition 5.3, we optimize $\psi$ by minimizing a masked regression loss against the terminal reward:

$$\mathcal{L}(\psi) = \mathbb{E}_{(x_t, t, x_1) \sim \mathcal{D}} \left[ \mathbb{I}_t \cdot \left( r(x_1) - V_\psi(x_t, t) \right)^2 \right], \quad (19)$$

where $x_1$ is the terminal state obtained from online roll-outs. We allocate compute budget efficiently by using $\mathbb{I}_t \triangleq \mathbb{1}(w(x_t) > \tau)$, which ensures the guidance $g_\psi$ is updated solely in regions showing high gradient conflict. While directly parameterizing the vector field $\nabla V(x_t, t)$ is common in diffusion models (Song & Kingma, 2021), unconstrained neural vector fields are not guaranteed to be conservative (i.e., curl-free) (Balcerak et al., 2025). See Appendix E.1 for further discussion.

## 6. Experiments

Our experiments are designed to answer two core questions: (1) Can $g^{\text{car}}$ effectively rectify off-manifold drift under compositional reward settings? (2) Is the $g^{\text{car}}$ compute light?

### 6.1. Experimental setup

**Baselines.** Our baselines span the full spectrum of inference-time guidance methods, letting us probe whether each exhibits off-manifold drift: (1) approximate guidance ($g^{\text{cov-G}}$), which is compute-light but susceptible to off-manifold drift; and (2) exact guidance, including the sample-based **GLASS-FKS** (Holderrieth et al., 2026) and the training-based **GM** (Feng et al., 2025). Since GM requires ground-truth samples $x_1$ that satisfy all constraints, its use is limited to synthetic datasets where such samples are accessible by construction. Task-specific SOTA baselines are also included, detailed in the following sections. Our $g^{\text{car}}$ corrects

*Table 1.* Comparison with baselines on the Synthetic dataset. Each number is evaluated over 10k generated samples. Here $[\cdot, \cdot]$ denotes the target labels of two classifiers, where $[1, 0]$ represents the gradient conflict scenario. For GLASS-FKS, we report $K=16$; For $g^{\text{car}}$, $\tau = 0.50$. Full results are in Appendix E.4.3. **Bold text** is the best performance.

| Method | Posterior Coverage (PC) (%) ↑ | | | | Constraint Satisfaction (CS) (%) ↑ | | | |
|---|---|---|---|---|---|---|---|---|
| | $[0, 0]$ | $[1, 0]$ | $[1, 1]$ | Avg. | $[0, 0]$ | $[1, 0]$ | $[1, 1]$ | Avg. |
| $g^{\text{cov-G}}$ | 94.00 | 71.70 | 89.00 | 84.90 | **100.00** | 89.56 | 99.87 | 96.48 |
| PCGrad | 94.05 | 75.20 | **89.15** | 86.13 | **100.00** | 92.45 | 99.88 | 97.44 |
| GM | 82.30 | 84.50 | 86.20 | 84.33 | 99.93 | 99.99 | 99.92 | 99.95 |
| GLASS-FKS | 88.80 | 90.80 | 88.60 | 89.40 | **100.00** | 99.71 | **100.00** | 99.90 |
| $g^{\text{car}}$ (ours) | **94.60** | **93.80** | 88.70 | **92.37** | **100.00** | **100.00** | 99.87 | **99.96** |

| Method | Inference Time (ms / sample) ↓ | | | | Data Usage ($\times 10^3$) ↓ | | | |
|---|---|---|---|---|---|---|---|---|
| | $[0, 0]$ | $[1, 0]$ | $[1, 1]$ | Avg. | $[0, 0]$ | $[1, 0]$ | $[1, 1]$ | Avg. |
| $g^{\text{cov-G}}$ | **0.36** | **0.38** | 0.36 | **0.37** | – | – | – | – |
| PCGrad | 0.37 | 0.42 | 0.38 | 0.39 | – | – | – | – |
| GM | 2.81 | 2.81 | 3.11 | 2.91 | 10240 | 10240 | 10240 | 10240 |
| GLASS-FKS | 298 | 296 | 302 | 299 | – | – | – | – |
| $g^{\text{car}}$ (ours) | 0.46 | 4.20 | **0.27** | 1.65 | 74.75 | 1573.89 | **0** | 549.55 |

off-manifold drift by adding only a fraction of extra compute. We include **PCGrad** (Yu et al., 2020), a gradient conflict resolution method from multi-objective optimisation, to show that our conflict-aware mechanism rectifies off-manifold drift more effectively than projection-based deconfliction. See details and additional results in Appendix E.

### 6.2. Synthetic dataset

**Tasks.** We begin with a 2-dimensional Mixture of Gaussians (see Appendix E.4 for details), where the ground-truth density $p_t$ is known, allowing for precise quantitative evaluation. We train a flow matching model (Lipman et al., 2024) to transport a standard Gaussian source to a Mixture of Gaussians target. We steer the generation using two pre-trained classifiers that impose differing label constraints on the target samples. These constraints are specifically configured to induce severe gradient conflicts, thereby creating a "stress test" to benchmark the robustness of different guidance methods against off-manifold drift.

*Table 2.* Comparison with baselines on Maze2D. Compositional reward settings: (1) static obstacle: two random static obstacles; (2) static goal: two random goals; (3) dynamic obstacle: two random dynamic agents; (4) hybrid composition: one static and one dynamic obstacles. Metrics include Inference Time (ms/sample), Safe (collision-free rate %), Violation (mean constraint violations #), Success (success rate %), and Steps (#). Results are averaged over 100 samples with conflict threshold $\tau = 0.20$. Note that we do not use inpainting, which allows us to better observe the capability of inference-time alignment methods in preserving the base model prior. Furthermore, comparing the success rates reveals that directly applying GLASS-FKS, MPPI, $g^{\text{cov-G}}$, and their PCGrad or $g^{\text{car}}$ corrections as inference-time techniques to the base flow to satisfy constraints compromises prior preservation. The $g^{\text{car}}$ method requires an online training period of $10.2 \pm 0.1$ min (mean $\pm$ std across 5 random seeds) for 4 training steps prior to being used for inference.

| Method | Time ↓ | (1) static obstacles | | | | (2) static goal | | | | (3) dynamic obstacles | | | | (4) hybrid composition | | | |
| --- | --- | --- | --- | --- | --- | --- | --- | --- | --- | --- | --- | --- | --- | --- | --- | --- | --- |
| | | Safety ↑ | Viol. ↓ | Succ. ↑ | Steps ↓ | Safety ↑ | Viol. ↓ | Succ. ↑ | Steps ↓ | Safety ↑ | Viol. ↓ | Succ. ↑ | Steps ↓ | Safety ↑ | Viol. ↓ | Succ. ↑ | Steps ↓ |
| GLASS-FKS | 532.7 ±53.2 | 78 ±3.2 | 0.3 ±0.1 | **100** ±0.0 | - | 56 ±4.5 | 0.6 ±0.2 | **100** ±0.0 | - | 71 ±3.8 | 0.4 ±0.1 | **100** ±0.0 | - | 68 ±4.1 | 0.5 ±0.2 | **100** ±0.0 | - |
| MPPI | 242.4 ±12.5 | **100** ±0.0 | **0.0** ±0.0 | 41 ±2.4 | 30 | 62 ±3.5 | 1.5 ±0.3 | 69 ±3.1 | 30 | 58 ±2.8 | 0.5 ±0.2 | 63 ±2.9 | 30 | 47 ±3.1 | 0.4 ±0.2 | 69 ±3.4 | 30 |
| MPPI + $g^{\text{car}}$ (ours) | 265.8 ±14.2 | **100** ±0.0 | **0.0** ±0.0 | $98^{\uparrow 57}$ ±1.2 | 30 | $100^{\uparrow 38}$ ±0.0 | $0.0^{\downarrow 1.5}$ ±0.0 | $96^{\uparrow 27}$ ±1.5 | 30 | $96^{\uparrow 38}$ ±1.4 | $0.1^{\downarrow 0.4}$ ±0.0 | $94^{\uparrow 31}$ ±1.8 | 30 | $95^{\uparrow 48}$ ±1.6 | $0.2^{\downarrow 0.2}$ ±0.1 | $92^{\uparrow 23}$ ±2.1 | 30 |
| $g^{\text{cov-G}}$ | 150.0 ±5.4 | 39 ±2.1 | 0.3 ±0.1 | 16 ±1.4 | - | 41 ±2.4 | 1.7 ±0.3 | 23 ±1.8 | - | 39 ±2.2 | 0.9 ±0.2 | 42 ±2.5 | - | 34 ±1.9 | 1.1 ±0.2 | 12 ±1.1 | - |
| PCGrad | 175.2 ±18.0 | $44^{\uparrow 5}$ ±2.3 | $0.2^{\downarrow 0.1}$ ±0.1 | $27^{\uparrow 11}$ ±1.7 | - | $47^{\uparrow 6}$ ±2.6 | $1.5^{\downarrow 0.2}$ ±0.3 | $32^{\uparrow 9}$ ±2.1 | - | $41^{\uparrow 2}$ ±2.1 | $0.7^{\downarrow 0.2}$ ±0.1 | $46^{\uparrow 4}$ ±2.8 | - | $35^{\uparrow 1}$ ±1.8 | $0.9^{\downarrow 0.2}$ ±0.2 | $18^{\uparrow 6}$ ±1.5 | - |
| $g^{\text{car}}$ (ours) | 168.4 ±8.6 | $74^{\uparrow 35}$ ±1.5 | $\mathbf{0.0}^{\downarrow 0.3}$ ±0.0 | $79^{\uparrow 63}$ ±1.8 | 4 | $63^{\uparrow 22}$ ±2.2 | $0.8^{\downarrow 0.9}$ ±0.1 | $72^{\uparrow 49}$ ±1.9 | 4 | $49^{\uparrow 10}$ ±1.7 | $0.2^{\downarrow 0.7}$ ±0.1 | $61^{\uparrow 19}$ ±2.0 | 4 | $43^{\uparrow 9}$ ±1.8 | $0.3^{\downarrow 0.8}$ ±0.1 | $36^{\uparrow 24}$ ±1.6 | 4 |

*Note:* **Bold text** indicates the best performance. Rows with  gray backgrounds  indicate methods that use our $g^{\text{car}}$ for conflict correction. Purple superscripts show the performance change of $g^{\text{car}}$ over $g^{\text{cov-G}}$, and teal superscripts show the change of PCGrad over $g^{\text{cov-G}}$, where ↑ denotes improvement and ↓ denotes degradation. For all metrics, we report the mean (top row) and standard deviation across 5 random seeds (bottom row).

**Evaluation metrics.** We present intuitive visualizations, and also designed four quantitative metrics: (i) posterior coverage (higher is better on data manifold); (ii) inference time (lower is better speed); (iii) data usage (lower indicates better training data efficiency); (iv) constraint satisfaction (higher indicates better recovery of ground-truth posterior).

**Results.** We first visualize the failure modes of the standard approximation guidance $g^{\text{cov-G}}$ in Figure 4. Notably, $g^{\text{cov-G}}$ generates off-manifold samples (highlighted by the red circle in Figure 4c). By analyzing the underlying energy landscape (Figure 4d), we observe an "energy trap", a spurious artifact arising from gradient conflict. This trap captures the sampling process (Figure 4e), forcing trajectories to ultimately diverge from the data manifold. Our $g^{\text{car}}$ rectifies the vanishing energy by learning a residual guidance within these conflict regions, thereby eliminating off-manifold drift.

Quantitative results in Table 1 show, under the conflict constraint $[1, 0]$, $g^{\text{cov-G}}$ drifts off-manifold for nearly 30% of samples (only 71.70% PC), and PCGrad provides only marginal correction (75.20% PC, ↑3.5), confirming that generic projection-based deconfliction is insufficient at inference time. Our $g^{\text{car}}$ reduces off-manifold drift to 6.2% (93.80% PC), incurring only a small compute overhead over $g^{\text{cov-G}}$ (1.65 vs. 0.37 ms/sample). The exact baselines come with their own trade-offs: GM requires roughly $20\times$ more training data than $g^{\text{car}}$ and shows higher variance, while

GLASS-FKS avoids off-manifold drift but suffers from high computational cost and is sensitive to the particle count.

### 6.3. Generative decision-making as planners

**Tasks.** We conduct experiments on generative decision making tasks where generative models have been used as planners. We focus on the Maze2D (Luo et al., 2024), involving a point-robot with state $s \in \mathbb{R}^4$ (position and velocity) and action $a \in \mathbb{R}^2$ (force). We aim to steer a base planner pre-trained on expert demonstrations, where the maze layout, start and goal are randomly generated. Following Luo et al. (2024), we collect a diverse set of collision-free demonstrations via BIT* to train the base CFM model (Tong et al., 2024a). The model serves as a learned prior, generating trajectories $x = (s_{0:H-1}, a_{0:H-1}) \in \mathbb{R}^{6H}$ conditioned on the maze layout, start, and goal. The $r(x_1)$ includes: (i) static obstacle avoidance for unseen environments; (ii) goal reaching (e.g., object grasping); (iii) dynamic collision avoidance against agents with random linear trajectories (Römer et al., 2024; Bouvier et al., 2025).

**Baselines and metrics.** Besides the baselines introduced in Section 6.1, we include MPPI (Williams et al., 2017), a classical optimization-based planner effective for constrained trajectory problems, and its $g^{\text{car}}$-augmented variant MPPI+$g^{\text{car}}$. We report five metrics: (i) safety rate: the percentage of trajectories that are collision-free with respect to the static maze layout, which reflects the prior preser-

*Table 3.* Comparison on the ManiSkill2 StackCube task. (1) static obstacles: two random static obstacles; (2) hybrid composition: two random static obstacles and trajectory smoothness. Full results, including PickCube, are in Table 8.

| Method | Time ↓ | (1) static obstacles | | (2) hybrid composition | |
|---|---|---|---|---|---|
| | | Violation ↓ | Success ↑ | Violation ↓ | Success ↑ |
| GLASS-FKS | 493.7 | 0.6 | 32 | 1.1 | 24 |
| $g^{\text{cov-G}}$ | 185.0 | 1.2 | 12 | 1.8 | 9 |
| PCGrad | 201.4 | $1.5^{\downarrow 0.3}$ | $0^{\downarrow 12}$ | $2.2^{\downarrow 0.4}$ | $0^{\downarrow 9}$ |
| $g^{\text{car}}$ (ours) | 203.2 | $\mathbf{0.1}^{\uparrow 1.1}$ | $\mathbf{72}^{\uparrow 60}$ | $\mathbf{0.4}^{\uparrow 1.4}$ | $\mathbf{61}^{\uparrow 52}$ |

*Note:* **Bold text** indicates the best performance. Rows with gray backgrounds indicate methods that utilize our $g^{\text{car}}$ for conflict correction. Purple superscripts show the performance change of $g^{\text{car}}$ over $g^{\text{cov-G}}$, and teal superscripts show the change of PCGrad over $g^{\text{cov-G}}$, where ↑ denotes improvement and ↓ denotes degradation.

vation. (ii) violations: average number of inference-time constraint violations (e.g., collisions with new obstacles or missed goals) per trajectory. (iii) success rate: the percentage of trajectories that successfully reach the task goals. (iv) steps: the number of iterations required to train the learnable guidance (for $g^{\text{car}}$) or optimize the action sequence via importance sampling (for MPPI). (v) time: wall-clock training and inference time per sample.

**Results.** Under the four compositional reward settings in Table 2, $g^{\text{cov-G}}$ generates hallucinated paths, trajectories that jump across obstacles or have sharp kinks (see Figure 16 for visualisations). Our $g^{\text{car}}$ rectifies these failures, with average gains of 19.0% safety, 38.75% success, and 0.68 fewer violations per trajectory, at only 4 training iterations; PCGrad provides only marginal correction. GLASS-FKS, while strong overall, reaches 100% success at the cost of non-zero violations and $\sim 3.2\times$ slower inference (533 vs. 168 ms). MPPI alone is a strong planning method, collision-free on static obstacles (100 safety, 0.0 violations), but its success drops to 63–69% in dynamic and hybrid settings. MPPI+$g^{\text{car}}$ outperforms vanilla MPPI and achieves the best performance, lifting average safety to 97.75% and success to 95%, including the previously hard static-goal setting (100 safety, 96 success).

We further evaluate robustness by increasing the number of clustered static obstacles from 2 to 6 (Figure 15, Appendix E.5): $g^{\text{cov-G}}$'s success rate collapses to 12%, while our method maintains 34% in the most complex setting.

### 6.4. Generative Decision-Making as Policies

**Tasks.** We evaluate our method on high-dimensional manipulation tasks (PickCube and StackCube) in ManiSkill2 (Gu et al., 2023), where the policy predicts action chunks of horizon $T$ from 3D point cloud observations. We aim to test the capability to reactively steer a general-purpose base policy to satisfy on-the-fly requirements. The action is defined as $\mathbf{a}_t = \left[\Delta\mathbf{p}, \Delta\mathbf{r}, g\right] \in \mathbb{R}^7$, representing delta translation,

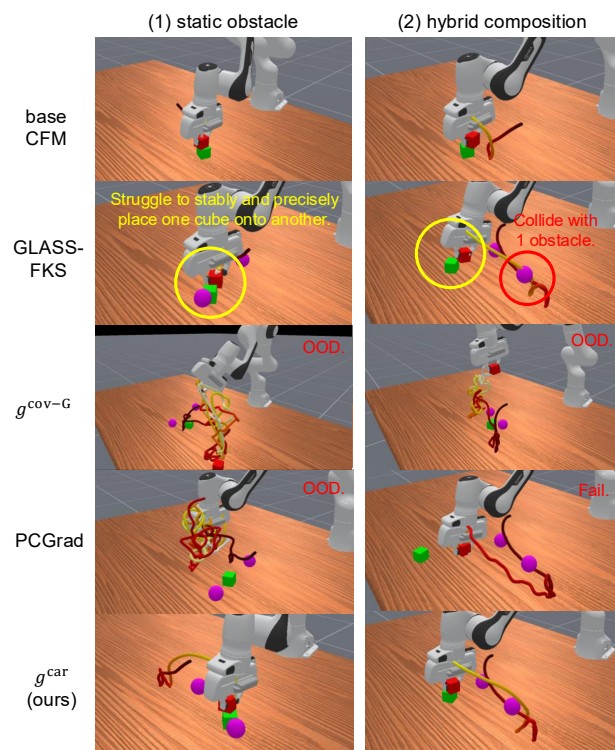

*Figure 5.* Visualization on ManiSkill2 StackCube with $\tau = 0.20$. OOD: the trajectory leaves the data manifold, producing physically incoherent motions (e.g., erratic spinning or tangled paths); Fail: the trajectory stays on the manifold but fails the task (e.g., does not reach the goal).

rotation, and gripper state. We train a base CFM model solely on unconstrained demonstrations. Reward functions $r(x)$ include: (i) static obstacles; (ii) trajectory smoothness costs.

**Evaluation metrics.** We evaluate violations, success rate, training steps and time, following the definitions in Section 6.3.

**Results.** The base CFM model achieves a 100% success rate on both training and test sets (Appendix E.6). Naïve inference-time guidance tends to drift off-manifold and produce out-of-distribution behavior: $g^{\text{cov-G}}$ generates hallucinated paths and fails to complete the tasks, and PCGrad even drives success to 0% on both StackCube settings; only $g^{\text{car}}$ rectifies these failures (Figures 5 and 19). Quantitatively (Table 3), under hybrid constraints, $g^{\text{car}}$ reduces the baseline's violation rate from 1.8 to 0.4 and boosts the success rate from 9% to 61%, with only $\sim 10\%$ inference overhead over $g^{\text{cov-G}}$ (203 vs. 185 ms). GLASS-FKS struggles on StackCube (24–32%), which requires stably and precisely placing one cube onto another, due to its high sampling variance.

*Table 4.* Comparison with baselines on CelebA-HQ. We evaluate composed prompts including *sad + angry*, *sad + happy*, and *sad + curly hair*. We report results with $\tau = 0.20$. Full results are in Table 10.

| Method | LPIPS ↓ | ID ↑ | CLIP ↑ | BLIP-ITM ↑ | VQAScore ↑ |
|---|---|---|---|---|---|
| FlowGrad | **0.203** | 0.677 | 0.274 | 0.326 | 0.596 |
| GLASS-FKS | 0.313 | 0.329 | 0.253 | 0.082 | 0.311 |
| $g^{\text{cov-G}}$ | 0.217 | 0.543 | 0.290 | 0.525 | 0.751 |
| PCGrad | $0.219^{\downarrow 0.002}$ | $0.602^{\uparrow 0.059}$ | $0.276^{\downarrow 0.014}$ | $0.444^{\downarrow 0.081}$ | $0.578^{\downarrow 0.173}$ |
| $g^{\text{car}}$ (ours) | $0.226^{\downarrow 0.009}$ | $\mathbf{0.681}^{\uparrow 0.138}$ | $\mathbf{0.291}^{\uparrow 0.001}$ | $\mathbf{0.597}^{\uparrow 0.072}$ | $\mathbf{0.755}^{\uparrow 0.004}$ |

*Note:* **Bold text** indicates the best performance. Rows with gray backgrounds indicate methods that use our $g^{\text{car}}$ for conflict correction. Purple superscripts show the performance change of $g^{\text{car}}$ over $g^{\text{cov-G}}$, and teal superscripts show the change of PCGrad over $g^{\text{cov-G}}$, where ↑ denotes improvement and ↓ denotes degradation.

angry + sad

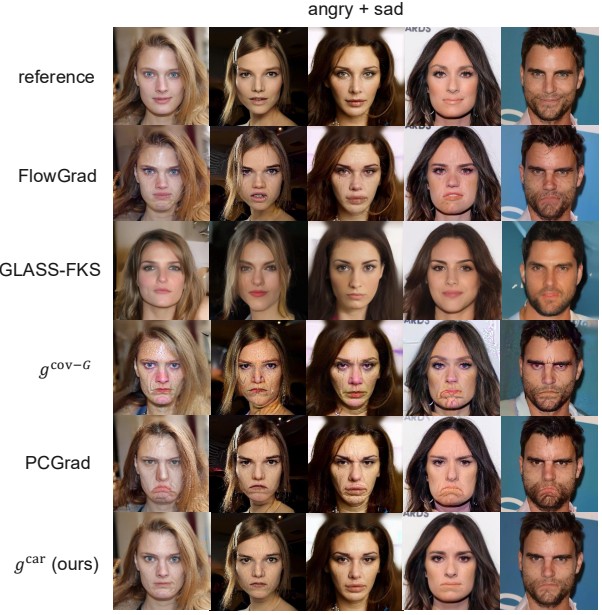

reference

FlowGrad

GLASS-FKS

$g^{\text{cov-G}}$

PCGrad

$g^{\text{car}}$ (ours)

*Figure 6.* Visualization of text-guided generated faces.

### 6.5. Text-guided Image Manipulation

**Tasks.** To evaluate the scalability of $g^{\text{car}}$ in high-dimensional pixel spaces, we conduct text-guided image manipulation on CelebA-HQ. Following Liu et al. (2023b), we use a pre-trained Rectified Flow model as the generative prior and steer it toward compositional text guidance {*sad + angry*, *sad + happy*, *sad + curly hair*}, which target different facial expressions or traits. Following the same setup as Liu et al. (2023b), given an image $x_1$, the reward for alignment with the text prompt is evaluated by the CLIP model (Radford et al., 2021), which is used to score the similarity between arbitrary image-text pairs. Experiment details are in Appendix E.7.

**Baselines and metrics.** Besides the baselines from Section 6.1, we additionally compare against the SOTA image editing method FlowGrad (Liu et al., 2023b). We report

metrics in three groups: (i) *Text-image alignment* (higher is better): CLIP (Radford et al., 2021), plus two stronger perceptual measures, BLIP-ITM (Li et al., 2022) (a strict binary image-text matcher) and VQAScore (Lin et al., 2024) (compositional reasoning via LLaVA-1.5); (ii) *Image quality*: LPIPS (lower is better preservation of the reference), ID similarity (higher is better identity preservation), and CLIP-IQA (Wang et al., 2023) (higher is fewer visual artifacts); (iii) *Efficiency*: training time and per-sample inference time.

**Results.** Visual comparisons (Figure 6) and quantitative metrics (Table 4) reveal that $g^{\text{cov-G}}$ is prone to off-manifold drift and produces hallucinated generations, with CLIP-IQA score only 0.535. PCGrad cannot recover from off-manifold drift and struggles to balance multiple constraints, leaving some targets unfulfilled (e.g., failing to generate an "angry" expression, with BLIP-ITM $P_1=0.650$ vs. $P_2=0.337$; see Table 10). FlowGrad similarly lacks the ability to balance multiple constraints. GLASS-FKS fails to preserve the reference image due to its high sampling variance, with the worst LPIPS (0.313) and ID (0.329). Our $g^{\text{car}}$ finds a sweet spot between text-image alignment and image quality, achieving the highest ID (0.681), CLIP-IQA (0.543), and alignment scores (CLIP 0.291, BLIP-ITM 0.597, VQAScore 0.755); it effectively rectifies the off-manifold drift observed in $g^{\text{cov-G}}$ and generates faces with minimal distortion.

## 7. Conclusions, Limitations, and Future Work

In this paper, we proposed Conflict-Aware Additive Guidance ($g^{\text{car}}$), a lightweight guided sampling method that incorporates a conflict-aware gating mechanism to actively detect and rectify trajectory deviations. Experimental results showed that flow models equipped with $g^{\text{car}}$ achieved state-of-the-art steerability at inference time across diverse domains, including text-guided image editing, robotic planning, and manipulation.

A remaining challenge lies in convergence under complex reward landscapes. In high-dimensional tasks like text-guided image editing, the CLIP reward signal is non-smooth, producing adversarial artifacts that maximize the score without semantic improvement, making $g_\psi$ hard to train stably. Lighter alternatives to $g_\psi$, richer reward compositions, and more structured latent representations (Yu et al., 2024; Dunion et al., 2023) replacing the CLIP reward to yield smoother landscapes are all promising directions for future work.

## Acknowledgment

This research is supported by the RIE2025 Industry Alignment Fund – Industry Collaboration Projects (IAF-ICP) (Grant No. I2501E0041), administered by A*STAR, as well as supported by Schaeffler (Singapore) PTE. LTD. and NTU Singapore through Schaeffler-NTU Corporate Lab: Intelligent Mechatronics Hub.

## Impact Statement

This work improves the reliability of inference-time guidance for generative models under multiple, potentially conflicting objectives. By identifying gradient misalignment as a key source of off-manifold drift and proposing a lightweight correction mechanism, our method enables more faithful and stable generation without retraining large pretrained models. These advances support safer and more robust deployment of generative models in applications such as planning, control, and interactive content generation, where adherence to heterogeneous constraints is critical.

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

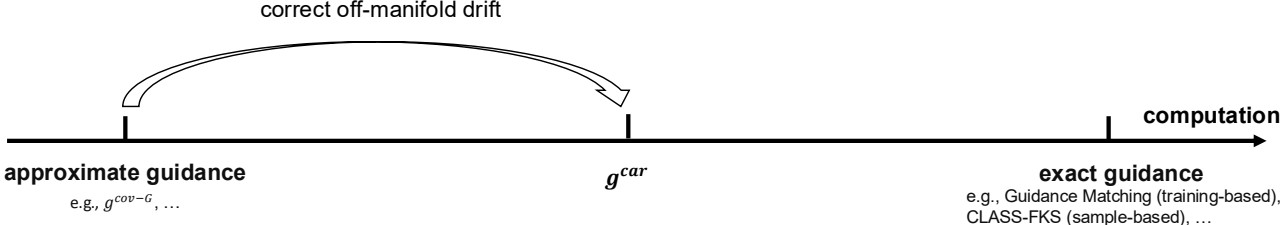

*Figure 7.* Inference-time guidance methods arranged by computational cost. Approximate guidance methods (e.g., $g^{\text{cov-G}}$) are lightweight but accumulate local approximation error, leading to off-manifold drift. Exact guidance methods (e.g., Guidance Matching, sample-based guidance) are exact but require substantially more compute. This work ($g^{\text{car}}$), which sits in the middle, aims to improve the compute-light approximate guidance by adding extra compute.

## A. Extended related works

### A.1. Inference-time alignment

Inference-time alignment methods for flow matching models refer to steering the generated samples toward some constraints, e.g., sampling from a distribution weighted with some objective function (Lu et al., 2023) or conditioned on class labels (Song et al., 2021), and all constraints can be framed as reward functions $r : \mathbb{R}^d \to \mathbb{R}$. Approaches to inference-time reward alignment for flow models can be divided into three broad paradigms:

**Inference-time guidance.** Inference-time guidance addresses the reward-tilted sampling problem by adding a guidance vector field $g_t(x_t)$ to the pretrained velocity $v_\theta(x_t, t)$ during ODE integration, leaving the pretrained model unchanged. A unified theoretical framework for this family was established by Feng et al. (2025), who derive the exact guidance vector field for general flow matching:

$$g_t(x_t) = \mathbb{E}_{z \sim p(z|x_t)}\left[\left(\frac{e^{r(x_1)}}{Z_t(x_t)} - 1\right) v_{t|z}(x_t|z)\right],$$

where $Z_t(x_t) = \mathbb{E}_{z \sim p(z|x_t)}[e^{r(x_1)}]$ is an intractable normalising constant. Methods in this family differ in how they approximate $g_t$.

*Approximate guidance* replaces the intractable posterior average with a point estimate via Tweedie's formula, adapting well-studied diffusion guidance methods including DPS (Chung et al., 2023), ΠGDM (Song et al., 2023a), and LGD (Song et al., 2023b) to the flow matching setting; Feng et al. (2025) unify these under the flow-matching extension $g^{\text{cov-G}}$. These methods are computationally lightweight but incur an approximation error, as we show in Section 4; see also Feng et al. (2025).

*Exact guidance* methods avoid this bias at the cost of additional computation. On the training-free side, Monte Carlo guidance ($g^{\text{MC}}$, Feng et al., 2025) estimates $g_t$ by drawing $N$ samples from the prior $p(z)$ and self-normalising; it is asymptotically exact but suffers from high variance, especially in high-dimensional spaces. GLASS-FKS (Holderrieth et al., 2026), which steers GLASS flows via Feynman-Kac sampling, improves sampling efficiency but still inherits the high variance. On the training-based side, Guidance Matching (Feng et al., 2025) learns a network $g_\psi$ to directly approximate $g_t$ via tractable surrogate losses, which however require ground-truth samples sastify all constraints.

**Optimization-based controlled generation.** A second paradigm frames controlled generation as a direct optimization problem: given a differentiable objective (cost or reward), one searches for an initial noise, latent trajectory, or auxiliary variable that, after running the generative ODE/SDE, produces a sample of high reward. Representative methods *differentiate through* the entire sampling ODE to back-propagate reward gradients into the input space, including D-Flow (Ben-Hamu et al., 2024), FlowGrad (Liu et al., 2023b), and source-guided flow matching (Wang et al., 2025). These methods pursue a fundamentally different objective from the guidance framework we adopt: rather than sampling from the reward-tilted distribution $p_1'(x_1) \propto p_1^{\text{base}}(x_1) e^{r(x_1)}$, they solve an optimization problem.

**Reward fine-tuning.** A third paradigm modifies the pretrained model weights to maximize the reward, based on GRPO (Liu et al., 2025a), stochastic optimal control (Domingo-Enrich et al., 2025), DPO (Wallace et al., 2024), or other reinforcement learning approaches. They differ in how this optimization problem is solved; for example, VGG-Flow (Liu

et al., 2025b) fine-tunes the velocity field via a reward-importance-weighted flow matching loss, whereas Adjoint Matching (Domingo-Enrich et al., 2025) back-propagates through the entire ODE trajectory using the continuous adjoint equations. Many fine-tuning methods require DDPM/SDE sampling for exploration during training (Liu et al., 2025a; Domingo-Enrich et al., 2025), which is significantly less efficient than ODE sampling and couples the method to a specific reward at training time; adapting to a new reward requires retraining from scratch. We instead focus on exploring how to best leverage the pretrained flow model at inference time, without any fine-tuning.

### A.2. Value gradient guidance

A related line of work defines the guidance signal as the gradient of a learned value function $g(x_t, t) \triangleq \nabla_{x_t} V(x_t, t)$, with $V(x_t, t) \approx \mathbb{E}[r(x_1) \mid x_t]$. VGG-Flow (Liu et al., 2025b) instantiates this idea by co-training a value-gradient network with the fine-tuned velocity via an HJB consistency loss. As a fine-tuning method, however, VGG-Flow couples the model to a single fixed reward at training time and targets a different problem from the one we tackle. In contrast, $g^{\mathrm{car}}$ focuses on off-manifold drift at inference time, redirecting trajectories back onto the data manifold via value-gradient guidance without modifying pretrained weights, and can be applied on top of any approximate guidance.

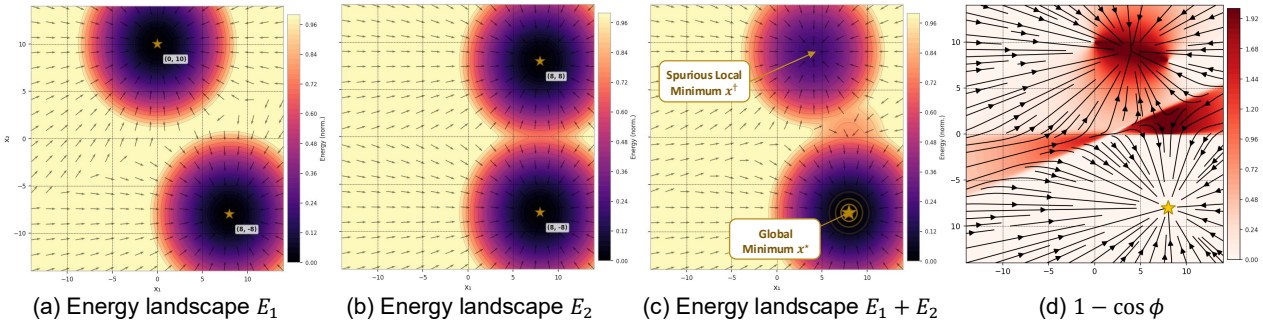

(a) Energy landscape $E_1$     (b) Energy landscape $E_2$     (c) Energy landscape $E_1 + E_2$     (d) $1 - \cos\phi$

*Figure 8.* **Spurious local minimum from gradient misalignment. (a, b)** Individual energy landscapes for two multi-modal reward functions, $E_j = -r_j$, each with two global minima (stars). One mode at $(8, -8)$ is shared, i.e., the $x_1^\star$. **(c)** The compositional energy landscape $E = E_1 + E_2 = -(r_1 + r_2)$ has a *spurious local minimum* $x^\dagger$ (top): $x^\dagger \neq x_1^\star$, and $x^\dagger$ maximizes neither any individual reward $r_j$ nor their sum. **(d)** The spurious local minimum coincides with the region of maximum gradient conflict (dark red, where $\nabla E_1 \approx -\nabla E_2$), where energy dissipation traps nearby trajectories rather than steering them to $x_1^\star$.

## B. Geometric interpretation: "energy trap" under gradient misalignment

Before the formal analysis in Appendix C, we provide a geometric interpretation of why gradient misalignment creates an "energy trap" in the guidance field.

**Global optimum and spurious local minimum.** Consider a compositional reward problem with $G$ reward functions $\{r_j\}_{j=1}^G$. The **global optimum** is expected to maximize all rewards:

$$x_1^\star = \arg\max_{x_1} \sum_{j=1}^G r_j(x_1). \tag{20}$$

From the energy-guided sampling perspective, the compositional energy landscape on the predicted terminal state $\hat{x}_1 = \mathbb{E}[x_1 \mid x_t]$ is $E(\hat{x}_1) \triangleq -\sum_{j=1}^G r_j(\hat{x}_1)$, with per-reward guidance $g_j(x_t) \triangleq \nabla_{x_t} r_j(\hat{x}_1)$ and compositional guidance $g_t(x_t) \triangleq \sum_{j=1}^G g_j(x_t)$. The guided trajectory evolves as $\dot{x}_t = v_t^{\text{base}}(x_t) + g_t(x_t)$, and the sampler's terminal states lie in the set of stable equilibria of $E$,

$$\mathcal{S} \triangleq \left\{ x : \nabla E(x) = 0, \ \text{Hess}\, E(x) \succ 0 \right\}. \tag{21}$$

By construction, $x_1^\star \in \mathcal{S}$: the global optimum is a stable equilibrium. In general, however, $\{x_1^\star\} \subsetneq \mathcal{S}$, formally:

**Definition B.1** (Spurious local minimum). A point $x^\dagger$ is a *spurious local minimum* of the compositional energy $E$ if it is a stable equilibrium that is *not* the global optimum:

$$x^\dagger \in \mathcal{S} \setminus \{x_1^\star\}, \quad \text{i.e.,} \quad \nabla E(x^\dagger) = 0, \ \text{Hess}\, E(x^\dagger) \succ 0, \ \text{and} \ x^\dagger \neq x_1^\star. \tag{22}$$

By construction, the compositional guidance vanishes at $x^\dagger$ ($\sum_{j=1}^G g_j(x^\dagger) = 0$), but $x^\dagger$ maximizes neither any individual reward $r_j$ nor their sum; the vanishing arises through *destructive interference between non-zero reward gradients* (Yu et al., 2020) rather than through reward maximization.[2]

As the trajectory approaches the basin of any $x^\dagger$, it drifts off-manifold.

**Energy dissipation under gradient misalignment.** To characterize the mechanism that drives trajectories off-manifold, we quantify the effective driving force of the compositional guidance via its squared norm. Expanding $\|g_t(x_t)\|^2$ at any

---

[2]From a probabilistic perspective, the log-density landscape $\sum_j r_j$ corresponds to a Product of Experts (PoE) formulation. A well-documented theoretical pathology of PoE and energy-based models is their propensity to generate *spurious modes* — unintended attractors that emerge between the unaligned peaks of the constituent distributions. In the optimization literature, these are formally referred to as *spurious local minima*.

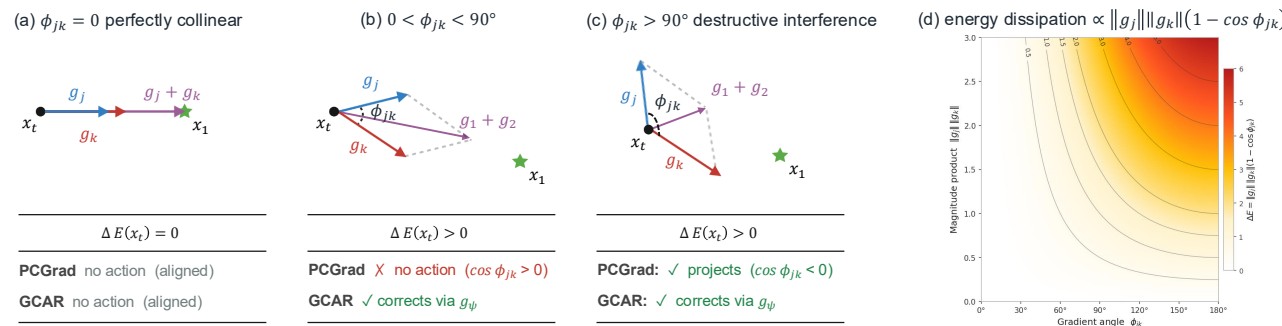

*Figure 9.* **Energy dissipation under gradient misalignment.** **(a)** When $\phi_{jk} = 0°$, reward gradients are perfectly collinear, $\Delta E(x_t) = 0$, and no correction is needed. **(b)** When $0° < \phi_{jk} < 90°$, gradients are misaligned but remain in the same half-space. PCGrad detects no conflict ($\cos \phi_{jk} > 0$) and takes no action, yet $\Delta E(x_t) > 0$; our $g^{\text{car}}$ identifies this misalignment and corrects it. **(c)** When $\phi_{jk} > 90°$, gradients undergo destructive interference. PCGrad intervenes via projection, whereas $g^{\text{car}}$ corrects the trajectory via learned residual guidance. **(d)** Energy dissipation $\Delta E \propto \|g_j\|\|g_k\|(1 - \cos \phi_{jk})$ scales with the gradient angle.

state $x_t$:

$$\left\|\sum_{j=1}^{G} g_j(x_t)\right\|^2 = \underbrace{\sum_{j} \|g_j(x_t)\|^2}_{\text{self-energy}} + \underbrace{2\sum_{j<k} \|g_j(x_t)\| \, \|g_k(x_t)\| \, \cos \phi_{jk}(x_t),}_{\text{cross-energy}} \tag{23}$$

where $\phi_{jk}(x_t)$ denotes the angle between $g_j(x_t)$ and $g_k(x_t)$. By the triangle inequality, the maximum compositional guidance is $\left(\sum_j \|g_j(x_t)\|\right)^2$, realized if and only if all gradients are perfectly collinear ($\cos \phi_{jk}(x_t) = 1$ for all $j < k$). We define the deficit between this collinear capacity and the realized compositional guidance as the energy dissipation:

**Definition B.2** (Energy dissipation under gradient misalignment)**.** For any state $x_t$, the *energy dissipation* of the compositional guidance is

$$\Delta E(x_t) \triangleq \left(\sum_j \|g_j(x_t)\|\right)^2 - \left\|\sum_j g_j(x_t)\right\|^2 = 2\sum_{j<k} \|g_j(x_t)\| \, \|g_k(x_t)\| \left(1 - \cos \phi_{jk}(x_t)\right) \geq 0. \tag{24}$$

$\Delta E(x_t) \geq 0$, with equality if and only if all reward gradients are perfectly aligned at $x_t$ ($\cos \phi_{jk}(x_t) \equiv 1$). As pairwise gradient misalignment grows, the compositional guidance energy structurally dissipates: the trajectory loses its driving force and becomes trapped at a spurious local minimum of the energy landscape (as visualised in Figure 9(d)).

At the terminal time step, the trajectory enters the basin of a spurious local minimum $x^{\dagger} \in \mathcal{S} \setminus \{x_1^{\star}\}$, where $\sum_j g_j(x^{\dagger}) = 0$ and the dissipated energy is $\Delta E(x^{\dagger}) = 2\sum_{j<k} \|g_j(x^{\dagger})\| \, \|g_k(x^{\dagger})\| \left(1 - \cos \phi_{jk}(x^{\dagger})\right)$.

**PCGrad and its structural limitations.** PCGrad (Yu et al., 2020) is a widely adopted gradient-conflict resolution method for multi-objective optimisation, but it addresses only destructive interference ($\cos \phi_{jk} < 0$) and resolves it via gradient surgery, projecting the conflicting component of each gradient onto the normal plane of the other. In standard multi-task optimisation, with smooth losses and thousands of accumulated optimiser steps, a transiently weakened gradient update is easily recovered in subsequent iterations, and the composite gradient reliably descends the loss landscape.

In inference-time guided sampling, however, this logic breaks down. The guidance signal $g_t(x_t)$ must steer the trajectory toward $x_1$ at every timestep, and there are no future updates at a fixed state to recover the dissipated guidance energy. Whenever pairwise misalignment $1 - \cos \phi_{jk} > 0$, trajectory loses driving force and drifts toward a spurious equilibrium $x^{\dagger}$.

# C. Guided sampling and approximation error

In this section, we analyze approximation errors utilizing the optimal coupling formulation and provide a detailed proof of the approximation error bound.

## C.1. Optimal coupling and triad decomposition

Optimal guided sampling modifies the generation process by reweighting the latent coupling $\pi(z)$ to target a tilted distribution $p_1^\star(x_1) \propto p_1(x_1)e^{r(x_1)}$. In the context of the transport problem, a change in the target marginal $p_1^\star(x_1)$ implies a modification of the optimal transport plan. To quantify this discrepancy, we consider the measures $\pi$ and $\pi^\star$ on the latent space. Assuming absolute continuity of the tilted coupling $\pi^\star(\cdot \mid x_1)$ with respect to the base coupling $\pi(\cdot \mid x_1)$ (i.e., $\pi^\star \ll \pi$), the Radon-Nikodym derivative exists and we term this derivative the coupling shift:

$$\mathcal{P}(z) \triangleq \frac{d\pi^\star(\cdot \mid x_1)}{d\pi(\cdot \mid x_1)}(z). \tag{25}$$

Formally, conditioned on a fixed terminal state $x_1$, $\mathcal{P}(z)$ measures the relative density shift between the conditional distribution of the true optimal transport plan $\pi^\star(\cdot \mid x_1)$ and the pre-trained base coupling $\pi(\cdot \mid x_1)$. If we were to simply perform Bayesian reweighting (as in standard classifier guidance), $\mathcal{P}(z) \equiv 1$. However, enforcing optimality in the transport cost introduces a shift $\mathcal{P}(z) \neq 1$.

Recall the definition of the conditional probability $\pi(z) = \pi(x_0 \mid x_1)p_1(x_1)$. Substituting the target marginal $p_1^\star(x_1) \propto p_1(x_1)e^{r(x_1)}$ and the coupling shift $\mathcal{P}(z) = \frac{\pi^\star(x_0|x_1)}{\pi(x_0|x_1)}$, we derive the triad decomposition (Equation (10) in main text) as follows:

$$\begin{aligned}
\pi^\star(z) &= \pi^\star(x_0 \mid x_1)\, p_1^\star(x_1) \\
&= [\mathcal{P}(z) \cdot \pi(x_0 \mid x_1)] \cdot \left[ \frac{1}{\mathcal{Z}^\star} p_1(x_1)e^{r(x_1)} \right] \\
&= \frac{1}{\mathcal{Z}^\star} \cdot \underbrace{\mathcal{P}(z)}_{\text{Coupling}} \cdot \underbrace{e^{R(z)}}_{\text{Reward}} \cdot \underbrace{\pi(z)}_{\text{Base Prior}},
\end{aligned} \tag{26}$$

where $R(z) \triangleq r(\Psi_1(z))$ denotes the trajectory-level reward, and $\pi(z) = \pi(x_0 \mid x_1)p_1(x_1)$. Note that, in a guided transport problem, the source distribution must remain anchored to the pre-defined prior (e.g., standard Gaussian) to ensure tractable inference; Thus, $\mathcal{P}(z)$ acts as a structural correction term: it represents the necessary re-organization of the transport plan, specifically the shift in the conditional $\pi(x_0 \mid x_1)$, necessary to satisfy the new target boundary $p_1^\star$ while simultaneously preserving the fixed source marginal $p_0$.

**Two-stage approximation.** As outlined in the main text, practical guided sampling simplifies Equation (26) via two approximations.

$$\pi^\star \xrightarrow[\mathcal{P}(z) \approx 1]{\text{Coupling-Invariant}} \pi^{\text{CI}} \xrightarrow[\hat{V}(x_t)]{\text{Local Approx.}} \pi^{\text{approx}}.$$

First, we have the **Coupling-Invariant Approximation (CIA)**, which assumes the conditional transport $\pi(x_0|x_1)$ remains unchanged, i.e., $\mathcal{P}(z) \equiv 1$:

$$\pi^{\text{CI}}(z) = \frac{e^{R(z)}}{\mathcal{Z}^{\text{CI}}}\, \pi(z), \qquad \mathcal{Z}^{\text{CI}} = \mathbb{E}_{\pi(z)}[e^{R(z)}]. \tag{27}$$

The coupling shift $\mathcal{P}(z)$ is required to anchor the transport to the fixed prior $p_0$. By assuming $\mathcal{P}(z) \equiv 1$, the Coupling-Invariant Approximation theoretically shifts the optimal source to a reweighted density $p_0^{\text{bias}}(x_0) \propto p_0(x_0)\mathbb{E}[e^{r(x_1)}|x_0]$. Since inference restricts sampling to $p_0$ rather than $p_0^{\text{bias}}$, a boundary mismatch arises. This discrepancy leads to off-manifold drift and error accumulation.

Second, to make the guidance realizable at any time step $t$, we apply a **Localized Approximation**. We approximate the trajectory reward $V(z)$ using a first-order Taylor expansion around the expected future state $\hat{x}_1 = \mathbb{E}[x_1|x_t]$: $\hat{V}(z) \approx V(x_t) + \nabla V(x_t)^\top(x_1 - \hat{x}_1)$. Since the constant terms cancel out during normalization, this results in the realizable coupling measure driven by the gradient:

$$\pi^{\text{approx}}(z) = \frac{e^{\hat{V}(z)}}{\hat{\mathcal{Z}}}\, \pi(z), \qquad \hat{\mathcal{Z}} = \mathbb{E}_{\pi(z)}[e^{\hat{V}(z)}]. \tag{28}$$

The effective guidance is thus determined solely by the gradient direction $\nabla V(x_t)$.[3]

Equation (26) explicitly demonstrates that the optimal coupling is governed by a triad of factors: the structural coupling shift ($\mathcal{P}(z)$), the trajectory reward ($e^{R(z)}$), and the base prior ($\pi(z)$). We formally derive the upper bound of the approximation error shortly. Before doing so, we revisit the two-stage approximation to clarify how it specifically targets these components: the coupling shift ($\mathcal{P}$) is neglected via the Coupling-Invariant assumption, and the trajectory reward ($e^{R(z)}$) is estimated via the Localized Approximation.

### C.2. Approximation error via the Benamou–Brenier theorem

Flow models construct probabilistic transport plans that move mass from a source measure $p_0$ to a target measure $p_1$, and the squared Wasserstein distance $W_2^2(p_0, p_1)$ quantifies the minimal kinetic energy required for this transport. By the Benamou–Brenier theorem:

$$W_2^2(p_0, p_1^\star) = \inf_{(p_t, v_t):\, p_0 \xrightarrow{v_t} p_1^\star} \int_0^1 \mathbb{E}_{x_t \sim p_t}\big[\|v_t(x_t)\|^2\big]\, dt = \int_0^1 \mathbb{E}\big[\|v_t^{\text{base}} + g_t^\star\|^2\big]\, dt, \tag{29}$$

where $g_t^\star$ is the optimal guidance field that steers mass toward the reward-tilted target $p_1^\star$. Under the two-stage approximation (CIA + Localized Approximation) for compositional rewards $R = \sum_j r_j$, $g_t^\star$ is replaced by the realized guidance $g_t^{\text{approx}} = \sum_j g_j^{\text{approx}}$, where each $g_j^{\text{approx}} = \nabla_{x_t} r_j(\hat{x}_1)$ with $\hat{x}_1 = \mathbb{E}[x_1 \mid x_t]$. The realized field $\hat{v}_t = v_t^{\text{base}} + g_t^{\text{approx}}$ therefore only transports $p_0$ to $\hat{p}_1 \neq p_1^\star$.

We quantify the resulting approximation error $\mathcal{E} \triangleq W_2^2(\hat{p}_1, p_1^\star)$ using the stability of the continuity equation (Villani et al., 2009), which bounds the terminal distributional discrepancy by the time-integrated squared velocity field difference along the optimal path $p_t^\star$:

$$\mathcal{E} \triangleq W_2^2(\hat{p}_1, p_1^\star) \leq \int_0^1 \mathbb{E}_{x_t \sim p_t^\star}\big[\|v_t^\star(x_t) - \hat{v}_t(x_t)\|^2\big]\, dt = \int_0^1 \mathbb{E}_{x_t \sim p_t^\star}\big[\|g_t^\star - g_t^{\text{approx}}\|^2\big]\, dt. \tag{30}$$

Since compositional guided sampling sums per-reward gradients directly, $g_t^{\text{approx}} = \sum_j g_j^{\text{approx}}$, the $\sum_j g_j^{\text{CI}}$ (distinct from $g_t^{\text{CI}}$ for $R$) sits naturally between $g_t^{\text{CI}}$ and $g_t^{\text{approx}}$. Young's inequality $\|a + b\|^2 \leq 2\|a\|^2 + 2\|b\|^2$ along the chain

$$g_t^\star \;\rightarrow\; g_t^{\text{CI}} \;\rightarrow\; \sum_j g_j^{\text{CI}} \;\rightarrow\; g_t^{\text{approx}},$$

yields three terms: the CIA step (first arrow), the Localized step (third), and an analytical decomposition (middle) specific to the compositional setting:

$$\mathcal{E} \leq \int_0^1 \mathbb{E}_{x_t \sim p_t^\star}\big[\|g_t^\star - g_t^{\text{approx}}\|^2\big]\, dt$$

$$\leq \int_0^1 \mathbb{E}_{x_t \sim p_t^\star}\left[2\|g_t^\star - g_t^{\text{CI}}\|^2 + 2\left\|g_t^{\text{CI}} - \sum_j g_j^{\text{CI}}\right\|^2 + 2\left\|\sum_j g_j^{\text{CI}} - g_t^{\text{approx}}\right\|^2\right] dt$$

$$\lesssim \underbrace{C_{\text{CI}} \int_0^1 \mathbb{E}\big[\mathbb{E}_{\pi^{\text{CI}}}[(\mathcal{P}(z) - 1)^2]\big]\, dt}_{\text{(A) Coupling shift error}} + \underbrace{\mathcal{K}_{\text{deficit}}}_{\text{(B) gradient misalignment error}} + \underbrace{G \int_0^1 \mathbb{E}\left[\left(\frac{\lambda_h \sigma_1 d}{e^{r(\hat{x}_1)}}\right)^2 (C_1 + C_2)\right] dt}_{\text{(C) Localized approximation error (Feng et al., 2025), scaled by } G}. \tag{31}$$

Each term corresponds to one step in the approximation chain. We detail each component below.

**(A) Coupling shift error.** Term (A) arises from replacing $g_t^\star$ with $g_t^{\text{CI}}$ under the Coupling-Invariant Approximation, $\mathcal{P}(z) := \frac{d\pi^\star(\cdot | x_1)}{d\pi(\cdot | x_1)}(z) \approx 1$, i.e., assuming the conditional transport plan requires no reorganisation when the target shifts

---

[3]Formally, let $C_t \triangleq V(x_t) - \nabla V(x_t)^\top \hat{x}_1$ denote the terms constant with respect to $z$. The normalization implies: $\pi^{\text{approx}}(z \mid x_t) = \frac{e^{C_t + \nabla V(x_t)^\top x_1} \pi(z|x_t)}{\int e^{C_t + \nabla V(x_t)^\top x_1} \pi(z|x_t)\, dz} = \frac{e^{C_t} e^{\nabla V(x_t)^\top x_1} \pi(z|x_t)}{e^{C_t} \int e^{\nabla V(x_t)^\top x_1} \pi(z|x_t)\, dz} = \frac{\cancel{e^{C_t}} e^{\nabla V(x_t)^\top x_1}}{\cancel{e^{C_t}} \hat{z}} \pi(z \mid x_t)$. This derivation shows that the guidance is driven purely by the gradient component $\nabla V(x_t)^\top x_1$, rendering the absolute magnitude of the value $V(x_t)$ irrelevant.

from $p_1$ to $p_1^\star$. By Cauchy–Schwarz:

$$\|g_t^\star(x_t) - g_t^{\mathrm{CI}}(x_t)\|_2^2 = \left\|\mathbb{E}_{z\sim\pi^{\mathrm{CI}}(\cdot|x_t)}\left[(\mathcal{P}(z) - 1)\, v_{t|z}(x_t|z)\right]\right\|_2^2$$
$$\leq \mathbb{E}_{z\sim\pi^{\mathrm{CI}}(\cdot|x_t)}\left[(\mathcal{P}(z) - 1)^2\right] \cdot \mathbb{E}_{z\sim\pi^{\mathrm{CI}}(\cdot|x_t)}\left[\|v_{t|z}(x_t|z)\|_2^2\right]. \tag{32}$$

Term (A) is small when the coupling shift is negligible ($\mathcal{P}(z) \approx 1$), which holds for flow matching methods with dependent couplings such as mini-batch OT-FM (Tong et al., 2024a), but not for vanilla OT-FM (Onken et al., 2021).

**(B) Gradient misalignment error.** We analyze Term (B) in two cases: (B1) When $G = 1$ or $\cos\phi_{jk} = 1$ for all pairs, Term (B) = 0. (B2) When $G > 1$ and $\cos\phi_{jk} < 1$ for some pair, Term (B) > 0.

In case (B2), we further quantify Term (B) by showing how the pointwise energy dissipation accumulates along the sampling trajectory, ultimately trapping it at a spurious local minimum at the terminal time step (Definition B.1).

**Proposition C.1** (Trajectory-level energy dissipation). *For $G \geq 2$, let $\cos\phi_t(x_t) := \frac{2}{G(G-1)}\sum_{j<k}\cos\phi_{jk}(x_t)$ denote the average pairwise cosine similarity at state $x_t$. Under the assumption $\|g_j^{\mathrm{CI}}\| \approx \mu$ for all $j$, the trajectory-level energy deficit is*

$$\mathcal{K}_{\mathrm{deficit}} \triangleq \int_0^1 \mathbb{E}_{x_t\sim p_t^\star}[\Delta E(x_t)]\, dt = G(G-1)\,\mu^2 \int_0^1 \mathbb{E}_{x_t\sim p_t^\star}[1 - \cos\phi_t(x_t)]\, dt \geq 0. \tag{33}$$

$\mathcal{K}_{\mathrm{deficit}} = 0$ *iff* $\cos\phi_t \equiv 1$ *almost everywhere, recovering case (B1). As conflict grows ($\cos\phi_t \to -1$), $\mathcal{K}_{\mathrm{deficit}}$ increases linearly in $(1 - \cos\phi_t)$ and quadratically in $G$ through $G(G-1)\mu^2$. This dissipated energy is structurally unavoidable under additive guidance and must be explicitly supplied by a corrective field $g_\psi$ to escape the basins of spurious local minima (Definition B.1).*

Note that the squared discrepancy $\|g_t^{\mathrm{CI}} - \sum_j g_j^{\mathrm{CI}}\|^2$ arises solely from gradient misalignment[4]. Hence, Term (B) in Eq. (31) is bounded by the energy deficit:

$$2\int_0^1 \mathbb{E}\left[\left\|g_t^{\mathrm{CI}} - \sum_j g_j^{\mathrm{CI}}\right\|^2\right] dt \lesssim \mathcal{K}_{\mathrm{deficit}}, \tag{34}$$

where absolute constants are absorbed into the $\lesssim$ symbol. In case (B1), $\mathcal{K}_{\mathrm{deficit}} = 0$.

**(C) Localized approximation error.** The third term arises from replacing each $g_j^{\mathrm{CI}}$ with its first-order Taylor estimate $g_j^{\mathrm{approx}}$ around $\hat{x}_1$. This linearization error has been analysed in detail by Feng et al. (2025) for the single-reward setting. For the compositional setting, Cauchy–Schwarz across the $G$ rewards yields

$$\left\|\sum_j g_j^{\mathrm{CI}} - g_t^{\mathrm{approx}}\right\|^2 \leq G\sum_j \|\delta g_j\|^2 \lesssim G\left(\frac{\lambda_h\,\sigma_1\,d}{e^{r(\hat{x}_1)}}\right)^2 (C_1 + C_2), \tag{35}$$

i.e., Feng et al. (2025)'s per-reward bound scaled linearly by $G$. Term (C) decreases when the reward is smooth (small $\lambda_h$), near $t \to 1$ (small $\sigma_1$), or $\hat{x}_1$ lies in a high-reward region (large $e^{r(\hat{x}_1)}$).

**Put it together.**

**Theorem C.2** (Upper bound of Approximation Error in Compositional Reward Setting). *Let $v_t^\star$ be the exact guided velocity field and $\hat{v}_t$ be the realized field under the Coupling-Invariant Approximation and Localized Approximation. The total*

---

[4]Expanding directly: $\|g_t^{\mathrm{CI}} - \sum_j g_j^{\mathrm{CI}}\|^2 = \|g_t^{\mathrm{CI}}\|^2 - 2\langle g_t^{\mathrm{CI}}, \sum_j g_j^{\mathrm{CI}}\rangle + \|\sum_j g_j^{\mathrm{CI}}\|^2$. Under the structural assumptions (i) $\|g_t^{\mathrm{CI}}\|^2 = (\sum_j \|g_j^{\mathrm{CI}}\|)^2$ (aligned-ideal magnitude) and (ii) $\langle g_t^{\mathrm{CI}}, \sum_j g_j^{\mathrm{CI}}\rangle = \|\sum_j g_j^{\mathrm{CI}}\|^2$ (full projection onto the sum direction), the self-terms $\sum_j \|g_j^{\mathrm{CI}}\|^2$ cancel, leaving $\|g_t^{\mathrm{CI}} - \sum_j g_j^{\mathrm{CI}}\|^2 = 2\sum_{j<k}\|g_j^{\mathrm{CI}}\|\|g_k^{\mathrm{CI}}\|(1 - \cos\phi_{jk}) = \Delta E(x_t)$.

*approximation error $\mathcal{E} \triangleq W_2^2(\hat{p}_1, p_1^\star)$ satisfies:*

$$\mathcal{E} \lesssim \underbrace{C_{\mathrm{CI}} \int_0^1 \mathbb{E}\big[\mathbb{E}_{\pi^{\mathrm{CI}}}\big[(\mathcal{P}(z) - 1)^2\big]\big]\, dt}_{\text{(A) coupling shift error}}$$

$$+ \underbrace{G(G-1)\mu^2 \int_0^1 \mathbb{E}[1 - \cos\phi_t(x_t)]\, dt}_{\text{(B) gradient misalignment error (Proposition C.1)}}$$

$$+ \underbrace{G \int_0^1 \mathbb{E}\left[\left(\frac{\lambda_h\, \sigma_1\, d}{e^{r(\hat{x}_1)}}\right)^2 (C_1 + C_2)\right] dt}_{\text{(C) localized approximation error (Feng et al., 2025)}}, \tag{36}$$

*where $C_{\mathrm{CI}} := \sup_{t \in [0,1]} \mathbb{E}_{\pi^{\mathrm{CI}}}[\|v_{t|z}\|_2^2]$ only depends on the base velocity field, $\cos\phi_t(x_t) := \frac{2}{G(G-1)}\sum_{j<k}\cos\phi_{jk}$ is the average pairwise cosine similarity at $(x_t, t)$, $\mu := \|g_j^{\mathrm{CI}}\|$ is the per-reward CI guidance magnitude. Following Feng et al. (2025), $\lambda_h$ is the spectral norm of the Hessian of $e^r$, $\sigma_1$ is the spectral norm of the conditional covariance $\Sigma_{1|t}$, and $C_1, C_2$ are constants depend on base flow. The error bound provides four key insights into the realized guidance $\hat{g}_t$:*

1. *The error is small when the **reward landscape is smooth**, i.e., small $\lambda_h = \|\nabla^2 e^r\|_2$. A flat reward landscape without sharp peaks or valleys implies less aggressive curvature, thereby minimizing the linearization error in Term (C).*

2. *The error is small when $\sigma_1$ **is small**, i.e., the conditional covariance $\Sigma_{1|t}$ has small spectral norm, meaning that at the current state $x_t$ the uncertainty about the terminal point $x_1$ is low. This is the case when the flow time $t \to 1$ (and $\sigma_t \to 0$), where $x_t$ reliably predicts $x_1$.*

3. *The **magnitude of** $e^{r(\hat{x}_1)}$ reflects how well the predicted endpoint $\hat{x}_1 = \mathbb{E}[x_1|x_t]$ matches the reward objective. If $\hat{x}_1$ lies inside the region where $r$ is large, the approximate guidance is more accurate, as the optimization is conducted locally and the gradient reflects the landscape well. If $e^{r(\hat{x}_1)}$ is small, the gradient explores the sample space almost randomly, producing larger approximation error.*

4. *The error scales with **the number of reward functions** $G$ and **gradient misalignment** $(1 - \cos\phi)$.*

# D. Guided sampling through the lens of fitted value evaluation

Unlike diffusion models, Flow models are governed by deterministic ODE processes. By leveraging this deterministic coupling and applying Jensen's inequality (or assuming the reward variance over the posterior is small), we approximate the soft value function with the expected return: $V(x_t, t) \approx \mathbb{E}_{z \sim \pi(z|x_t)}[r(x_1)]$. This simplification avoids the computational instability of the log-sum-exp operation while preserving the guidance direction.

However, a central challenge remains: the optimal guidance depends on the future endpoint $x_1 \sim p(x_1 \mid x_t)$, making analytical evaluation computationally prohibitive. We address this by introducing a value (reward-to-go) function $V(x, t)$ to summarize the expected terminal reward, modeled via the Bellman backup operator induced by the guided velocity field.

**Proposition D.1** (Fitted Value Evaluation). *Let $\mathcal{F}$ denote the function class (e.g., neural networks) used to approximate the value function. We collect a dataset of transitions $\mathcal{D} = \{(x, t, r, x', t')\}$ generated under the current guided dynamics, where $t' = t + \Delta t$ and $r$ is the reward. The value function can be estimated empirically by minimizing a least-squares Bellman residual:*

$$\hat{V}_{k+1} = \arg \min_{V \in \mathcal{F}} \ \mathbb{E}_{\mathcal{D}} \left[ \left( r + \gamma \hat{V}_k(x', t') - V(x, t) \right)^2 \right]. \tag{37}$$

*subject to the boundary condition $\hat{V}_k(x, 1) \equiv r(x)$ for terminal states. Here, $\gamma \in (0, 1]$ is the discount factor, and $\hat{V}_k$ is the target from the previous iteration [5]. Equation (37) empirically approximates the Bellman backup operator $\mathcal{T}^{v'}$ using finite data and a function class $\mathcal{F}$. Upon convergence, the learnable guidance is derived as the gradient of the estimated value:*

$$g(x_t, t) \triangleq \nabla_x \hat{V}(x_t, t), \tag{38}$$

*which provides a Markovian surrogate for the computationally expensive exact guidance (Equation (6)).*

Notice that the definition $g(x_t, t) \triangleq \nabla_x \hat{V}(x_t, t)$ has also appeared in prior work (Liu et al., 2025b), where it is motivated from an optimal control perspective in the context of controlled generation via differentiating through the ODE sampling process. Recall that our goal here is to estimate the Bellman backup operator $\mathcal{T}^{v'}$ (i.e., the generative dynamic from an intermediate state $x_t$ to the terminal state $x_1$). So we do not rely on $g^\star$ to quantify the optimality of the guidance term; instead, we use it purely as a tractable surrogate for the dependence of guidance on future states. Below, we provide a simple proof to justify this construction.

*Proof.* We view the generative dynamics as a Markov process whose policy is given by the guided velocity $v'(x, t) = v^{\text{base}}(x, t) + g(x, t)$. Under this policy, we define a value (reward-to-go) function that summarizes the expected future reward induced by the guided dynamics. Specifically, the value function is required to satisfy Bellman consistency

$$V(x, t) = \mathbb{E}[r + \gamma V(x', t') \mid x], \qquad x' \sim \mathcal{T}^{v'}(\cdot \mid x), \tag{39}$$

where $\mathcal{T}^{v'}$ denotes the transition operator induced by the guided velocity field (stepping from $t$ to $t'$) and $\gamma \in (0, 1]$ is a discount factor. In the generative setting considered here, the reward is sparse: $r = 0$ for all $t \in [0, 1)$, and reward is accrued only at the terminal state $x_1$.

The value function $V^{v'}$ is thus characterized as a fixed point of the Bellman operator $\mathcal{T}^{v'}$, i.e., $V^{v'} = \mathcal{T}^{v'} V^{v'}$. One possible approach is to compute the Bellman backup operator $\mathcal{T}^{v'}$ by exhaustive bootstrapping. However, in high-dimensional state spaces, it is infeasible to enumerate or traverse all states. We therefore approximate the Bellman operator by data and function approximation, implemented via Fitted Value Evaluation (FVE).

Let $\mathcal{F}$ denote a function class used to approximate the value function, and let $\mathcal{D} = \{(x^{(i)}, t^{(i)}, r^{(i)}, x'^{(i)}, t'^{(i)})\}_{i=1}^N$ be a dataset of one-step transitions collected from rollouts under the guided dynamics. We define the empirical Bellman backup

$$\widehat{\mathcal{T}}^{v'} V(x, t) \triangleq r + \gamma V(x', t'), \qquad (x, t, r, x', t') \sim \mathcal{D}. \tag{40}$$

Using this empirical operator, we perform fitted value evaluation (FVE) by iteratively projecting the Bellman backup onto $\mathcal{F}$:

$$V_{k+1} = \arg \min_{V \in \mathcal{F}} \ \mathbb{E}_{\mathcal{D}} \left[ \left( r + \gamma V_k(x', t') - V(x, t) \right)^2 \right]. \tag{41}$$

---

[5] For pure terminal optimization, $r = 0$ and $\gamma = 1$.

This procedure yields an empirical approximation $V$ that is Bellman-consistent in expectation with respect to the guided dynamics.

When $\mathcal{F}$ is large (or infinite) and $V$ is parameterized as $V_\theta \in \mathcal{F}$, Equation (41) is typically solved by stochastic optimization. In particular, treating the bootstrap target $y = r + \gamma V_{\theta_k}(x', t')$ as fixed, we minimize the squared regression error via a semi-gradient update:

$$\theta \leftarrow \theta - \alpha \, \nabla_\theta \Big( V_\theta(x, t) - \big[ r + \gamma V_{\theta_k}(x', t') \big] \Big)^2, \tag{42}$$

where $\alpha > 0$ is the learning rate. Repeating Equation (41) (or its stochastic variant Equation (42)) yields a Bellman-consistent value approximation for the guided dynamics.

The Bellman-consistent value function $V(x, t)$ summarizes the expected terminal reward attainable from the current state under the guided dynamics. Indeed, $\nabla_x V(x, t)$ points in the direction of steepest increase of the expected future reward, and therefore represents the locally optimal infinitesimal adjustment to the dynamics. This observation provides a principled bridge between value estimation and guidance construction: rather than explicitly conditioning on future endpoints $x_1$, guidance can be implemented as a local ascent direction induced by the value function gradient.

Once a Bellman-consistent value function $\hat{V}$ is obtained, we define the guidance vector field as

$$g(x, t) \triangleq \nabla_x \hat{V}(x, t), \tag{43}$$

which induces a local ascent direction in state space that maximally increases the expected terminal reward. This construction yields a Markovian and tractable surrogate for the otherwise future-dependent guidance implied by exact importance weighting. $\qquad \square$

This transformation effectively converts the intractable integral in Term (C) (localized approximation error) of Theorem 4.2 into a differentiable, Markovian vector field. However, Fitted Value Evaluation (FVE) can diverge even when theoretical conditions are met.

**Proposition D.2** (Divergence of Fitted Value Evaluation). *Fitted value evaluation (FVE) can diverge even when all of the following conditions hold:*

1. *The dataset is infinite, i.e., $|\mathcal{D}| = \infty$;*

2. *The Bellman residual minimization is solved exactly at each iteration;*

3. *The function class $\mathcal{F}$ is simple enough to be estimated, e.g., a one-dimensional linear function class $f_\theta(x) = \theta^\top \phi(x)$;*

4. *The realizability assumption holds, i.e., the true value function satisfies $V \in \mathcal{F}$.*

This phenomenon is commonly referred to as the *deadly triad* in empirical deep reinforcement learning, which arises from the interaction of function approximation, off-policy data, and bootstrapping. In the flow matching setting, however, the dynamics are deterministic and rewards are sparse (evaluated at terminal state $x_1$). We exploit this property to propose Terminal Value Regression, a method that directly fits the terminal reward. By removing the need for bootstrapping, this approach effectively breaks the deadly triad and ensures stable convergence.

**Proposition D.3** (Terminal Value Regression). *Let $\mathcal{F}$ denote a function class used to approximate the value function. We collect a dataset of terminal rollouts $\mathcal{D} = \{(x_t, t, x_1)\}$, where $x_1$ is the terminal state reached from $x_t$ by integrating the current guided dynamics. The value function is estimated by minimizing the following regression objective:*

$$\hat{V} = \arg \min_{V \in \mathcal{F}} \mathbb{E}_{(x_t, t, x_1) \sim \mathcal{D}} \Big[ \big( r(x_1) - V(x_t, t) \big)^2 \Big]. \tag{44}$$

*Unlike the bootstrapped target in Equation (37), the terminal reward $r(x_1)$ serves as a stable, unbiased regression target, which is enabled by the deterministic nature of the flow.*

Unlike fitted value evaluation, Equation (44) does not rely on bootstrapping and therefore avoids the instability associated with the deadly triad. Since the flow matching dynamics are deterministic, the terminal reward $r(x_1)$ serves as an unbiased Monte Carlo target for value estimation, yielding a stable procedure tailored to flow matching models.

# E. Experimental details

## E.1. Parameterization: value function vs. vector field

While directly parameterizing the vector field is common in diffusion models (Song & Kingma, 2021), unconstrained neural vector fields are not guaranteed to be conservative (i.e., curl-free) (Balcerak et al., 2025). Therefore, in Equation (19), we explicitly parameterize the scalar value function $V(x_t, t)$ and derive the guidance via automatic differentiation $\nabla_{x_t} V_\psi(x_t)$, ensuring that the learned guidance corresponds to the gradient of a valid scalar reward landscape. Crucially, as shown in Figure 10 (c-e), simply parameterizing $\nabla V(x_t, t)$ fails to rectify the off-manifold drift.

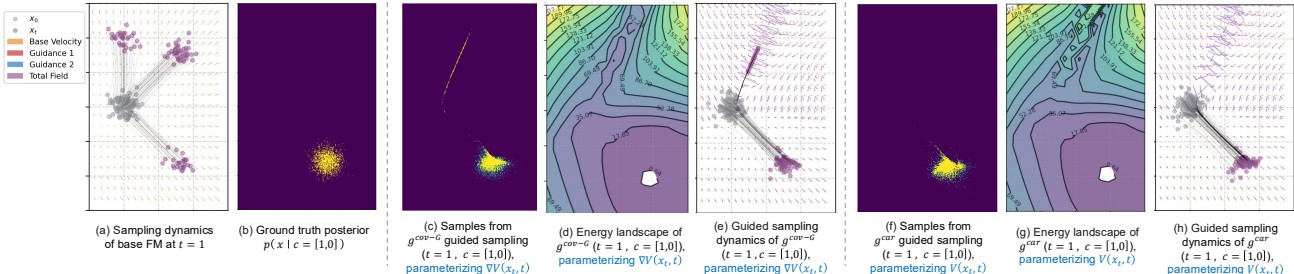

*Figure 10.* **Comparative empirical results on parameterization strategies.** We compare two architectures for learning the residual guidance: (Left: c–e) Directly parameterizing the unconstrained vector field (denoted as $\nabla V$). As shown in (d), this lack of structural constraint leads to a non-conservative field with a distorted, incoherent energy landscape, causing the "energy trap" and off-manifold drift in (e). (Right: f–h) Explicitly parameterizing the scalar value function $V$. By taking the gradient of a learned scalar $V$, we enforce the field to be curl-free by construction. This results in the smooth, globally consistent energy landscape in (g), effectively rectifying the drift as shown in (h).

Finally, to stabilize optimization when backpropagating through the parameterized value function $V(x_t, t)$, we apply gradient clipping to the derived gradients $\nabla V_\psi(x_t, t)$. This prevents exploding gradients, particularly in regions where the learned energy surface becomes steep or singular.

## E.2. Ablation: hard gate $\mathbb{I}_t$ and conflict threshold $\tau$

The $\tau$ controls the hard gate $\mathbb{I}_t$ in the training loss:

$$\mathcal{L}(\psi) = \mathbb{E}_{(x_t, t, x_1) \sim \mathcal{D}} \left[ \mathbb{I}_t \cdot \left( r(x_1) - V_\psi(x_t, t) \right)^2 \right]$$

and serves two purposes: (1) reducing unnecessary computation by skipping low-conflict regions, and (2) preserving $g^{\text{approx}}$ in those regions, where the approximate guidance is already accurate and adding a learned correction would introduce spurious perturbations. If $\tau$ is too small, neither purpose is met, as the gate activates almost everywhere. Conversely, too large a $\tau$ skips too many training steps, leaving $g_\psi$ under-trained.

We suggest that the threshold $\tau$ can be tuned according to the specific domain, and we empirically find that $\tau = 0.2$ is a robust sweet spot for most of our evaluated tasks (Maze2D, CelebA-HQ image editing, and ManiSkill2). For the synthetic benchmark, we use $\tau = 0.5$, which works better under its different conflict distribution. Therefore, we report experimental results using $\tau = 0.2$ for real-world domains and $\tau = 0.5$ for the synthetic benchmark. We show the ablation results of the conflict threshold $\tau$ in Figure 11. A threshold that is too small (e.g., $\tau = 0.0$) introduces spurious guidance in non-conflict regions (where approximation guidance $g^{\text{approx}}$ is already good enough), degrading performance (e.g., in the synthetic experiment, CS drops to 68.4% vs. ∼94% for $\tau \in [0.2, 0.5]$). Conversely, excessively high thresholds (e.g., $\tau = 0.8$) skip too many updates, leaving $g_\psi$ under-trained.

## E.3. Ablation: learned correction $g_\psi$ and conflict-aware weight $w_t$

Our method $g^{\text{car}}$ integrates a learned correction $g_\psi(x_t, t)$ and a conflict-aware weight $w_t$ into the guided velocity field:

$$v'_t(x_t, t) = v_t^{\text{base}}(x_t, t) + g^{\text{car}}(x_t, t),$$
$$g^{\text{car}}(x_t, t) = (1 - w_t) g^{\text{approx}} + w_t g_\psi(x_t, t).$$

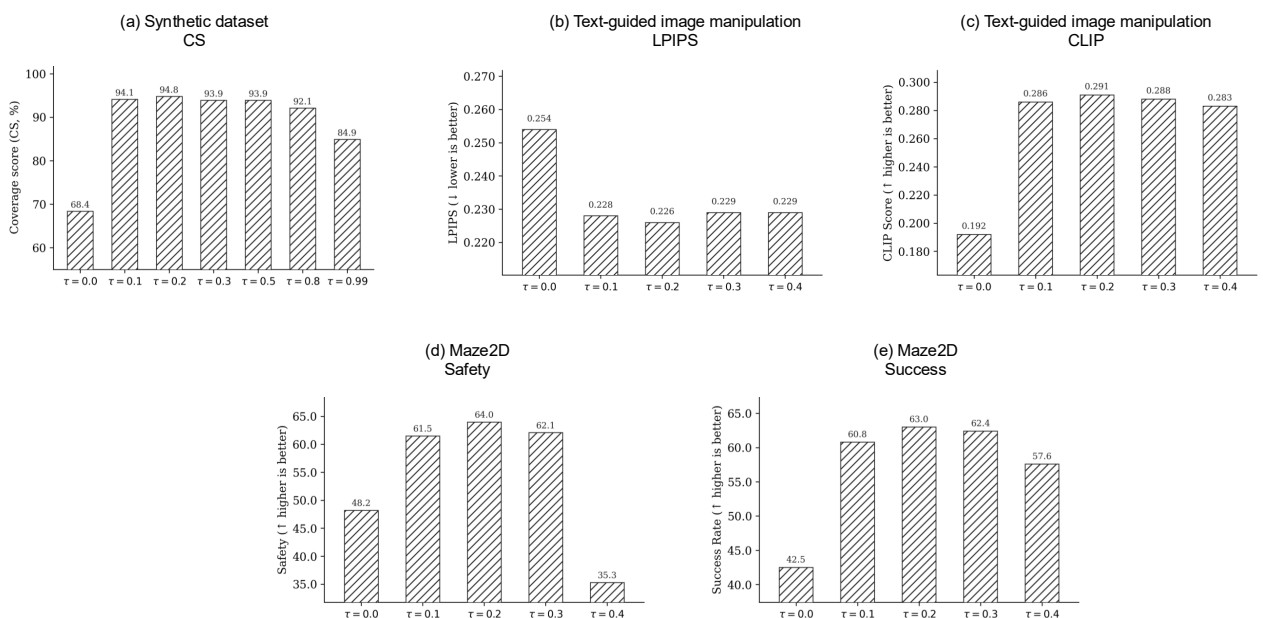

*Figure 11.* Ablation results on the conflict threshold $\tau$.

To understand the contribution of each component, we ablate $g_\psi$ and $w_t$ independently. Table 5 summarizes the three ablation studies, and Figure 12 reports the quantitative results on the synthetic benchmark.

Figure 12 reports results across three domains. Adding $g_\psi$ without the gate to $g^{\text{cov-G}}$ yields modest gains (synthetic CS +0.5pp, CelebA-HQ LPIPS $-0.014$, Maze2D success +5), confirming that the learned correction provides a useful residual signal to maximize rewards. Constraining the correction $g_\psi$ to conflict regions via adding $w_t$ gives much larger improvements (synthetic CS +9.8pp, PC +3.5pp; CelebA-HQ LPIPS $-0.021$, CLIP +0.011; Maze2D safety +15, success +9), demonstrating that the conflict-aware weight is the more critical component. Training loss curves in Figure 12(c,f,i) confirm stable convergence across all settings.

*Table 5.* Ablation study design for learned correction $g_\psi$ and conflict-aware weight $w_t$. All configurations share the same pretrained base velocity field $v^{\text{base}}$ and same approximation guidance $g^{\text{approx}}$ (i.e., $g^{\text{cov-G}}$).

| # | Guidance | $g^{\text{approx}}$ | $g_\psi$ | $w_t$ gate | Question to answer |
|---|---|---|---|---|---|
| (1) | $g^{\text{cov-G}}$ | ✓ | ✗ | ✗ | Baseline |
| (2) | $g^{\text{approx}} + g_\psi$ | ✓ | ✓ | ✗ | How much does the learned correction $g_\psi$ contribute? |
| (3) | $g^{\text{car}}$ (ours) | ✓ | ✓ | ✓ | How much does the conflict-aware weight $w_t$ contribute? |

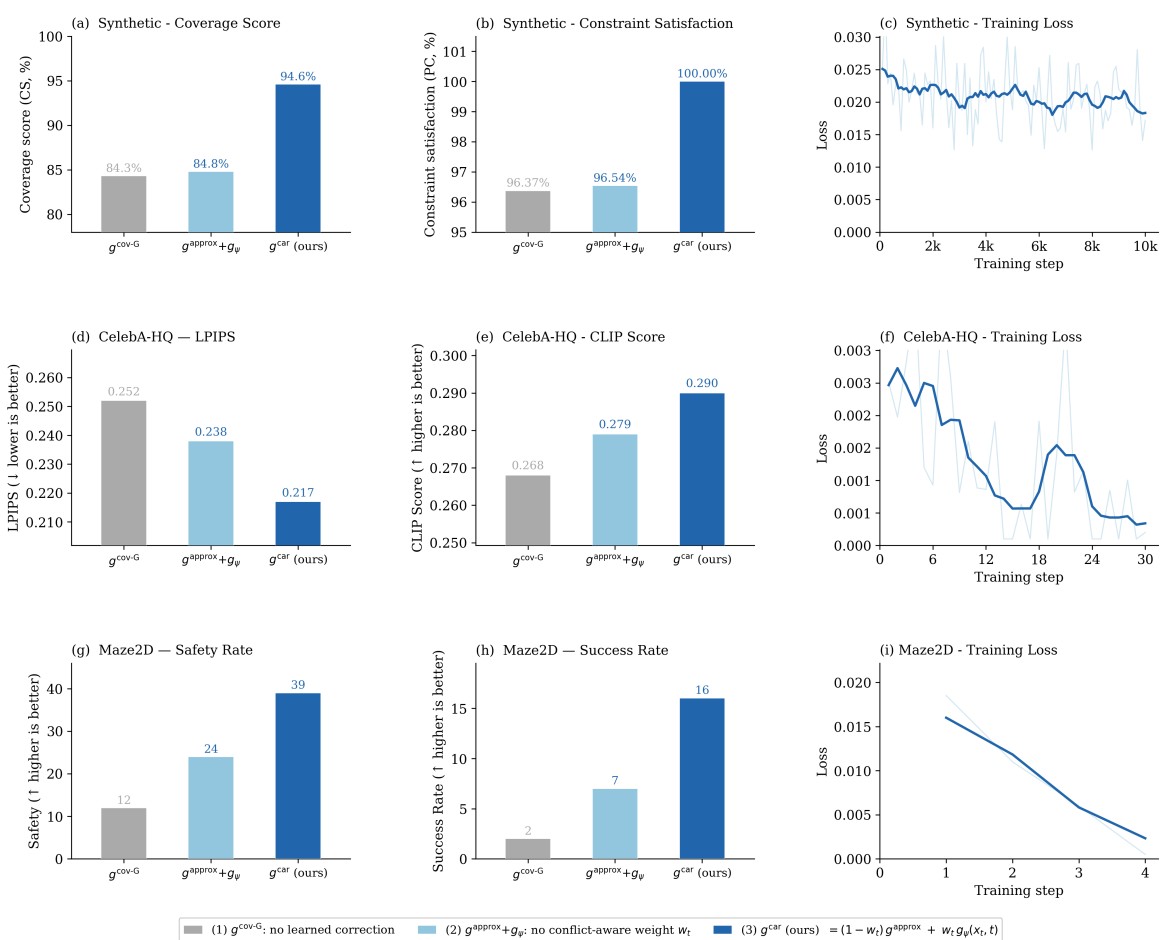

*Figure 12.* Component ablation on the synthetic benchmark. (a) Mode Coverage (CS) and (b) Prior Preservation (PC). The baseline $g^{\text{cov-G}}$ suffers from severe gradient conflicts. Applying the learned correction without the conflict gate ($g^{\text{approx}} + g_\psi$) improves PC but hurts CS due to spurious updates in low-conflict regions. Our full method $g^{\text{car}}$ leverages the gate $w_t$ to restrict corrections strictly to high-conflict states, achieving optimal performance in both metrics.

## E.4. Synthetic dataset

We consider a 2-dimensional Mixture of Gaussians toy example (see Figure 13), where the ground-truth density $p_t$ is known analytically, allowing for precise quantitative evaluation. The source distribution $\pi_0$ is a standard Gaussian $\pi_0(x) = \mathcal{N}(x \mid \mu_0, \Sigma_0)$, where $\mu_0 = [0.0, 0.0]$ and $\Sigma_0 = I$. The target distribution $\pi_1$ is a Mixture of Gaussians consisting of $K = 3$ modes, i.e., $\pi_1(x) = \frac{1}{3} \sum_{k=1}^{3} \mathcal{N}(x \mid \mu_k, \Sigma_1)$, where each component shares the covariance $\Sigma_1 = I$. We use a fixed configuration with centers located at $\mu_1 = [8.0, 8.0]$, $\mu_2 = [8.0, -8.0]$, and $\mu_3 = [0.0, 10.0]$, corresponding to the base posterior visualized in Figure 13 (b).

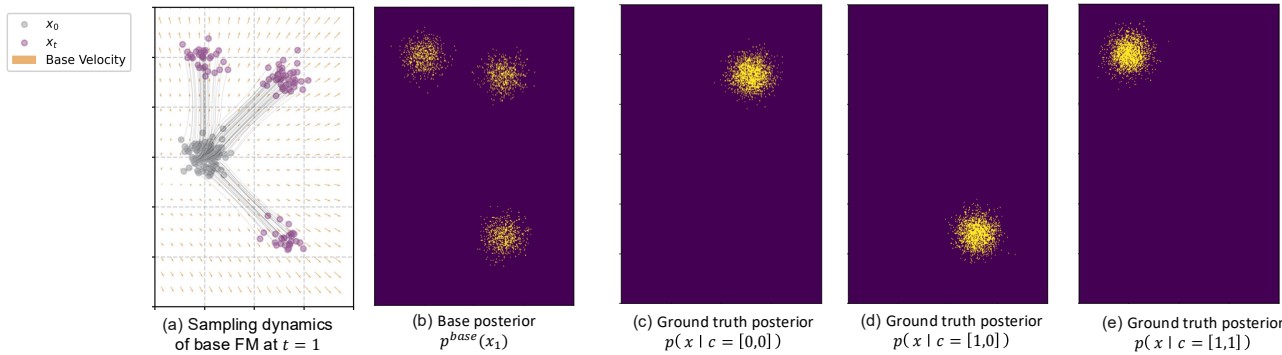

(a) Sampling dynamics of base FM at $t = 1$

(b) Base posterior $p^{base}(x_1)$

(c) Ground truth posterior $p(x \mid c = [0,0])$

(d) Ground truth posterior $p(x \mid c = [1,0])$

(e) Ground truth posterior $p(x \mid c = [1,1])$

*Figure 13.* **Visualization of synthetic experiments.** (a) The sampling dynamics of the base Rectified Flow model at $t = 1$. (b) The base posterior distribution $p^{base}(x_1)$ consisting of three Gaussian modes. (c)–(e) Ground-truth posteriors under different classifier constraints ($c = [0, 0]$, $[1, 0]$, and $[1, 1]$), estimated via rejection sampling with 10k samples.

### E.4.1. INFERENCE-TIME CONSTRAINTS

To evaluate the system under conflicting guidance, we employ two pre-trained binary classifiers, $\mathcal{C}_1$ and $\mathcal{C}_2$, which act as independent reward signals. Each classifier assigns a label $y \in \{0, 1\}$ to the generated samples. The classifier labels for the three target modes are designed to create varying degrees of gradient alignment. We visualize the ground-truth posteriors under these different compositional rewards in Figure 13 (c–e), generated via rejection sampling. Specifically, the constraint $c = [1, 0]$ (shown in Figure 13 (d)) represents a scenario with significant gradient conflict (or misalignment), serving as a primary stress test for off-manifold drift.

### E.4.2. EVALUATION METRICS

**(1) Posterior Coverage (PC) ($\uparrow$).** Fraction of generated samples residing within the $2\sigma$ boundary of ground-truth target mixture components, measured by anisotropic Mahalanobis distance:

$$\text{PC} = \frac{1}{N} \sum_{i=1}^{N} \mathbb{I} \left[ \min_{k \in \mathcal{K}_{\text{target}}} d(x_i, \mu_k) \leq 2 \right], \tag{45}$$

where $\mathcal{K}_{\text{target}}$ is the set of cluster indices satisfying the target labels. Unlike soft classifier probabilities (CS), PC is a strict geometric oracle: a sample is valid only if it physically resides within the correct high-density mode. **A drop in PC indicates off-manifold drift or biased sampling.**

**(2) Constraint satisfaction (CS) ($\uparrow$).** The average probability assigned to the target label $y$ by the guidance classifiers.

$$\text{CS} = \frac{1}{N} \sum_{i=1}^{N} p_\phi(y | x_i). \tag{46}$$

High CS indicates the guidance successfully optimizes the reward, potentially including adversarial examples that satisfy the classifier but fail PC.

**(3) Inference Time ($\downarrow$).** Wall-clock time per generated sample, aggregating: (i) trajectory generation (if applicable), (ii) learnable guidance training, and (iii) forward ODE solving. Note that GM collects data offline (highly parallelized), whereas $g^{car}$ relies on much slower online rollouts.

*Table 6.* Quantitative comparison on the synthetic benchmark. (1) Posterior Coverage (PC): lower values indicate off-manifold drift or biased sampling. (2) Constraint Satisfaction (CS): average classifier probability for the target labels. (3) Time ($\downarrow$): wall-clock time per sample (ms). (4) Data Usage ($\times 10^3$, $\downarrow$): total training samples consumed, reported in units of $10^3$. All results use conflict threshold $\tau=0.50$; $\epsilon$ denotes the early-stopping threshold. Each number is evaluated with 10k generated samples. Data Usage of $g^{\text{car}}$ reports the total number of training samples consumed before the fraction of conflict samples drops below $\epsilon$ (see early-stopping criterion in Appendix E.4.3).

| Metric | Method | Single Guidance | | | | | Composed Guidance | | | |
|---|---|---|---|---|---|---|---|---|---|---|
| | | $[0, \varnothing]$ | $[1, \varnothing]$ | $[\varnothing, 0]$ | $[\varnothing, 1]$ | Avg | $[0, 0]$ | $[1, 0]$ | $[1, 1]$ | Avg |
| PC (%) ↑ | $g^{\text{cov-G}}$ | **95.10**±0.02 | **91.50**±0.03 | 90.80±0.02 | **89.90**±0.04 | **91.83**±0.03 | 94.00±0.03 | 71.70±0.04 | 89.00±0.03 | 84.90±0.03 |
| | PCGrad | **95.10**±0.02 | **91.50**±0.03 | 90.80±0.02 | **89.90**±0.04 | **91.83**±0.03 | 94.05±0.04 | 75.20±0.08 | 89.15±0.05 | 86.13±0.06 |
| | GM | 87.40±0.12 | 82.00±0.15 | 85.30±0.11 | 88.70±0.14 | 85.85±0.13 | 82.30±0.22 | 84.50±0.38 | 86.20±0.18 | 84.33±0.26 |
| | GLASS-FKS ($K=4$) | 85.20±0.80 | 86.20±3.60 | 86.20±2.40 | 80.00±7.10 | 84.40±3.40 | 84.20±2.30 | 78.60±3.00 | 79.80±1.60 | 80.90±2.30 |
| | GLASS-FKS ($K=16$) | 87.80±4.40 | 87.80±3.30 | 88.80±4.60 | 87.20±5.20 | 87.90±4.30 | 88.80±3.30 | 90.80±1.60 | 88.60±3.60 | 89.40±2.80 |
| | $g^{\text{car}}$ ($\epsilon=0.10$) | **95.10**±0.02 | **91.50**±0.04 | **91.00**±0.03 | 89.70±0.05 | **91.83**±0.03 | 93.90±0.04 | 83.70±0.06 | 88.80±0.03 | 88.80±0.04 |
| | $g^{\text{car}}$ ($\epsilon=0.05$) | 95.00±0.03 | **91.50**±0.02 | **91.00**±0.02 | 89.60±0.04 | 91.78±0.03 | 94.60±0.03 | 93.80±0.05 | 88.70±0.04 | 92.37±0.04 |
| | $g^{\text{car}}$ ($\epsilon=0.00$) | **95.10**±0.01 | **91.50**±0.01 | 90.90±0.02 | 89.70±0.03 | 91.80±0.02 | **96.80**±0.02 | **94.40**±0.04 | **94.10**±0.03 | **95.10**±0.03 |
| CS (%) ↑ | $g^{\text{cov-G}}$ | **100.00**±0.00 | **100.00**±0.00 | **100.00**±0.00 | 99.87±0.01 | **99.97**±0.00 | **100.00**±0.00 | 89.56±0.03 | 99.87±0.01 | 96.48±0.01 |
| | PCGrad | **100.00**±0.00 | **100.00**±0.00 | **100.00**±0.00 | 99.87±0.01 | **99.97**±0.00 | 100.00±0.00 | 92.45±0.12 | 99.88±0.02 | 97.44±0.05 |
| | GM | 99.91±0.08 | 99.91±0.09 | 99.95±0.05 | **100.00**±0.01 | 99.94±0.05 | 99.93±0.07 | 99.99±0.01 | 99.92±0.08 | 99.95±0.05 |
| | GLASS-FKS ($K=4$) | 92.64±2.44 | 99.80±0.45 | 99.80±0.45 | 90.00±2.55 | 95.56±1.47 | 93.70±1.48 | 90.31±1.76 | 90.90±2.74 | 91.64±1.99 |
| | GLASS-FKS ($K=16$) | **100.00**±0.00 | **100.00**±0.00 | **100.00**±0.00 | **100.00**±0.00 | **100.00**±0.00 | **100.00**±0.00 | 99.71±0.27 | **100.00**±0.00 | 99.90±0.09 |
| | $g^{\text{car}}$ ($\epsilon=0.10$) | **100.00**±0.00 | **100.00**±0.00 | **100.00**±0.00 | 99.86±0.02 | **99.97**±0.00 | **100.00**±0.00 | 99.34±0.04 | 99.87±0.01 | 99.74±0.01 |
| | $g^{\text{car}}$ ($\epsilon=0.05$) | **100.00**±0.00 | **100.00**±0.00 | **100.00**±0.00 | 99.86±0.01 | **99.97**±0.00 | **100.00**±0.00 | **100.00**±0.00 | 99.87±0.02 | 99.96±0.00 |
| | $g^{\text{car}}$ ($\epsilon=0.00$) | **100.00**±0.00 | **100.00**±0.00 | **100.00**±0.00 | 99.87±0.01 | **99.97**±0.00 | **100.00**±0.00 | **100.00**±0.00 | **100.00**±0.00 | **100.00**±0.00 |
| Time↓ | $g^{\text{cov-G}}$ | 0.35±0.01 | 0.37±0.01 | 0.35±0.01 | 0.35±0.01 | 0.36±0.01 | 0.36±0.01 | **0.38**±0.01 | 0.36±0.01 | **0.37**±0.01 |
| | PCGrad | 0.35±0.01 | 0.37±0.01 | 0.35±0.01 | 0.35±0.01 | 0.36±0.01 | 0.37±0.01 | 0.42±0.02 | 0.38±0.01 | 0.39±0.01 |
| | GM | 3.09±0.12 | 3.02±0.14 | 2.96±0.11 | 2.96±0.13 | 3.01±0.12 | 2.81±0.15 | 2.81±0.16 | 3.11±0.12 | 2.91±0.14 |
| | GLASS-FKS ($K=4$) | 293.00±37.00 | 293.00±36.00 | 295.00±36.00 | 294.00±34.00 | 294.00±35.00 | 295.00±36.00 | 297.00±33.00 | 292.00±36.00 | 295.00±35.00 |
| | GLASS-FKS ($K=16$) | 299.00±40.00 | 301.00±34.00 | 295.00±35.00 | 294.00±35.00 | 297.00±36.00 | 298.00±34.00 | 296.00±36.00 | 302.00±36.00 | 299.00±35.00 |
| | $g^{\text{car}}$ ($\epsilon=0.10$) | **0.25**±0.01 | 0.25±0.01 | 0.25±0.01 | 0.25±0.01 | **0.25**±0.01 | **0.27**±0.02 | 0.72±0.03 | **0.27**±0.02 | 0.42±0.02 |
| | $g^{\text{car}}$ ($\epsilon=0.05$) | **0.25**±0.01 | 0.25±0.01 | 0.25±0.01 | 0.25±0.01 | **0.25**±0.01 | 0.46±0.02 | 4.20±0.05 | **0.27**±0.01 | 1.65±0.03 |
| | $g^{\text{car}}$ ($\epsilon=0.00$) | 0.30±0.02 | **0.18**±0.01 | **0.22**±0.01 | **0.21**±0.01 | **0.23**±0.01 | 25.81±0.12 | 25.81±0.15 | 26.01±0.11 | 25.88±0.12 |
| Data↓ | $g^{\text{cov-G}}$ | – | – | – | – | – | – | – | – | – |
| | PCGrad | – | – | – | – | – | – | – | – | – |
| | GM | 10240.00 | 10240.00 | 10240.00 | 10240.00 | 10240.00 | 10240.00 | 10240.00 | 10240.00 | 10240.00 |
| | GLASS-FKS | – | – | – | – | – | – | – | – | – |
| | $g^{\text{car}}$ ($\epsilon=0.10$) | 0 | 0 | 0 | 0 | 0 | **0** | 180.22±15.45 | **0** | 60.07±5.15 |
| | $g^{\text{car}}$ ($\epsilon=0.05$) | 0 | 0 | 0 | 0 | 0 | 74.75±6.33 | 1573.89±85.30 | **0** | 549.55±30.54 |
| | $g^{\text{car}}$ ($\epsilon=0.00$) | 0 | 0 | 0 | 0 | 0 | 10240.00 | 10240.00 | 10240.00 | 10240.00 |

*Note:* **Bold** indicates best performance. The $[1, 0]$ column highlights the gradient conflict scenario. Mean±std are reported over 5 random seeds.

**(4) Data Usage** ($\downarrow$) denotes the total number of training trajectories (from $x_0$ to $x_1$) required to learn the learnable guidance; lower is more data-efficient.

### E.4.3. EXPERIMENTAL RESULTS

We present comprehensive quantitative results in Table 6, evaluated with 10k generated samples per setting. The evaluation covers all valid constraint configurations: (i) Single Guidance: $[0, \varnothing]$, $[1, \varnothing]$, $[\varnothing, 0]$, and $[\varnothing, 1]$; (ii) Composed Guidance: $[0, 0]$, $[1, 0]$, and $[1, 1]$. Note that no data samples satisfy $[0, 1]$.

$g^{\text{car}}$ **resolves off-manifold drift efficiently.** Under single guidance, all methods achieve competitive Posterior Coverage (PC, >85%) and Constraint Satisfaction (CS, >99%). However, in the compositional reward setting—especially the $[1, 0]$ gradient conflict scenario in Table 6—significant performance gaps emerge. With $\epsilon=0.05$, $g^{\text{car}}$ achieves $93.80\% \pm 0.05$ PC and a perfect $100.00\% \pm 0.00$ CS on the $[1, 0]$, outperforming GLASS-FKS ($K=16$) by 3.0 PC points while operating at a $70\times$ lower inference cost ($4.20 \pm 0.05$ vs. $\approx 296$ ms/sample). Employing a tighter threshold ($\epsilon=0.00$) further maximizes composed guidance fidelity (averaging $95.10\% \pm 0.03$ PC) at the expense of maximum training data usage, whereas $\epsilon=0.10$ provides highly competitive composed performance ($88.80\% \pm 0.04$ PC average) with near-zero transition data requirements.

Key observations are:

- $g^{\text{cov-G}}$ collapses under gradient conflict. $g^{\text{cov-G}}$ degrades sharply to $71.70\% \pm 0.04$ PC on $[1,0]$.

- GLASS-FKS (sample-based) avoids off-manifold drift but is computationally costly and is highly sensitive to the particle count $K$. As shown in Table 6, GLASS-FKS maintains consistent Constraint Satisfaction (CS) scores across all compositional scenarios (i.e., $[0,0]$, $[1,0]$, and $[1,1]$), and does not have a severe performance drop under the conflicting $[1,0]$ setting. However, when the number of particles is restricted (e.g., $K{=}4$), the variance increases.

- Guidance Matching suffers from confounding errors inherent in learning a guidance network from scratch, yielding a lower average PC of $84.33\%$. Moreover, GM requires over $10^7$ training samples per compositional reward (approx. $20\times$ more than $g^{\text{car}}$).

- PCGrad didn't manage to correct off-manifold drift.

- Our $g^{\text{car}}$ efficiently corrects off-manifold drift while remaining compute-light.

**Data efficiency via early stopping.** To evaluate the impact of the conflict-aware module on **data efficiency**, we introduce an early-stopping mechanism parameterized by $\epsilon$. Training is halted when the proportion of generated samples with conflict score exceeding $\tau$ drops below $\epsilon$, i.e., $P(\text{score} > \tau) < \epsilon$. This criterion indicates that $x_1$ has sufficiently resolved gradient conflicts. Table 6 reports $g^{\text{car}}$ under $\epsilon \in \{0.10, 0.05, 0.00\}$; the training dynamics are visualized in Figure 14. As shown, the conflict score decreases stably across all settings, demonstrating that $g^{\text{car}}$ reliably learns to minimize gradient conflicts and rectify off-manifold drift over time.

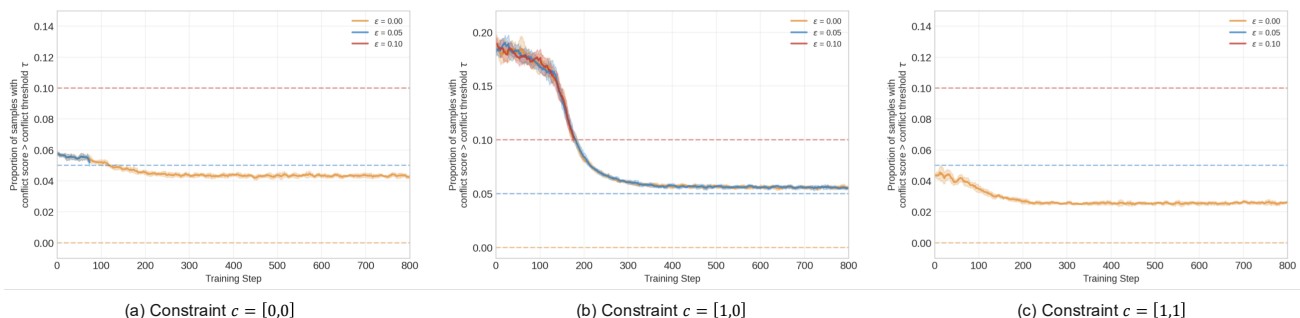

(a) Constraint $c = [0,0]$      (b) Constraint $c = [1,0]$      (c) Constraint $c = [1,1]$

*Figure 14.* **Convergence of conflict scores of $g^{\text{car}}$.** The figure tracks the fraction of online samples with a conflict score larger than the early-stopping threshold $\epsilon$. This metric serves as an indicator for training stability. Results are shown for targets (a) $c = [0,0]$, (b) $c = [1,0]$, and (c) $c = [1,1]$ across three early-stopping threshold ($\epsilon = 0.00, 0.05$ and $0.10$), with a conflict threshold $\tau = 0.50$. The downward trend indicates that the conflict scores of online samples progressively decrease, demonstrating that $g^{\text{car}}$ effectively learns to minimize gradient conflicts and rectify off-manifold drift. Shaded areas represent the standard error across 5 random seeds.

## E.5. Generative decision-making as planners

### E.5.1. INFERENCE-TIME CONSTRAINTS

**Static obstacle rewards.** We formulate static obstacle avoidance as a differentiable, energy-based reward function $r_{\text{static}}(\mathbf{x})$, following (Luo et al., 2024), which provides a smooth, bounded penalty landscape, thereby stabilizing the gradient-based guidance $\nabla_{\mathbf{x}} r_{\text{static}}(\mathbf{x})$ at inference time.

Formally, the obstacles are defined as a set of $K$ centers $\{\mathbf{c}_k\}_{k=1}^K$. The compositional static obstacle reward at state $\mathbf{x}$ is defined as:

$$r_{\text{static}}(\mathbf{x}) = -\sum_{k=1}^K \exp\left(-\frac{\|\mathbf{x} - \mathbf{c}_k\|^2}{\sigma^2}\right) \tag{47}$$

where $\sigma$ determines the spatial decay rate of the repulsive potential, i.e., the influence diminishes as the distance from the center increases. We set $\sigma = 2.0$, with the remaining settings unchanged.

**Static goal rewards.** We consider instruction-following scenarios (e.g., "fetch an apple"), where the objective is to reach specific spatial locations. We formulate the guidance for reaching these goals using the same differentiable, energy-based formulation as Equation (47).

Formally, we define the goals as a set of $K$ centers $\{\mathbf{g}_k\}_{k=1}^K$. The compositional static goal reward at state $\mathbf{x}$ is defined as:

$$r_{\text{goal}}(\mathbf{x}) = \sum_{k=1}^K \exp\left(-\frac{\|\mathbf{x} - \mathbf{g}_k\|^2}{\sigma^2}\right) \tag{48}$$

where $\sigma$ is the spatial decay rate.

**Dynamic obstacle rewards.** We consider the dynamic agent avoidance task, where obstacles follow randomly generated linear trajectories.

Formally, we define a set of $K$ dynamic obstacles. The generated robot trajectory is $\boldsymbol{\tau} = \{\mathbf{x}_1, \ldots, \mathbf{x}_H\}$ over a planning horizon $H$. Each obstacle $k$ moves over a horizon of $N$ steps ($N \leq H$; i.e., its position $\mathbf{c}_k(t)$ updates for the first $N$ steps) and remains stationary thereafter (i.e., $\mathbf{c}_k(t) = \mathbf{c}_k(N)$ for $t > N$). The compositional reward for the entire trajectory $\boldsymbol{\tau}$ is defined as:

$$r_{\text{dynamic}}(\boldsymbol{\tau}) = -\sum_{t=1}^H \sum_{k=1}^K \exp\left(-\frac{\|\mathbf{x}_t - \mathbf{c}_k(t)\|^2}{\sigma^2}\right) \tag{49}$$

where $\mathbf{x}_t$ denotes the agent state at time step $t$, and $\sigma$ is the spatial decay rate. We set the trajectory horizon $H = 48$ and the obstacle trajectory horizon $N = 3$.

**Trajectory smoothness rewards.** We set the trajectory smoothness cost (Urain et al., 2023). Formally, given a trajectory $\boldsymbol{\tau} = \{\mathbf{x}_0, \ldots, \mathbf{x}_T\}$, the smoothness reward is defined as:

$$r_{\text{smooth}}(\boldsymbol{\tau}) = -\sum_{t=0}^{T-1} \|\mathbf{x}_{t+1} - \mathbf{x}_t\|^2 \tag{50}$$

as the minimization of the relative distance between the neighbour points in the trajectory. This reward can be thought as a spring making all the point in the trajectory be attracted between each other.

### E.5.2. HYPERPARAMETER

We provide the detailed hyperparameters used for the Maze2D experiments in Table 7.

### E.5.3. EXPERIMENTAL RESULTS

We report all experimental results in Table 2, the key observations are:

*Table 7.* Hyperparameters for $g^{\text{car}}$ used in Maze2D.

| Hyperparameter | Value |
| --- | --- |
| Euler discretization steps (training) | $N = 100$ |
| Euler discretization steps (inference) | $N = 100$ |
| Guidance training Steps | $M = 4$ |
| Trajectory horizon $H$ | 48 |
| ODE inference steps | 100 |
| Guidance scale | 1.0 |

1. Inference-time guidance applied to pre-trained generative policy models is prone to off-manifold drift, leading to poor prior preservation (e.g., failing to reach the end point) and constraint violations (e.g., colliding with obstacles or maze walls), as shown for $g^{\text{cov-G}}$ in Figure 16.

2. PCGrad cannot recover from off-manifold drift.

3. GLASS-FKS performs well on robot planning tasks.

4. $g^{\text{car}}$ consistently corrects off-manifold drift across all settings, improving success rate and reducing constraint violations.

5. MPPI is a strong planning baseline that refines generated paths from the base CFM model to satisfy runtime constraints. Sometimes, it still suffers from prior preservation issues under compositional constraints. When $g^{\text{car}}$ is applied on top of MPPI, MPPI + $g^{\text{car}}$ achieves the best overall performance, correcting off-manifold drift while satisfying constraints.

We further evaluate robustness to clutter by varying the number of static obstacles from 2 to 6 (Figure 15). $g^{\text{cov-G}}$ degrades sharply as the environment becomes more cluttered, with its success rate collapsing to $12\%$ at 6 obstacles, whereas $g^{\text{car}}$ maintains $34\%$ in the same setting.

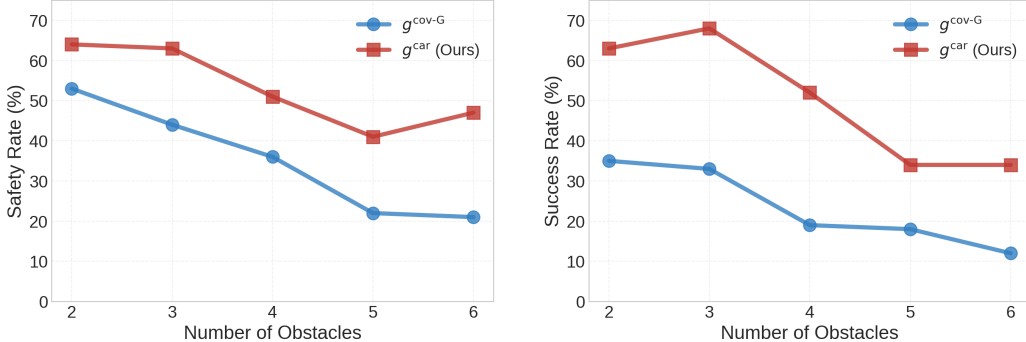

*Figure 15.* Robustness to clustered environments on Maze2D. We evaluate the safety and success rates by varying the number of static obstacles from 2 to 6. Our $g^{\text{car}}$ (red) exhibits robustness even with 6 obstacles, whereas $g^{\text{cov-G}}$ suffers degradation.

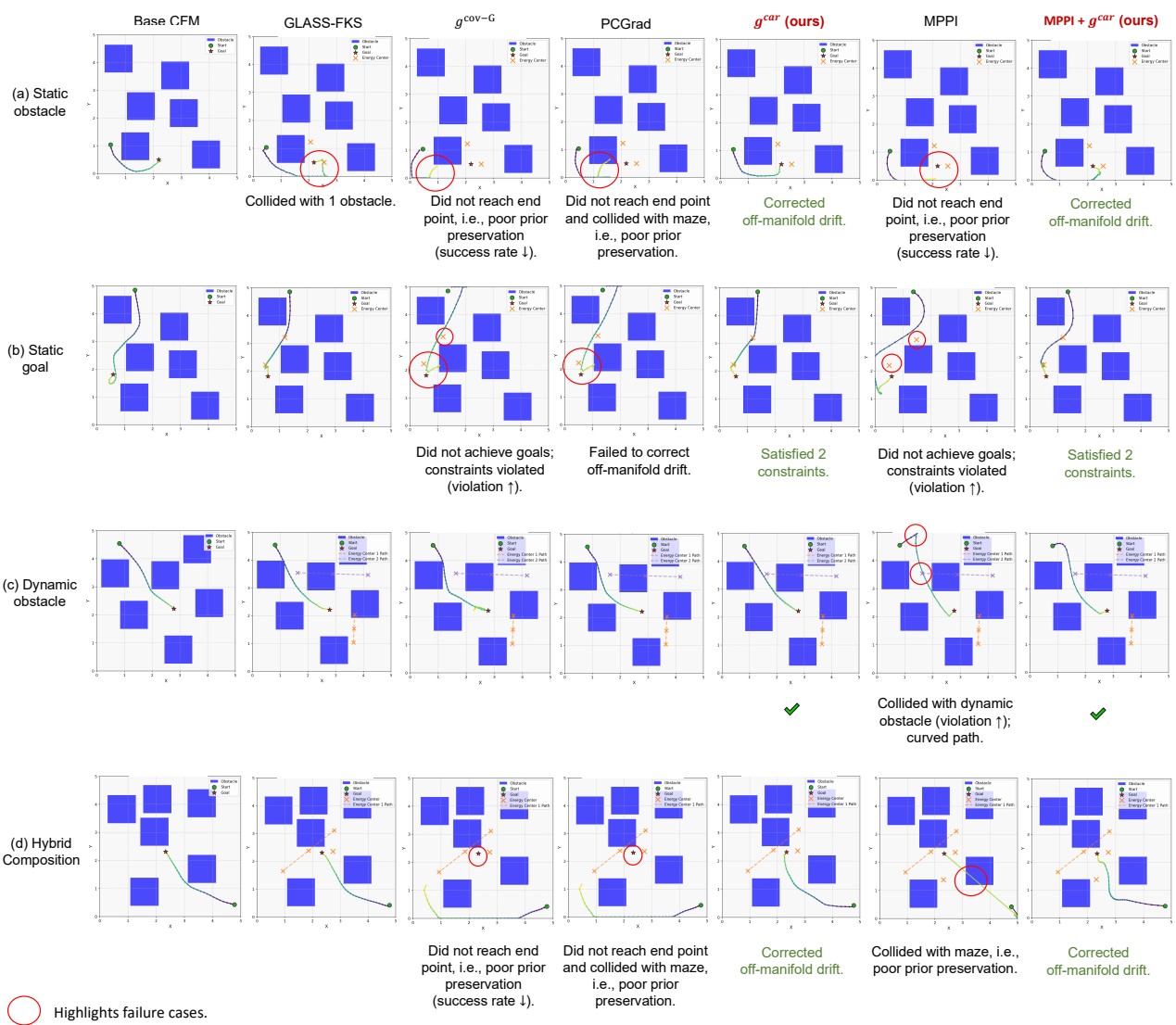

📌 **We do not use inpainting**, enabling better observation of inference-time alignment methods' ability to preserve the base model prior (the base CFM is trained to generate collision-free paths conditioned on maze layout, start, and end points); Comparing success rates reveals that directly applying GLASS-FKS, MPPI, $g^{\mathrm{cov-G}}$, and their PCGrad or $g^{car}$ corrections as inference-time techniques compromises prior preservation.

*Figure 16.* **Visualisation of guided trajectory generation under compositional constraints in Maze2D.** (1) static obstacles, (2) goal reachability, (3) dynamic obstacles, and (4) hybrid composition. Observe that $g^{\mathrm{cov-G}}$ produces erratic, off-manifold trajectories, while $g^{\mathbf{car}}$ yields smooth, feasible trajectories.

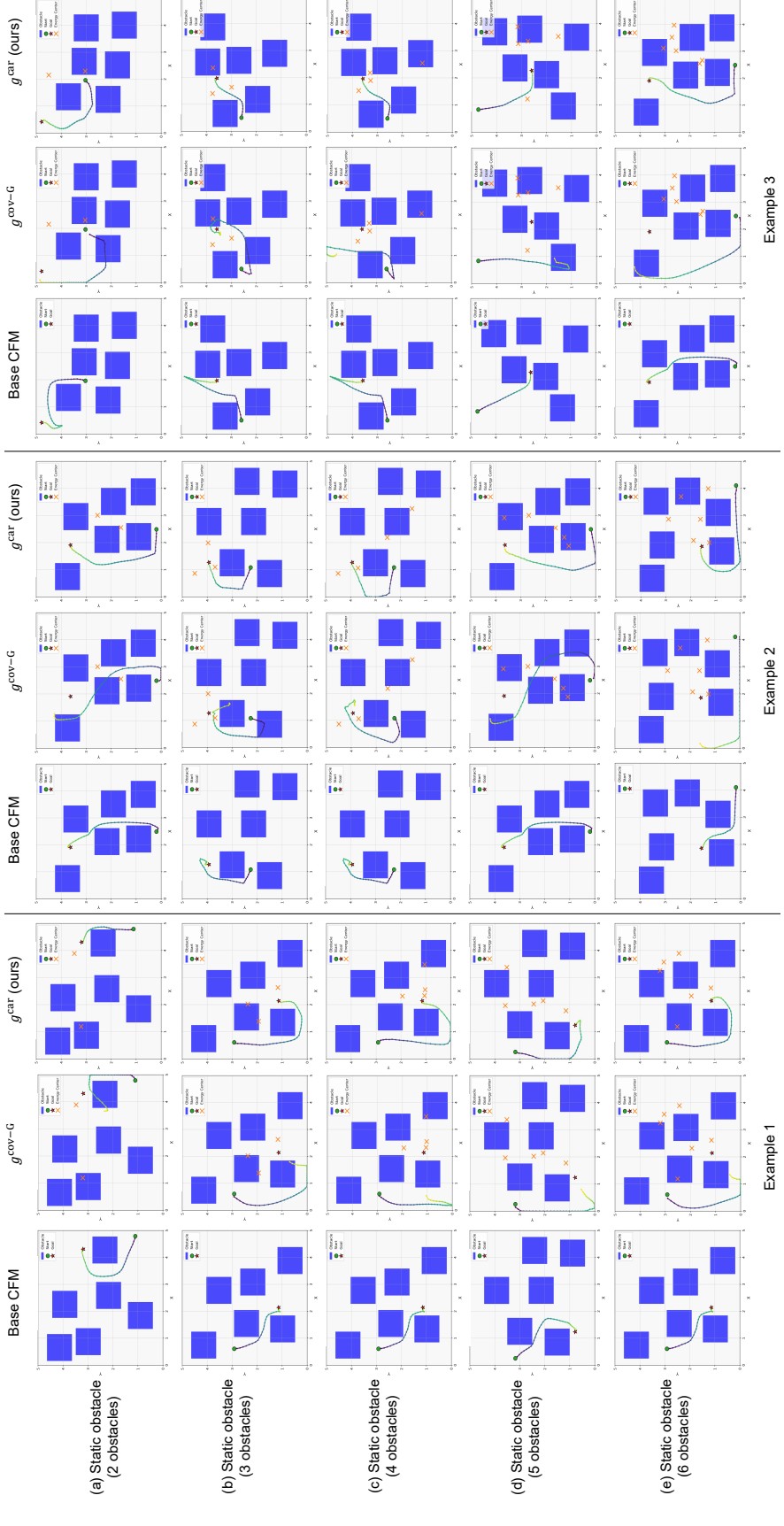

*Figure 17.* **Visual comparison of guided generation under increasing environmental complexity.** We scale the number of static obstacles to evaluate the solver's ability to handle dense constraints. As shown, traditional planning baselines like MPPI struggle with high-dimensional constraint landscapes, often failing to find feasible paths. In contrast, $g^{car}$ effectively navigates through dense clutter, generating smooth, collision-free trajectories that match the quality of those in simpler environments, highlighting its superior constraint-satisfaction capabilities.

## E.6. Generative decision-making as policies

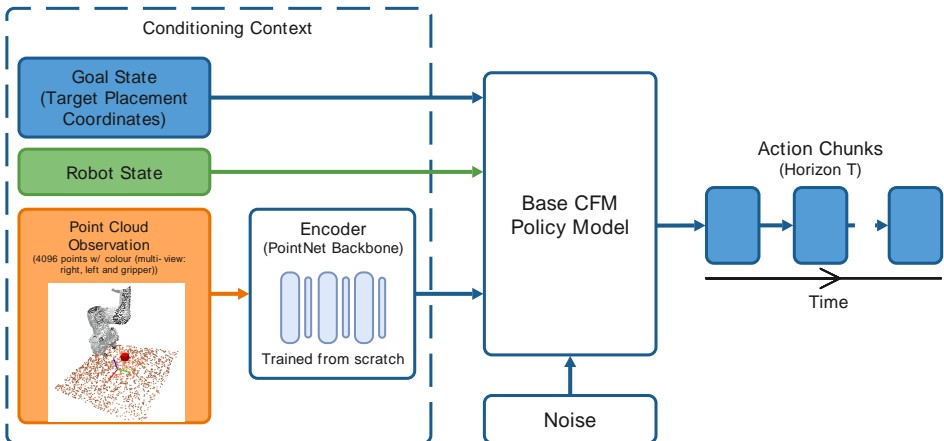

*Figure 18.* **Architecture of the Base CFM Policy.** The conditioning context includes the goal state (e.g., target placement coordinates), the point cloud observation, and the robot state. The observation (4096 colored points) is compressed via an encoder using a PointNet backbone trained from scratch. The model outputs action chunks of horizon $T$ generated from noise.

### E.6.1. BASE CFM POLICY

We implement a base Conditional Flow Matching (CFM) policy by adapting the PointFlowMatch architecture (Chisari et al., 2024) for goal-conditioned manipulation. Specifically, we incorporate explicit goal conditioning, and improve success rates. The detailed architecture is illustrated in Figure 18.

**Conditioning.** The policy is conditioned on a multimodal context vector $c$, constructed as follows:

1. Observation: Raw 3D point clouds ($N = 4096$) with RGB features are fused from multi-view cameras (left, right, and gripper). These are processed by a PointNet backbone to extract a dense feature vector.

2. Proprio State: A vector containing the robot's joint angles and gripper status.

3. Goal: The 3D coordinates representing the target placement location (e.g., the stacking position).

These components are concatenated to form the conditioning $c$.

**Output.** The model predicts action chunks of horizon $T$. The generative component is a Conditional 1D U-Net that predicts the time-dependent velocity field $v_\theta(x_t, t \mid c)$. Here, the flow state $x_t \in \mathbb{R}^{T \times 7}$ represents the flattened action chunk sequence (translation, rotation, and gripper action). Trajectories are generated by integrating the learned ODE from a standard Gaussian distribution at $t = 0$ to the target action distribution at $t = 1$.

**Training Objective.** Given an expert action chunk $\mathbf{A}_{\text{gt}} \in \mathbb{R}^{T \times 7}$ (denoted as $x_1$) and a random initial sample $x_0 \sim \mathcal{N}(0, I)$, we sample $t \sim \mathcal{U}(0, 1)$ and interpolate $x_t = (1-t)x_0 + tx_1$. The model is trained to regress the target velocity $v^{\text{gt}} = x_1 - x_0$ via mean-squared error.

**Dataset and evaluation.** For each task (PickCube and StackCube), we collect 100 expert demonstrations to train the base CFM model. The trained policy achieves 100% success rate on both training and test sets (100 episodes with unseen random seeds), confirming strong generalization. The base CFM policy is lightweight yet sufficient to complete the manipulation tasks without constraints. Our focus is on whether inference-time guidance can satisfy runtime constraints while preserving the base flow prior and staying on the data manifold.

*Table 8.* Comparison on ManiSkill2 StackCube and PickCub tasks. Compositional reward settings: (1) static obstacle: two random static obstacles; (2) hybrid composition: two random static obstacles and trajectory smoothness. Metrics include Inference Time (ms/sample), Violation (mean constraint violations #), Success (success rate %), and Steps (#). Results are averaged over 100 samples with conflict threshold $\tau = 0.20$. Note that we do not use inpainting, which allows us to better observe the capability of inference-time alignment methods in preserving the base model prior. For GLASS-FKS, we use $K = 8$ particles with a convergence coefficient $\rho = 0.95$, involving 24 internal steps per inference. The $g^{\text{car}}$ method requires an online training period of $20.4 \pm 0.4$ min for 8 training steps prior to inference.

| | | StackCube | | | | | | | PickCube | | | | | |
| | | (1) static obstacles | | | (2) hybrid composition | | | | (1) static obstacles | | | (2) static goal | | |
| Method | Time↓ | Viol.↓ | Succ.↑ | Steps↓ | Viol.↓ | Succ.↑ | Steps↓ | Time↓ | Viol.↓ | Succ.↑ | Steps↓ | Viol.↓ | Succ.↑ | Steps↓ |
|---|---|---|---|---|---|---|---|---|---|---|---|---|---|---|
| GLASS-FKS | 493.7 | 0.6 | 32 | 24 | 1.1 | 24 | 24 | 485.2 | 0.2 | 90 | 24 | 0.5 | 82 | 24 |
| | ±29.2 | ±0.4 | ±9.8 | | ±0.7 | ±12.5 | | ±45.2 | ±0.2 | ±5.2 | | ±0.3 | ±6.4 | |
| $g^{\text{cov-G}}$ | 185.0 | 1.2 | 12 | – | 1.8 | 9 | – | 180.0 | 0.4 | 58 | – | 0.8 | 46 | – |
| | ±9.3 | ±0.5 | ±4.1 | | ±0.8 | ±3.2 | | ±6.4 | ±0.1 | ±4.2 | | ±0.2 | ±3.5 | |
| PCGrad | 201.4 | $1.5^{\downarrow 0.3}$ | $0^{\downarrow 12}$ | – | $2.2^{\downarrow 0.4}$ | $0^{\downarrow 9}$ | – | 194.8 | $0.8^{\downarrow 0.4}$ | $10^{\downarrow 48}$ | – | $1.3^{\downarrow 0.5}$ | $7^{\downarrow 39}$ | – |
| | ±12.5 | ±0.6 | ±0.0 | | ±0.9 | ±0.0 | | ±11.2 | ±0.3 | ±2.4 | | ±0.4 | ±1.8 | |
| $g^{\text{car}}$ (ours) | 203.2 | $\mathbf{0.1}^{\uparrow 1.1}$ | $\mathbf{72}^{\uparrow 60}$ | 8 | $\mathbf{0.4}^{\uparrow 1.4}$ | $\mathbf{61}^{\uparrow 52}$ | 8 | 205.6 | $\mathbf{0.0}^{\uparrow 0.4}$ | $\mathbf{100}^{\uparrow 42}$ | 8 | $\mathbf{0.1}^{\uparrow 0.7}$ | $\mathbf{94}^{\uparrow 48}$ | 8 |
| | ±8.2 | ±0.1 | ±8.2 | | ±0.2 | ±9.6 | | ±10.8 | ±0.0 | ±0.0 | | ±0.1 | ±2.4 | |

*Note:* **Bold text** indicates the best performance. Rows with gray backgrounds indicate methods that utilize our $g^{\text{car}}$ for conflict correction. Purple superscripts show the performance change of $g^{\text{car}}$ over $g^{\text{cov-G}}$, and teal superscripts show the change of PCGrad over $g^{\text{cov-G}}$, where ↑ denotes improvement and ↓ denotes degradation. For all metrics except Steps, we report the mean (top row) and standard deviation across 5 random seeds (bottom row).

### E.6.2. EXPERIMENTAL RESULTS

Table 8 presents the comparative results under constrained settings. In the challenging StackCube task, $g^{\text{car}}$ reduces the violation rate from 1.2 to 0.1 (static obstacles) and from 1.8 to 0.4 (hybrid composition), while boosting the success rate from 12% to 72% and from 9% to 61% respectively. In the PickCube task, $g^{\text{car}}$ achieves perfect safety (0.0 violations) in static environments and boosts the success rate from 46% to 94% in the static goal setting. Notably, PCGrad degrades performance relative to $g^{\text{cov-G}}$ across both tasks (e.g., StackCube success drops to 0%), confirming that gradient surgery cannot handle high-precision manipulation tasks under compositional constraints. $g^{\text{car}}$ achieves these gains efficiently, consistently converging in just 8 steps.

Key observations are:

1. Adding inference-time guidance to pre-trained generative policy models is prone to OOD, and often fails to finish tasks, e.g., the failure shown in Figure 19 of $g^{\text{cov-G}}$.

2. PCGrad cannot recover from off-manifold drift.

3. GLASS-FKS generally performs well, but struggles in high-precision tasks such as StackCube (i.e., stably and precisely placing one cube onto another), due to its high transition variance. On tasks such as conditional generation (e.g., decision-making tasks), as long as the condition often appears in the dataset, GLASS-FKS performs well because it is easier to obtain an accurate estimation of $g_t$.

4. $g^{\text{car}}$ corrects off-manifold drift (success rate ↑) and shows decreased violation rate.

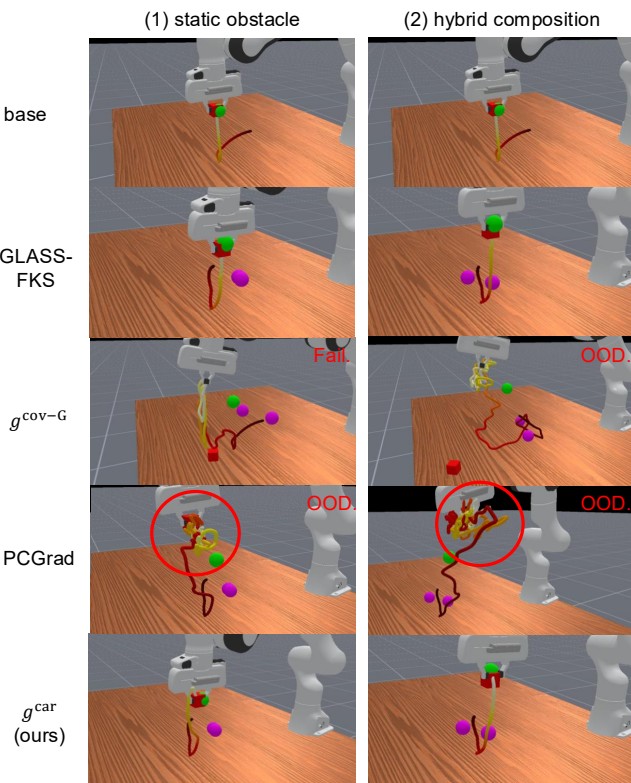

*Figure 19.* Visualization on ManiSkill2 PickCube task with conflict threshold $\tau = 0.20$. OOD: the trajectory leaves the data manifold, producing physically incoherent motions (e.g., erratic spinning or tangled paths); Fail:the trajectory stays on the manifold but fails the task (e.g., does not reach the goal).

### E.7. Text-guided image manipulation

#### E.7.1. EXPERIMENTAL DETAILS

To ensure fair comparisons, all text-guided image manipulation experiments, including the training of the online guidance network and the inference latency measurements, were conducted on a dedicated local workstation. The hardware specifications include an AMD EPYC 7543 Processor and a single NVIDIA RTX A5000 GPU (24GB VRAM). All algorithms and neural network architectures were implemented using the PyTorch framework with CUDA acceleration.

*Table 9.* Hyperparameter of $g^{car}$ in image editing.

| Hyperparameter | Value |
|---|---|
| Euler discretization steps (training) | $N = 10$ |
| Euler discretization steps (inference) | $N = 100$ |
| Guidance training Steps | $M = 30$ |
| Cutouts | 150 |
| Guidance scale | 30 |
| Batch size | 30 |

#### E.7.2. INFERENCE-TIME CONSTRAINTS

In our text-to-image generation experiment, we adopted the pipeline presented in Liu et al. (2023b), utilizing the generative prior from Liu et al. (2023a). The terminal reward function is:

$$r(x_1) = \text{CLIP}(x_1, T), \tag{51}$$

Baseline configurations were aligned with those reported in Liu et al. (2023b), and the complete results presented in Table 10 reflect the same experimental conditions. For quantitative comparison, we used the CelebA-HQ dataset, randomly sampling 1,000 images, which were manipulated based on standard single text guidance (i.e., *sad*, *angry*, *happy*, *smiling*, *curly hair*) and composed text guidance (i.e., *sad + angry*, *sad + happy*, *sad + curly hair*).

#### E.7.3. EVALUATION METRIC

We evaluate our method using six quantitative metrics across two categories:

**Text-Image Alignment:** We first use (1) **CLIP** (Higher is better), which measures basic text-image alignment by calculating image and text embeddings separately and measuring the distance between them. Because fine-grained misalignments are often left undetected by standard multi-modal models like CLIP (Singh & Zheng, 2023), we also report (2) **BLIP-ITM** (Li et al., 2022) (Higher is better). BLIP-ITM utilizes cross-attention between a ViT image encoder and a BERT-base text processor to act as a strict binary classifier, predicting whether an image and prompt are an exact match. Finally, we use (3) **VQAScore** (Lin et al., 2024) (Higher is better) to evaluate complex compositional reasoning by reframing image evaluation as a visual question answering task using LLaVA-1.5.

**Image Quality and Preservation:** To evaluate visual fidelity, we use (1) **CLIP-IQA** (Wang et al., 2023) (Higher is better) to assess intrinsic visual quality and penalize blurry or artifact-heavy generations. To evaluate how well the original inputs are maintained, we report (2) **LPIPS** (Lower is better) for the preservation of overall image content, and (3) **ID** (Higher is better) for the preservation of subject identity.

#### E.7.4. EXPERIMENTAL RESULTS

Table 10 presents a detailed quantitative comparison. We observe that compositional constraints significantly increase the difficulty of maintaining manifold adherence across all baselines, as conflicting objectives lead to higher LPIPS and lower ID scores. FlowGrad achieves LPIPS of 0.203 and ID of 0.677 under composed prompts, and struggles to balance multiple objectives simultaneously (i.e., the uneven text-image alignment between $P_1$ and $P_2$). $g^{car}$ outperforms $g^{cov-G}$ by a large margin in identity preservation (0.681 vs. 0.543), proving its ability to rectify off-manifold drift where approximate guidance fails. Furthermore, the large gap between $P_1$ and $P_2$ scores for PCGrad (BLIP-ITM: 0.650 vs. 0.337; VQAScore: 0.785 vs. 0.371) means that gradient surgery fails to balance multiple constraints, whereas $g^{car}$ achieves consistent alignment across both prompts. Visualization results are provided in Figure 20.

*Table 10.* Comparison of methods on image quality metrics (LPIPS, CLIP-IQA, and ID), text-image alignment metrics (CLIP, BLIP-ITM, and VQAScore), and computational efficiency for text-guided face manipulation on CelebA-HQ. To demonstrate the imbalance issue in multi-objective optimization (i.e., optimizing for two prompts simultaneously), we report the text-image alignment metrics separately for the first prompt ($P_1$), the second prompt ($P_2$), and their average (Avg). A significant discrepancy between $P_1$ and $P_2$ indicates a severe optimization imbalance. We report results separately for *composed text guidance* (i.e., *sad + angry, sad + happy, sad + curly hair*).

| | Image Quality | | | Text-Image Alignment | | | | | | | | | Efficiency | |
| | | | | CLIP ↑ | | | BLIP-ITM ↑ | | | VQAScore ↑ | | | | |
| Method | LPIPS ↓ | ID ↑ | CLIP-IQA ↑ | $P_1$ | $P_2$ | Avg | $P_1$ | $P_2$ | Avg | $P_1$ | $P_2$ | Avg | Train | Infer. |
|---|---|---|---|---|---|---|---|---|---|---|---|---|---|---|
| FlowGrad | **0.203** ±0.004 | 0.677 ±0.008 | 0.495 ±0.005 | 0.277 ±0.002 | 0.272 ±0.003 | 0.274 ±0.002 | 0.405 ±0.012 | 0.246 ±0.009 | 0.326 ±0.010 | 0.768 ±0.015 | 0.424 ±0.011 | 0.596 ±0.012 | – | 124.5 s ±5.8 |
| GLASS-FKS | 0.313 ±0.018 | 0.329 ±0.025 | 0.586 ±0.019 | 0.255 ±0.012 | 0.252 ±0.014 | 0.253 ±0.013 | 0.127 ±0.022 | 0.037 ±0.018 | 0.082 ±0.020 | 0.393 ±0.031 | 0.229 ±0.028 | 0.311 ±0.029 | – | 635.2 s ±58.4 |
| $g^{\text{cov-G}}$ | 0.217 ±0.005 | 0.543 ±0.010 | 0.535 ±0.006 | 0.288 ±0.002 | 0.293 ±0.002 | 0.290 ±0.002 | 0.521 ±0.014 | 0.528 ±0.013 | 0.525 ±0.014 | 0.748 ±0.016 | 0.753 ±0.015 | 0.751 ±0.014 | – | 11.08 s ±0.14 |
| PCGrad | $0.219^{\downarrow0.002}$ ±0.004 | $0.602^{\uparrow0.059}$ ±0.011 | $0.535^{\uparrow0.000}$ ±0.005 | $0.285^{\downarrow0.003}$ ±0.002 | $0.266^{\downarrow0.027}$ ±0.003 | $0.276^{\downarrow0.014}$ ±0.002 | $\mathbf{0.650}^{\uparrow0.129}$ ±0.010 | $0.337^{\downarrow0.191}$ ±0.010 | $0.444^{\downarrow0.081}$ ±0.013 | $\mathbf{0.785}^{\uparrow0.037}$ ±0.018 | $0.371^{\downarrow0.382}$ ±0.012 | $0.578^{\downarrow0.173}$ ±0.015 | – | 13.45 s ±0.38 |
| $g^{\text{car}}$ (ours) | $0.226^{\downarrow0.009}$ ±0.004 | $\mathbf{0.681}^{\uparrow0.138}$ ±0.010 | $\mathbf{0.543}^{\uparrow0.008}$ ±0.006 | $\mathbf{0.290}^{\uparrow0.002}$ ±0.002 | $\mathbf{0.293}^{\uparrow0.000}$ ±0.002 | $\mathbf{0.291}^{\uparrow0.001}$ ±0.002 | $0.592^{\uparrow0.071}$ ±0.013 | $\mathbf{0.602}^{\uparrow0.074}$ ±0.014 | $\mathbf{0.597}^{\uparrow0.072}$ ±0.012 | $0.751^{\uparrow0.003}$ ±0.015 | $\mathbf{0.758}^{\uparrow0.005}$ ±0.014 | $\mathbf{0.755}^{\uparrow0.004}$ ±0.013 | 20.4 m ±0.6 | 11.26 s ±0.12 |

*Note:* **Bold text** indicates the best performance. Rows with gray backgrounds indicate methods that use our $g^{\text{car}}$ for conflict correction. Purple superscripts show the performance change of $g^{\text{car}}$ over $g^{\text{cov-G}}$, and teal superscripts show the change of PCGrad over $g^{\text{cov-G}}$, where ↑ denotes improvement and ↓ denotes degradation. For all metrics, we report the mean (top row) and standard deviation (bottom row) across 5 random seeds.

Overall, the key observations are:

- $\mathbf{g^{cov-G}}$ is prone to off-manifold drift and has hallucinated generation; also it is too sensitive to the guidance scale.

- **FlowGrad** fails to balance multiple constraints.

- **PCGrad** attempts to resolve conflicts via gradient surgery but fails to balance multiple constraints, leaving some targets unfulfilled (e.g., failing to generate an "angry" expression). Furthermore, it cannot recover from off-manifold drift.

- **GLASS-FKS** fails to preserve the reference image. This is largely due to the high variance of sampling (i.e., ODE-based transition sampling) given a limited number of particles. Specifically, estimating $g_t$ requires samples from regions where $e^r$ is significantly higher than average, i.e., images already closely resembling the reference, which is unlikely to be achieved with a limited particle budget.

- **Our $\mathbf{g^{car}}$** achieves superior compositional reward alignment across multiple prompts, corrects off-manifold drift, and eliminates the hallucinated visual artifacts observed in $g^{\text{cov-G}}$.

**More about GLASS-FKS** In text-guided image manipulation, GLASS-FKS fails to preserve the reference image. This is largely due to the high variance of sampling (i.e., ODE-based transition sampling) given a limited number of particles. Specifically, estimating $g_t$ requires samples from regions where $e^r$ is significantly higher than average, i.e., images already closely resembling the reference, which is unlikely to be achieved with a limited particle budget. This failure mode is similar to Monte Carlo guidance in Feng et al. (2025), where more advanced sampling techniques help GLASS-FKS preserve more prior than Monte Carlo guidance but do not fully resolve the issue. We also note that GLASS-FKS's original evaluation uses a stronger base model (FLUX) and a richer reward composition (CLIP, Pick, HPSv2, ImageReward), whereas our setting uses a Rectified Flow trained on CelebA-HQ with CLIP score as the sole reward. **Richer reward composition likely provides more informative evaluation for particle steering**, which helps explain the strong performance reported in the original paper.

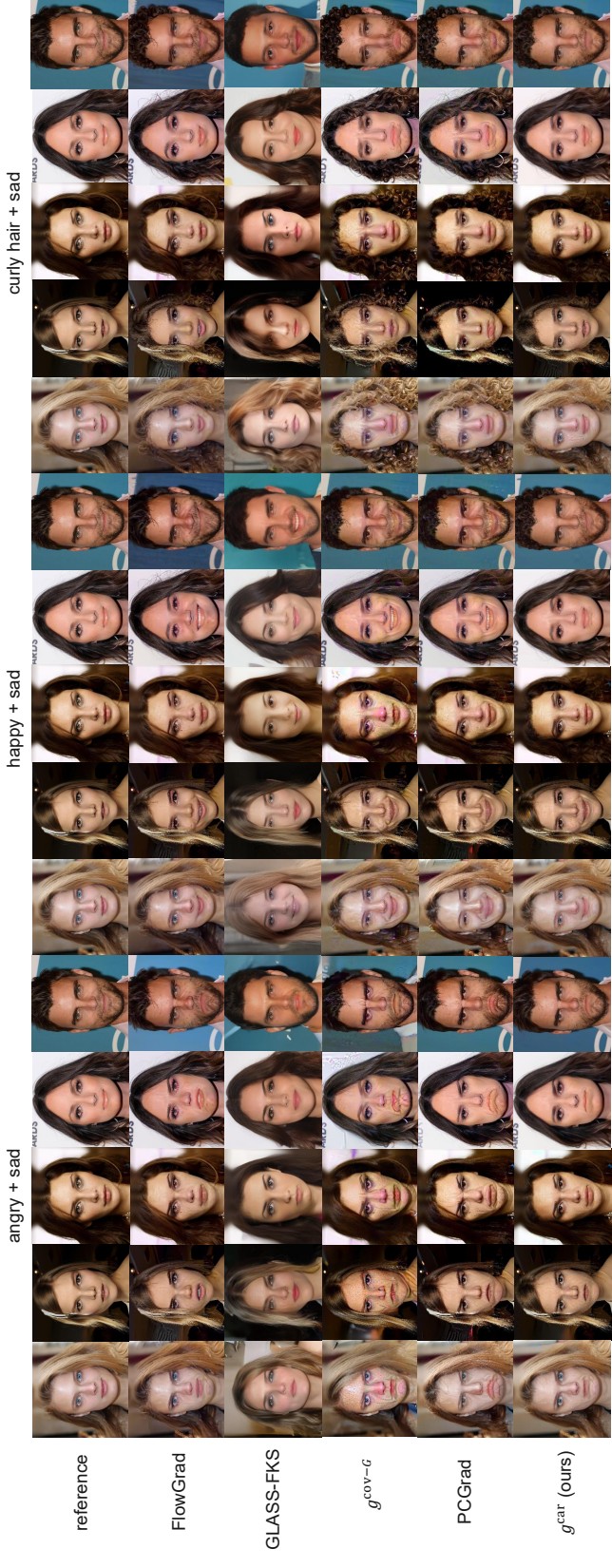

*Figure 20.* **Additional visualization of text-guided image manipulation.** This figure complements Figure 6 by showing further results of $g^{car}$ on the CelebA-HQ dataset under various composed text prompts.

