# OpenReview forum: "Conflict-Aware Additive Guidance for  Flow Models under Compositional Rewards"
_ICML.cc/2026/Conference — ICML 2026 regular_

### Official Review · Reviewer_ke1m · 2026-03-08

**Soundness:** 2
**Presentation:** 1
**Significance:** 3
**Originality:** 3
**Overall Recommendation:** 4
**Confidence:** 3

**Summary:**

The authors propose a new inference-time guided sampling method designed to work better in compositional reward settings. The core motivation is that, in these settings, especially when rewards conflict, misaligned gradients can cause off-manifold drift. To address this, they introduce Conflict-Aware Additive Guidance (gcar), which combines approximate guidance with a corrective guidance term controlled by a conflict score computed from pairwise cosine similarities among reward gradients. The authors evaluate the method on synthetic data, generative decision-making tasks, and text-guided image manipulation, and report good to mixed results across these domains.

**Compliance With Llm Reviewing Policy:**

Affirmed.

**Final Justification:**

My main concerns were two, as addressed in my initial review: first, the moderate empirical gains; second, paper quality issues such as the duplicated references. The former has been fully addressed by the rebuttal and the following round of discussion. Regarding the latter, the authors seem to have acknowledged the quality issues and have committed to fix them. My remaining concern is the discussion around cosine similarity being rigorously derived from the theorem and fully encapsulating all failure modes. Theorem 4.2 shows that it is part of the approximation error under gradient misalignment, not that it is fully representative of it. While I encourage the authors to soften their claims regarding this last point, I do not think it is a reason to reject the paper. Therefore, I raise my score to "Weak Accept."

**Key Questions For Authors:**

1. Please address the points in "Strength and Weaknesses"
2. Which matters more, the learned correction or the conflict-aware gate?
3. Does gcar need to be retrained for each new reward composition, or can it generalize?
4. Could the authors discuss the increase in compute time needed to train gcar compared to the training-free baselines?

**Limitations:**

yes

**Strengths And Weaknesses:**

**Strengths:
1. The problem tackled is important and has significant real-world implications. The paper correctly identifies a weakness of current inference-time guidance methods.
2. The link between reward conflict, gradient conflict, and off-manifold drift is clearly described.
3. The method is evaluated on a wide range of synthetic and real-world experiments.
4. The results on the synthetic benchmark are promising.

**Weaknesses:
1. While the results on the synthetic dataset are promising, the gains on the real-world tasks are more mixed and not always strong.
2. Using pairwise cosine similarity between gradients captures only a subset of possible failure modes, such as poor reward calibration or higher-order effects. The paper does not provide enough theoretical or empirical evidence that this metric is broadly sufficient.
3. Most tables do not report variance, which makes it harder to assess the robustness of the gains.
4. There are severe paper quality issues, including several duplicated references

---

> ### Author Rebuttal · Authors · 2026-03-30
>
> Thank you very much for your feedback. We are also grateful for the reviewer's recognition of the novelty, contributions and presentation of our work. We hope our clarifications and updated results will convince the reviewer to reconsider their score. Please let us know if further clarification is required.
>
> ---
>
> ## R1 to W1: Gains on the Real-World Tasks
>
> Thank you for this comment. We have strengthened the real-world tasks in two ways: (1) stronger perceptual evaluation metrics for image editing (please refer to "R(2) to W(2): Image Evaluation Metrics" in our response to Reviewer sXnK); (2) expanded baselines spanning the full spectrum of inference-time guidance methods, including GLASS-FKS (please refer to "R1 to Q2 & W3: Baseline Overview and GLASS-FKS" in our response to Reviewer vjZE). The full experimental results are available at [this link](https://anonymous.4open.science/r/Conflict-Aware-Additive-Guidance-for-Flow-Models-under-Compositional-Rewards-B484/rebuttal_images_tables.pdf).
>
> ---
>
> ## R2 to W2: Cosine Similarity
>
> We clarify that cosine similarity is not a heuristic but a theorem-driven design choice. Two key theoretical takeaways underpin our method: (1) in compositional reward settings, gradient misalignment is a root cause of off-manifold drift (Theorem 4.2); (2) detecting gradient misalignment $\approx$ detecting off-manifold drift, so we only need to learn the exact correction in conflict regions, minimizing compute. Cosine similarity directly operationalizes this: regions of low cosine similarity correspond precisely to regions where approximate guidance induces off-manifold drift. The weight $w_t$ determines *when and where* to correct, and $g_\psi$ provides the correction by directly regressing to the reward signal. So we believe our design is fully theorem-driven.
>
> Empirically, we compare with PCGrad to show that $g^\text{car}$ is sufficient for correcting off-manifold drift:
>
> 1. Image editing ([Figure](https://anonymous.4open.science/r/Conflict-Aware-Additive-Guidance-for-Flow-Models-under-Compositional-Rewards-B484/Image_CelebaHQ.jpg)): our failure mode analysis visualizes per-pixel conflict scores, showing that $g^{\text{car}}$ accurately captures conflict regions in semantically meaningful facial structures (e.g., the lip region for "happy + sad"), and corrects off-manifold drift in those regions. PCGrad, by contrast, fails to balance multiple constraints (e.g., failing to generate a clear "angry" face in "sad + angry" task) and cannot recover from off-manifold drift.
> 2. Robot planning ([Figure](https://anonymous.4open.science/r/Conflict-Aware-Additive-Guidance-for-Flow-Models-under-Compositional-Rewards-B484/Image_maze2d.jpg)): without inpainting, $g^{\text{car}}$ achieves better prior preservation than baselines, i.e., trajectories stay on the data manifold and reach end points that the base model could originally reach.
> 3. Robot manipulation ([Figure](https://anonymous.4open.science/r/Conflict-Aware-Additive-Guidance-for-Flow-Models-under-Compositional-Rewards-B484/Image_maniskilll.jpg)): inference-time guidance often induces OOD behavior (i.e., physically incoherent motions such as erratic spinning); $g^{\text{car}}$ effectively corrects this off-manifold drift across all manipulation settings.
>
> Across all three domains, $g^{\text{car}}$ consistently corrects the off-manifold drift that baselines fail to address, providing strong empirical evidence that cosine similarity is broadly sufficient as a conflict detector.
>
> ---
>
> ## R3 to W3: Variance
>
> Thank you for this comment. We've added variance across 5 random seeds to all tables. Full results at [this link](https://anonymous.4open.science/r/Conflict-Aware-Additive-Guidance-for-Flow-Models-under-Compositional-Rewards-B484/rebuttal_images_tables.pdf). Variance analysis reveals:
>
> - GLASS-FKS (sample-based) has the highest variance overall.
> - $g^{\text{car}}$ achieves variance comparable to the training-free baseline $g^{\text{cov-G}}$, the training process is stable.
>
> ---
>
> ## R4 to W4: Duplicated References
>
> Thank you, we've fixed it in the revised manuscript.
>
> ---
>
> ## R5 to Q2: Ablation of the Conflict-Aware Weight $w_t$
>
> Thank you, please refer to "R4 to Q1: Ablation of the Conflict-Aware Weight $w_t$" in our response to Reviewer pD4Z for a detailed analysis.
>
> ---
>
> ## R6 to Q3 & Q4: Generalisation and Compute
>
> We acknowledge the computational overhead of training $g_\psi$; however, once trained, it generalizes across all samples within a given compositional reward setting. In contrast, training-free approximations must be re-optimized per sample and per reward composition, which may offset their apparent efficiency advantage. Exploring lighter alternatives remains an interesting direction for future work.
>
> We provide inference and training time for all tasks at [this link](https://anonymous.4open.science/r/Conflict-Aware-Additive-Guidance-for-Flow-Models-under-Compositional-Rewards-B484/rebuttal_images_tables.pdf).

---

> > ### Author Rebuttal · Reviewer_ke1m · 2026-04-03
> >
> > Thank you for your response. Most of my concerns have been adequately addressed: I will raise my score.

---

> > > ### Author Response · Authors · 2026-04-05
> > >
> > > We are glad to hear that the reviewer's concerns have been resolved. Thank you for recognizing our efforts and for raising the score. We appreciate the valuable feedback and the time spent improving our work.

---

### Official Review · Reviewer_vjZE · 2026-03-13

**Soundness:** 3
**Presentation:** 3
**Significance:** 3
**Originality:** 3
**Overall Recommendation:** 4
**Confidence:** 3

**Summary:**

This paper studies compositional guidance for flow models under multiple rewards and argues that standard additive score guidance can fail when the reward gradients conflict, causing off-manifold drift and larger approximation error; to address this, it proposes CAR guidance, which measures gradient conflict using pairwise cosine similarity and interpolates between an approximate guidance term and a learned value-gradient correction term trained by terminal value regression.

**Compliance With Llm Reviewing Policy:**

Affirmed.

**Key Questions For Authors:**

Can the authors show that the proposed updates in Equations (13) and (14) do not introduce bias into the sampling process, and that they indeed pushforward the initial noisy distribution toward the intended target distribution?

Why is there no discussion or comparison with Feynman–Kac Correctors, which explicitly target annealed, geometric-averaged, or product distributions in a principled way rather than repairing heuristic score mixing afterward? Even if a full apples-to-apples benchmark is difficult, this seems highly relevant for positioning the contribution.

The paper argues that TVR avoids the instability of bootstrapped fitted value evaluation, but the final method still depends on learning a scalar value network whose gradient is used directly for guidance. How stable is this in practice across seeds, architectures, and reward compositions, and are there failure cases where gradient of the value function becomes noisy or misleading?

**Limitations:**

yes

**Strengths And Weaknesses:**

Strengths.
1.The paper is clearly written and straightforward to follow.
2.The error bound in Theorem 4.2 is one of the strongest parts of the work: it is both technically meaningful and genuinely insightful. More broadly, the analysis in Section 4 is careful, coherent, and theoretically well grounded.

Weakness:
1. The main algorithmic design appears less principled than the theory. In particular, the use of cosine similarity to determine the weighting coefficients is heuristic and is not directly derived from the theoretical results, so the method feels more theorem-motivated than theorem-driven.
2. The practical reliance on a learned value function and its gradient may introduce substantial instability and computational overhead, which raises concerns about scalability; lighter alternatives would be worth exploring.
3. The paper should better position itself relative to prior work on gradient-conflict-free sampling of product targets, especially the Feynman–Kac Corrector line of work. That framework provides a more principled way to target the product distribution directly, thereby avoiding the gradient conflict issue at its source. While the original FKC formulation is developed for SDEs, later works such as GLASS Flow extend related ideas to the ODE setting and should be discussed. Although the present paper avoids the Sequential Monte Carlo requirement in FKC, its own value-function fitting procedure may in practice make training and inference even more difficult.

---

> ### Author Rebuttal · Authors · 2026-03-30
>
> Thank you for your recognition of our work. We appreciate your acknowledgement of our presentation and theoretical contribution, and especially your positive feedback on Section 4, it is very helpful to us. We also greatly appreciate the reviewer's suggestion to compare against GLASS Flows with Feynman-Kac Steering (GLASS-FKS, Holderrieth et al., 2026), which has enabled a more thorough comparison spanning the full spectrum of inference-time guidance methods, and we found the results particularly insightful. We hope our clarifications and updated results will convince the reviewer to reconsider their score. Please let us know if further clarification is required.
>
> ---
>
> ## R1 to Q2 & W3: Baseline Overview and GLASS-FKS
>
> We thank the reviewer for pointing out GLASS-FKS. Since GLASS Flows has not yet released an open-source implementation, we reproduced it based on the minimal implementation for low-dimensional data provided by the authors.
>
> ### Baseline Overview
>
> Our baselines now span the full spectrum of inference-time guidance methods:
> - (1) Approximate guidance: $g^\text{cov-G}$
> - (2) Exact guidance: GLASS-FKS (sample-based, Holderrieth et al., 2026) and Guidance Matching (GM, training-based, Feng et al., 2025)
> This work ($g^\text{car}$), which sits in the middle, aims to improve the compute-light approximate guidance by adding a fraction of extra compute ([A visual overview](https://anonymous.4open.science/r/Conflict-Aware-Additive-Guidance-for-Flow-Models-under-Compositional-Rewards-B484/inference-time_guidance.png)).
>
> We also include PCGrad (Yu et al., 2020), a gradient conflict resolution method in multi-objective optimisation, to show that our conflict-aware mechanism corrects off-manifold drift more effectively. Task-specific SOTA baselines are included where applicable (e.g., MPPI for robot planning, with MPPI + $g^\text{car}$ showing that our learnable guidance corrects the off-manifold drift of MPPI-generated paths).
>
> Key observations across methods:
> 1. $g^{\text{cov-G}}$ is prone to off-manifold drift and overly sensitive to guidance scale.
> 2. PCGrad fails to balance multiple constraints and cannot recover from off-manifold drift.
> 3. GM requires ground-truth samples satisfying all constraints, limiting applicability, and suffers from confounding errors.
> 4. GLASS-FKS shows high variance particularly in high-dimensional tasks (e.g., failing to preserve reference images in image editing). On tasks such as conditional generation (e.g., decision-making tasks), as long as the condition often appears in the dataset, GLASS-FKS performs well because it is easier to obtain an accurate estimation of $g_t$; but struggles in high-precision tasks such as StackCube (i.e., struggles to stably and precisely place one cube onto another).
> 5. $g^{\text{car}}$ achieves superior compositional alignment across all tasks, corrects off-manifold drift, and remains compute-light relative to GLASS-FKS and GM.
>
> ---
>
> ## R2 to W1: Cosine Similarity
>
> Thank you, please refer to "R2 to W2: Cosine similarity" in our response to Reviewer ke1m for a detailed clarification.
>
> ---
>
> ## R3 to W2: Lighter Alternatives
>
> Thank you, please refer to "R6 to Q3 & Q4: Generalisation and Compute" in our response to Reviewer ke1m for a detailed discussion.
>
> ---
>
> ## R4 to Q3: Instability of Bootstrap
>
> We clarify that the core difference lies in the regression target. Fitted Value Evaluation (FVE) regresses $\hat{V}(x_t, t)$ against $r + \gamma \hat{V}_k(x', t')$, where $\hat{V}_k$ is the estimate from the previous iteration, so the target is itself moving, introducing instability. TVR instead directly regresses against the fixed terminal reward $r(x_1)$, which makes it more stable. We provide stability analysis across 5 random seeds; please refer to [this link](https://anonymous.4open.science/r/Conflict-Aware-Additive-Guidance-for-Flow-Models-under-Compositional-Rewards-B484/rebuttal_images_tables.pdf).
>
> ---
>
> ## R4 to Q1: Biased Sampling
>
> We clarify that our goal is to sample from the reward-tilted distribution $e^{r(x_1)}p^\text{base}(x_1)$ rather than perform biased sampling. Approximate guidance $g^\text{approx}$ inherently introduces bias; however, $g^\text{car}$ does not introduce additional bias — rather, it reduces bias in conflict regions by replacing the approximate guidance with the learned exact guidance $g_\psi$.
>
> To empirically verify this, we report Posterior Coverage (PC) on synthetic tasks where the ground-truth tilted distribution is known by construction. PC measures the fidelity of generated samples against the ground-truth target distribution, capturing both off-manifold drift and mode collapse. As shown in Table 1, $g^\text{car}$ achieves the highest average PC (92.37%) across all compositional settings, compared to $g^\text{cov-G}$ and GM.
>
> So $g^\text{car}$ indeed pushforwards the initial noisy distribution toward the intended target distribution.

---

> > ### Author Rebuttal · Reviewer_vjZE · 2026-04-05
> >
> > Thanks the authors for the detailed rebuttal. I shall keep my positive rating and recommend the authors to claim their assumptions more carefully

---

> > > ### Author Response · Authors · 2026-04-07
> > >
> > > We appreciate the reviewer’s valuable feedback and time, as well as their positive acknowledgement of our work. These insights have been very helpful, and we've carefully clarified our assumptions in the revised manuscript.

---

### Official Review · Reviewer_pD4Z · 2026-03-13

**Soundness:** 3
**Presentation:** 3
**Significance:** 2
**Originality:** 2
**Overall Recommendation:** 4
**Confidence:** 3

**Summary:**

The paper addresses inference-time steering of flow models under multiple compositional rewards. It identifies gradient misalignment as a key source of off-manifold drift in standard additive guidance, and proposes a hybrid method that mixes cheap approximate guidance with a learned value-gradient correction, activated only in conflict regions. The learned correction is trained from terminal rewards on guided rollouts. Experiments across synthetic benchmarks, manipulation, and image editing show improved  alignment than approximate baselines, with lower cost than exact guidance methods.

**Compliance With Llm Reviewing Policy:**

Affirmed.

**Final Justification:**

During the rebuttal, the author clarified that one most related work VGG-flow is a finetuning method and the proposed method is an inference-time approach.  However, no qualitative comparison is being done compared to VGG-flow.  The problem that the authors proposed is novel but there is no evidence that these finetuning methods like VGG-flow will not solve them which makes the work weak. I raised my score given the problem identification and the novelty of the inference time approach.

**Key Questions For Authors:**

**Questions**

- How stable is training of the learned guidance term in practice, given the number of interacting components, including the conflict-aware weighting term w(t)?
- Could the authors report VGG-Flow performance on the same tasks and compare it directly against the proposed method?
- Why is Guidance Matching not included in Tables 2 and 3? Since conditional examples appear to exist in these settings, it would be helpful for the authors to clarify whether GM is inapplicable here.
- The paper states that GM suffers from confounding errors. Could the authors explain more what is meant by this?

**Limitations:**

yes

**Strengths And Weaknesses:**

**Strengths**
- The paper highlights an important and practically relevant failure mode of inference-time guidance under compositional rewards.
- Error bound in theorem 4.2 provides useful insights on the multi-reward problem.
- The visualizations are helpful, and the method section is intuitive and well motivated. The overall presentation makes the core idea easy to follow.


**Weaknesses**

- The main novelty appears limited relative to prior work on learned value-guidance hybrids, especially VGG-Flow. In particular, the idea of combining an estimated/heuristic guidance term with a learned correction seems closely related, yet VGG-Flow is not included in the experiments as one of the baselines.
- Given this close relationship, the paper would benefit from an explicit discussion in the main text clarifying the distinction from VGG-Flow.
- Several strong baselines from the inverse problem literature also seem to be missing, i.e. Adjoint Matching.
- Some of the image outputs still contain noticeable artifacts, which weakens the empirical case for the method’s practical advantage.

---

> ### Author Rebuttal · Authors · 2026-03-30
>
> Thank you for your thoughtful feedback, especially the helpful comments on the conflict-aware weight $w(t)$ ablation study. We appreciate your recognition of our theoretical contributions, our motivation, and the clarity of our presentation. We have provided a more thorough ablation study to help readers better understand the contribution of each component, and an extended related work discussion to clarify our contributions relative to prior work. We hope our clarifications and analysis will convince the reviewer to reconsider their score. Please let us know if further clarification is required. Please kindly refer to [this link](https://anonymous.4open.science/r/Conflict-Aware-Additive-Guidance-for-Flow-Models-under-Compositional-Rewards-B484/rebuttal_images_tables.pdf) for all results.
>
> ---
>
> ## R1 to W1, W2, W3 & Q2: Distinction from VGG-Flow and Adjoint Matching
>
> Thank you for pointing out VGG-Flow and Adjoint Matching. We have carefully reviewed both papers, and we believe our work didn't contribute to fine-tune base flow models (i.e., VGG-Flow and Adjoint Matching). To help readers understand our contribution, i.e., correcting the off-manifold drift of inference-time guidance methods, we have added Appendix A: Extended Related Works to the revised manuscript.
>
> Our baselines span the full spectrum of inference-time guidance methods; please refer to "R1 to Q2 & W3: Baseline Overview and GLASS-FKS" in our response to Reviewer vjZE for a detailed baseline overview and discussion.
>
> ---
>
> ## R2 to W1: Combining a Heuristic Guidance Term with a Learned Correction
>
> Thank you for this thoughtful comment. Our work contributes on: we find and proved when and where will have off-manifold drift, and how to correct off-manifold drift of inference-time guidance methods, and we learn a residual correction $g_\psi$ to rectify the off-manifold drift incurred by any heuristic guidance.
>
> Our learned correction term is therefore fundamentally different from that in VGG-Flow: VGG-Flow identifies the gap between the pretrained base velocity and the fine-tuned velocity as the value gradient, and just co-trained a value gradient to align and help with fine-tune velocity; also they don't contribute on off-manifold correction.
>
> ---
>
> ## R3 to W4: Image Artifacts
>
> We've strengthened the perceptual evaluation with additional metrics for fairer text-image alignment and image quality assessment. These results demonstrate that our $g^\text{car}$ effectively corrects the off-manifold drift of $g^\text{cov-G}$. We also provided additional comparisons against FlowGrad, GLASS-FKS, and PCGrad. Please kindly refer to "R(2) to W(2): Image Evaluation Metrics" in our response to Reviewer sXnK for detailed discussion.
>
> ---
>
> ## R4 to Q1: Ablation of the Conflict-Aware Weight $w_t$
>
> Thank you for this comment. We provide a systematic ablation in Appendix F.3 studying the contribution of each component in $g^{\text{car}}(x_t, t) = (1 - w_t) g^{\text{approx}} + w_t g_\psi(x_t, t)$:
>
> | # | Guidance | $g^{\text{approx}}$ | $g_\psi$ | $w_t$ | Question to answer |
> |---|----------|:---:|:---:|:---:|---|
> | (1) | $g^{\text{cov-G}}$ | ✓ | ✗ | ✗ | |
> | (2) | $g^{\text{approx}} + g_\psi$ | ✓ | ✓ | ✗ | How much does the learned correction $g_\psi$ contribute? |
> | (3) | $g^{\text{car}}$ (ours) | ✓ | ✓ | ✓ | How much does the conflict-aware weight $w_t$ contribute? |
>
> The key observations are:
>
> - Adding $g_\psi$ without the gate shows modest gains, so, the learned correction helps.
> - Adding the conflict-aware gate $w_t$ gives larger improvements, showing that $w_t$ is the more critical component.
> - Training loss curves in Figure 14(c,f,i) confirm stable convergence across all settings.
>
> ---
>
> ## R5 to Q3 & Q4: Guidance Matching
>
> We thank the reviewer for this question; we've clarified those in the revised manuscript. Guidance Matching (GM) requires ground-truth samples $x_1$ that simultaneously satisfy all compositional reward constraints. In image editing and robotics tasks, the reward constraints are specified at inference time and were unseen during training; furthermore, no demonstrations satisfying the full set of compositional constraints are available. GM is therefore only applicable to our synthetic benchmarks, where ground-truth samples from the joint reward-tilted distribution are accessible by construction.
>
> Confounding errors refer to the failure mode caused by finite approximation capacity of neural networks. Specifically, GM has no approximation error in the loss target, i.e., it uses ground-truth $(x_0, x_1)$ pairs with exact $v_{t|z}$ supervision. The failure is purely one of capacity: $g^\text{GM}$ must fit a complex guidance landscape over all of $(x_t, t)$ space, and a finite-capacity network cannot do this perfectly everywhere. In contrast, with the same network capacity, $g^\text{car}$ concentrates $g_\psi$ within the conflict region, where the regression problem is simpler.

---

> > ### Author Rebuttal · Reviewer_pD4Z · 2026-04-04
> >
> > Thank you for your answers.  I would like to point out that for VGG-flow, even though it is a fine-tuning method, but the concept of using $g = \nabla V$ is very close conceptually to the proposed method. I would highly recommend the author to include in the main manuscript for discussing the relationship.

---

> > > ### Author Response · Authors · 2026-04-04
> > >
> > > Thank you for your comment and suggestion. **We have already added the discussion in Appendix A. Extended Related Works (Section A.2) of our revised manuscript**, available at: [extended_appendix.pdf](https://anonymous.4open.science/r/Conflict-Aware-Additive-Guidance-for-Flow-Models-under-Compositional-Rewards-B484/extended_appendix.pdf). We kindly encourage reviewers to revisit our rebuttal and revised manuscript. We remain fully available to address any further questions, and we hope our clarifications will convince the reviewer to reconsider their score and acknowledgement.
> > >
> > >
> > > We summarize the value gradient part briefly here:
> > >
> > > # Appendix A. Extended Related Works
> > > ## A.2 Value gradient guidance
> > > A related line of work defines the guidance signal as the gradient of a learned value function $g(x_t, t) \triangleq \nabla_{x_t} V(x_t, t)$, where $V(x_t, t) \approx \mathbb{E}[r(x_1) \mid x_t]$ summarizes the expected terminal reward. VGG-Flow (Liu et al., 2025b) instantiates this idea by identifying the gap between the pretrained and fine-tuned velocity fields as the value gradient — it co-trains a value gradient network via an HJB consistency loss alongside the fine-tuned velocity, using each to supervise the other. As a fine-tuning method, VGG-Flow couples the model to a single fixed reward at training time and does not address off-manifold drift: the learned value gradient is a general, unconstrained vector field with no mechanism to correct trajectories that have already deviated from the data manifold. In contrast, our $g^\text{car}$ targets the failure mode of inference-time guidance methods, where off-manifold drift arises specifically in regions of gradient conflict — when multiple reward gradients point in opposing directions, their sum vanishes or points into low-density regions, trapping trajectories away from the data manifold. Rather than fine-tuning the model, $g^\text{car}$ learns a lightweight residual correction that is selectively activated in these conflict regions at inference time, redirecting trajectories back onto the data manifold without modifying pretrained weights, and can be applied on top of any heuristic approximate guidance.
> > >
> > > We really appreciate the reviewer's valuable time and effort in helping us strengthen this work.

---

### Official Review · Reviewer_sXnK · 2026-03-14

**Soundness:** 3
**Presentation:** 3
**Significance:** 3
**Originality:** 3
**Overall Recommendation:** 4
**Confidence:** 4

**Summary:**

This paper studies inference-time guidance for flow models when multiple rewards are composed and their gradients conflict. The main claim is that off-manifold drift in this setting is driven by local gradient misalignment, and the proposed method detects those conflict regions and learns a residual guidance field to correct them. Across synthetic data, Maze2D, ManiSkill2, and text-guided image manipulation, the method looks broadly effective, with Table 1 giving the clearest evidence.

**Compliance With Llm Reviewing Policy:**

Affirmed.

**Final Justification:**

The rebuttal addressed my concern

**Key Questions For Authors:**

see weakness and:
- How sensitive is `gcar` to the conflict threshold `τ` in the image-editing and robotics settings?
- In the high-dimensional tasks where convergence is harder, what are the most common failure modes of the learned residual guidance?
- How well does  work when the reward models themselves are noisy or slightly misaligned with the true constraints?

**Limitations:**

Partly. The paper does acknowledge convergence issues in complex reward landscapes, but it should say more clearly when the learned correction fails and how sensitive the method is to its conflict-detection hyperparameters.

**Strengths And Weaknesses:**

Pro:
- The paper isolates a real failure mode instead of just proposing another guidance tweak. Figure 2 makes the gradient-conflict story easy to understand, and that framing carries through the rest of the paper.
- The method is tested in more than one toy domain. The paper covers synthetic data, Maze2D planning, ManiSkill2 manipulation, and CelebA-HQ face editing, which makes the idea feel less brittle.
- The image-editing results are solid. In Table 4, the method gets the best LPIPS and ID under composed prompts, which is exactly the regime where a conflict-aware method should help.

Con:
- The robotics evidence is weaker than the synthetic evidence because Tables 2 and 3 are hard to parse and the strongest claims are left in prose. The paper should make those tables complete and legible, with clear per-setting numbers for every metric, so readers can tell where the gains really come from.
- The image-editing evaluation still leans on LPIPS, ID, and CLIP, which are useful but not enough for a claim about preserving realism under conflicting edits. The paper should add a stronger perceptual check, such as human preference or a more direct artifact assessment, for the composed-prompt setting.
- The method appears most fragile exactly where it matters most: the high-dimensional settings. Section 7 admits that the guidance network has convergence issues in complex reward landscapes, so the paper should show representative failures and explain when `gcar` stops being reliable.
- The conflict threshold `τ` looks important, but sensitivity to it is not made very clear in the main paper. A small sweep and a practical rule for choosing `τ` across domains would make the method much easier to trust and use.

---

> ### Author Rebuttal · Authors · 2026-03-30
>
> Thank you for the reviewer's comments, especially the helpful suggestions on the image evaluation metrics and the ablation study on the conflict threshold $\tau$. We appreciate your recognition of the novelty and contributions of our work, and your positive assessment of our experimental results. Based on your feedback, we have provided a more thorough evaluation with stronger perceptual metrics and a systematic ablation study to help readers better understand the contribution of each component and how to select hyperparameters in practice. We hope our clarifications and analysis can convince the reviewer to reconsider their score. Please let us know if further clarification is required.
>
> ---
>
> ## R1 to W1: Redesign Tables
>
> We have redesigned the tables to clearly show the improvements of $g^{\text{car}}$ ([link](https://anonymous.4open.science/r/Conflict-Aware-Additive-Guidance-for-Flow-Models-under-Compositional-Rewards-B484/rebuttal_images_tables.pdf)), and added intuitive visualizations: robot planning in [Figure](https://anonymous.4open.science/r/Conflict-Aware-Additive-Guidance-for-Flow-Models-under-Compositional-Rewards-B484/Image_maze2d.jpg) and robot manipulation in [Figure](https://anonymous.4open.science/r/Conflict-Aware-Additive-Guidance-for-Flow-Models-under-Compositional-Rewards-B484/Image_maniskilll.jpg). For a thorough comparison, please refer to "R1 to Q2 & W3: Baseline Overview and GLASS-FKS" in our response to Reviewer vjZE.
>
> ---
>
> ## R2 to W2: Image Evaluation Metrics
>
> We have strengthened the perceptual evaluation by incorporating two text-image alignment metrics (BLIP-ITM (Li et al., 2022) and VQAScore (Lin et al., 2024)) and one image quality metric (CLIP-IQA, Wang et al., 2023). BLIP-ITM is a strict binary image-text matcher; VQAScore evaluates compositional reasoning via LLaVA-1.5; CLIP-IQA penalizes blurry or artifact-heavy generations. Results at [CelebaHQ Table](https://anonymous.4open.science/r/Conflict-Aware-Additive-Guidance-for-Flow-Models-under-Compositional-Rewards-B484/Table_CelebaHQ.png) and [full results](https://anonymous.4open.science/r/Conflict-Aware-Additive-Guidance-for-Flow-Models-under-Compositional-Rewards-B484/rebuttal_images_tables.pdf)).
>
> The key observations are: (1) FlowGrad and PCGrad both fail to balance multiple constraints, as evidenced by the large gap between $P_1$ and $P_2$ scores. (2) $g^{\text{cov-G}}$ is prone to off-manifold drift. (3) GLASS-FKS fails to preserve the reference image due to high variance. (4) Our $g^{\text{car}}$ achieves superior compositional alignment across prompts, corrects off-manifold drift, and eliminates the visual artifacts observed in $g^{\text{cov-G}}$.
>
> ---
>
> ## R3 to W3 & Q2: Convergence
>
> We've added a failure analysis in [this link](https://anonymous.4open.science/r/Conflict-Aware-Additive-Guidance-for-Flow-Models-under-Compositional-Rewards-B484/rebuttal_images_tables.pdf).
>
> Non-smooth reward landscapes may cause convergence instability. As shown in `happy + sad`, the trajectory zigzags severely in PCA space and $g_\psi$ receives an inconsistent training signal, producing artifacts. A practical remedy is to monitor the per-sample loss curve: a non-converging or oscillating loss signals that $g_\psi$ should not be trusted.
>
> ---
>
> ## R4 to W4 & Q1: Conflict Threshold $\tau$
>
> We have added "Appendix F.2: hard weight $\mathbb{I}_t$ and conflict threshold $\tau$" to provide a more thorough analysis of its sensitivity and practical selection guidelines.
>
> $\tau$ serves two purposes: (1) skipping low-conflict regions to reduce unnecessary computation, and (2) preserving $g^{\text{approx}}$ where it is already accurate, avoiding spurious perturbations. Too small a $\tau$ activates the gate almost everywhere, defeating both purposes; too large a $\tau$ leaves $g_\psi$ under-trained.
>
> We find $\tau = 0.2$ to be a robust default across all tasks. Full results are available at [this link](https://anonymous.4open.science/r/Conflict-Aware-Additive-Guidance-for-Flow-Models-under-Compositional-Rewards-B484/rebuttal_images_tables.pdf).
>
> ---
>
> ## R5 to Q3: Robustness to Noisy Reward Models
>
> A noisy reward model will generally increase the approximation error, as analysed in our error bound (Section 4): the error scales with $\mu = |\nabla^2 r|_2$, so a smoother, less noisy reward landscape leads to a better guidance.
>
> A more intuitive philosophy underlying our work is: suppose there exists a point x* on the data manifold that is a minimum for both Reward A and Reward B; at such a point, the gradients of the two rewards should be almost well-aligned. So basically, $g^\text{car}$ is trying to retrieve the x*, where the gradients of Reward A and Reward B are almost well-aligned, and also we have conflict threshold $\tau$ further tolerates a certain degree of misalignment (may be caused by noisy reward signals).

---

> > ### Author Rebuttal · Reviewer_sXnK · 2026-04-03
> >
> > my concerns are solved, i will raise my score

---

> > > ### Author Response · Authors · 2026-04-05
> > >
> > > We really appreciate the reviewer's valuable time and effort in helping us strengthen this work.

---

### Decision · Program_Chairs · 2026-04-30

**Decision:**

Accept (regular)

**Comment:**

This submission considers inference-time guidance for flow models in settings with multiple composed (summed) rewards. The authors identify misalignment between competing rewards as a source of off-manifold drift during generation. The proposed method, conflict-aware additive guidance, detects conflict regions via pairwise cosine similarity of reward gradients, and interpolates toward a learned residual correction (trained via terminal value regression) in regions where conflicts are detected. Evaluations consider 2D synthetic benchmarks, robot planning (Maze2D), robot manipulation (ManiSkill2) and text-guided image editing (CelebA-HQ).

After author-reviewer discussion all reviewers arrived at a weak accept score. Reviewers noted that the problem of guidance with multiple conflicting rewards is a real and practically relevant failure mode, found the error bound in theorem 4.2 technically meaningful and insightful, and overall were positive about motivation, presentation and evaluations. Comments about weaknesses varied from reviewer to reviewer, and the author response addressed a number of issues, including addition of GLASS-FKS and PCGrad baselines, additional image evaluation baselines, variance across 5 seeds, and additional appendices. Overall this seems a submission where no reviewer is strongly championing acceptance, but also one that does not have any issues that would preclude it from appearing, so the area chair will recommend acceptance.